# Deciphering neuronal variability across states reveals dynamic sensory encoding

Shailaja Akella [1] ✉, Peter Ledochowitsch[1,4], Joshua H. Siegle [1], Hannah Belski[1], Daniel D. Denman [1,5], Michael A. Buice [1], Severine Durand [1], Christof Koch [1], Shawn R. Olsen [1] & Xiaoxuan Jia [2,3] ✉

Influenced by non-stationary factors such as brain states and behavior, neurons exhibit substantial response variability even to identical stimuli. However, it remains unclear how their relative impact on neuronal variability evolves over time. To address this question, we designed an encoding model conditioned on latent states to partition variability in the mouse visual cortex across internal brain dynamics, behavior, and external visual stimulus. Applying a hidden Markov model to local field potentials, we consistently identified three distinct oscillation states, each with a unique variability profile. Regression models within each state revealed a dynamic composition of factors influencing spiking variability, with the dominant factor switching within seconds. The state-conditioned regression model uncovered extensive diversity in source contributions across units, varying in accordance with anatomical hierarchy and internal state. This heterogeneity in encoding underscores the importance of partitioning variability over time, particularly when considering the influence of non-stationary factors on sensory processing.

The amount of information a sensory neuron carries about external stimuli is reflected in its repeated activity pattern in response to the same stimuli[1]. However, trial-to-trial variability, ubiquitous in the nervous system[2], constrains the amount of sensory information in single-trial neural responses to the stimulus. It follows that the time course of this variance mimics the highly non-stationary dynamics of the underlying neuronal processes[3,4]. For example, when animals actively explore their environment, the sensory cortex shows desynchronized responses in a manner that increases their responsiveness to stimuli[5]. Conversely, during periods of sleep or quiet wakefulness, cortical neurons tend to synchronize their activity, resulting in decreased sensitivity to external stimuli[6]. Dissecting these non-stationary dynamics is critical to comprehending their role in information encoding and ultimately, perception.

Even with well-controlled experiments and behavior-monitoring techniques[7,8], understanding how neuronal variability changes over time is challenging[9]. This is further complicated by the high-dimensional interactions between the various sources of neuronal variability: external stimuli, behavior, and internal brain dynamics[10]. To address this complexity, a common strategy involves the identification of meaningful temporal patterns and potential latent variables that can capture the evolving dynamics of neural activity. These patterns, which accurately capture the internal brain dynamics, are typically referred to as "brain states"[5,11–13].

Brain states, characterized by distinct patterns of neural activity and functional connectivity, play a pivotal role in shaping the dynamics of neuronal variability[6,12], influencing how sensory information is processed[4,14] and behaviors are executed[5,11]. For instance, during heightened attention, decreases in the correlations between the trial-to-trial fluctuations in the responses of pairs of neurons serve to enhance the signal-to-noise ratio of the entire population, improving behavior[15]. Likewise, several studies have shown that random fluctuations in the processing of sensory stimuli originate from rapid shifts in the animal's arousal state[11,16]. Tightly linking internal brain dynamics to

[1]Allen Institute, Seattle, WA, USA. [2]School of Life Science, Tsinghua University, Beijing, China. [3]IDG/McGovern Institute for Brain Research, Tsinghua University, Beijing, China. [4]Present address: Canaery, Alachua, FL, USA. [5]Present address: Anschutz Medical Campus School of Medicine, University of Colorado, Aurora, CO, USA. ✉e-mail: shailaja.akella@alleninstitute.org; jxiaoxuan@gmail.com

behavior, brain states serve as an ideal temporal framework to study the dynamics of neuronal variability.

Recently, researchers have leveraged advanced machine-learning tools to explain single-trial neural activity by incorporating extensive stimulus and behavioral features[17–19]. While these studies reveal the multi-dimensional nature of neuronal variability, they often assume that neuronal variability remains constant over time. To address this gap, several parallel lines of research have used latent dynamical models to study the temporal patterns of neuronal variability[5,12,20,21]. However, these studies have not explicitly explored the different sources contributing to variability, as it changes over time. Consequently, our understanding of how various sources dynamically contribute to the non-stationarity of neuronal variability remains limited (Fig. 1A).

Here, we present a comprehensive investigation on how internal and external factors collectively shape the time course of neuronal variability to influence sensory coding. We used the Allen Brain Observatory Visual Coding dataset, which comprises simultaneous recordings of local field potentials (LFPs) and spiking activity from hundreds of Neuropixels channels in multiple visual areas along the anatomical hierarchy[22]. As mice passively viewed natural movies, we applied Hidden Markov Models (HMMs)[23] on LFP data extracted from six visual cortical regions to establish a global temporal framework of internal latent states. Quantifying various aspects of variability across individual trials and neuronal populations, we uncovered significant changes in neuronal variability across states. These findings indicated dynamic shifts in the efficiency of sensory processing over time. To disentangle the sources of non-stationary sensory processing, we designed a neural encoding framework conditioned on internal states to partition variability across three crucial factors: internal brain dynamics, spontaneous behavior, and external visual stimuli. Through this model, we quantified the time-varying contributions of these sources to single-trial neuronal and population dynamics. Our findings revealed that, even during persistent sensory drive, neurons

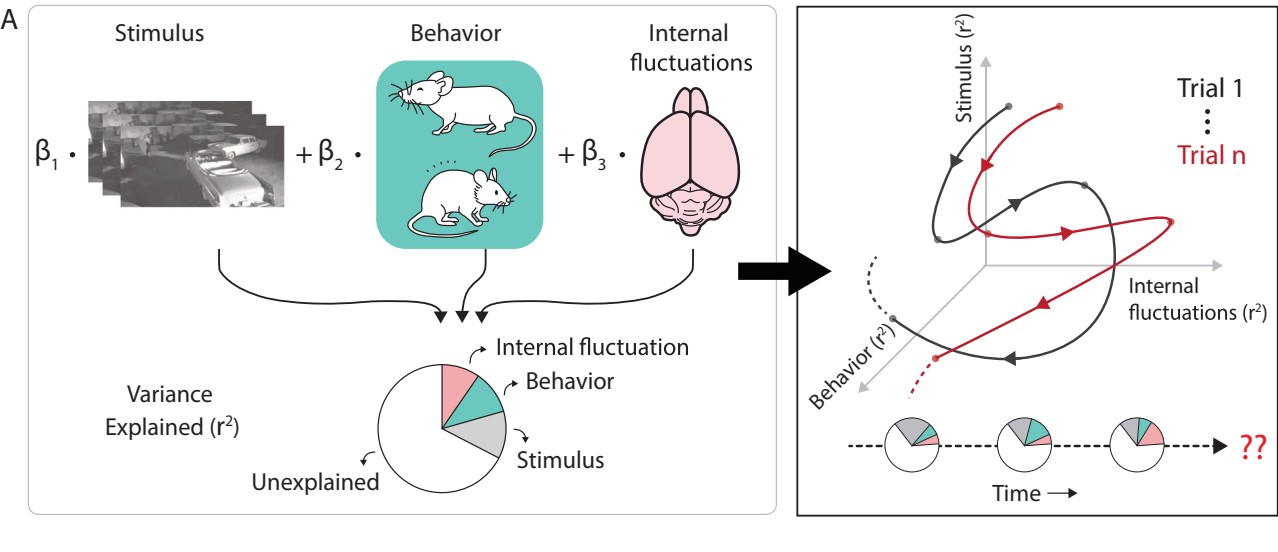

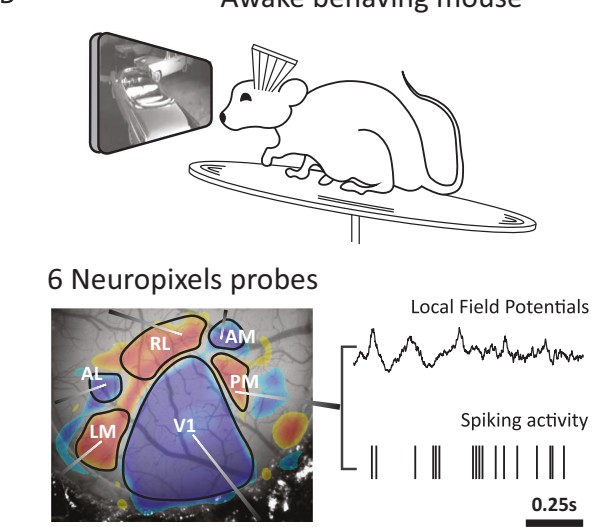

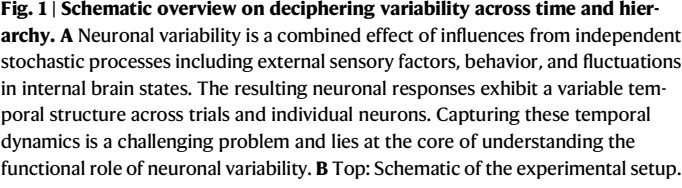

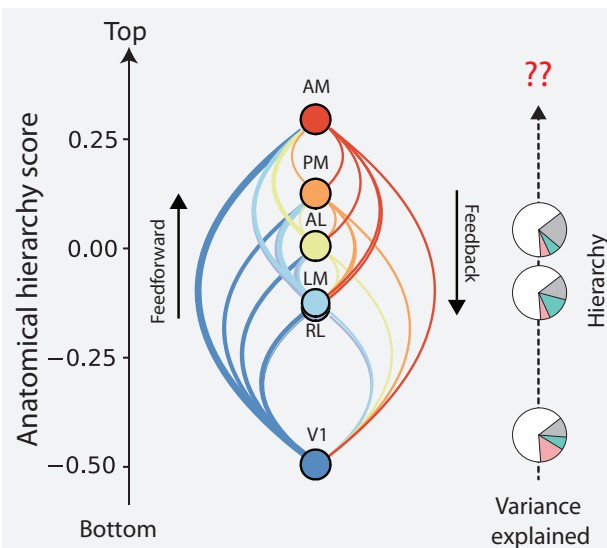

**Fig. 1 | Schematic overview on deciphering variability across time and hierarchy. A** Neuronal variability is a combined effect of influences from independent stochastic processes including external sensory factors, behavior, and fluctuations in internal brain states. The resulting neuronal responses exhibit a variable temporal structure across trials and individual neurons. Capturing these temporal dynamics is a challenging problem and lies at the core of understanding the functional role of neuronal variability. **B** Top: Schematic of the experimental setup. Bottom: Neuropixels probes in six visual cortical areas simultaneously record local field potentials and spiking activity. A retinotopic sign map overlaid on the vasculature image guides area-specific targeting. **C** Anatomical hierarchy scores of the six visual areas recomputed from ref. 25. Studying variability along the visual hierarchy can reveal important insights about information propagation and encoding at each stage of signal processing.

dramatically changed the degree to which they were impacted by sensory and non-sensory factors within seconds. Additionally, we observed considerable diversity in neural encoding across visual cortical units, with the relative influence of these sources varying based on their anatomical location and cell type. Taken together, our results provide compelling evidence for the dynamic nature of sensory processing, while emphasizing the role of latent internal states as a dynamic backbone of neural coding.

## Results

We analyzed the publicly available Allen Brain Observatory Neuropixels dataset, previously released by the Allen Institute[22]. This dataset comprises simultaneous recordings of spiking activity and local field potentials (LFPs) from six interconnected areas in the visual cortex of mice ($n = 25$) passively viewing a variety of natural and artificial visual stimuli (Fig. 1B). To estimate the dynamic nature of internal state fluctuation during sensory processing, we focused our analysis on data recorded during repeated presentations of a 30 s natural movie. We used a continuous stimulus to mitigate sudden transients in activity induced by abrupt changes in the visual stimuli. Lastly, the application of quality control metrics yielded, on average, $304 \pm 83$ (mean ± std) simultaneously recorded neurons distributed across layers and areas per mouse (see Methods).

Previous studies[22,24] demonstrated that the functional hierarchy of visual areas aligns with their anatomical organization[25]. This hierarchy places the primary visual cortex (V1) at the bottom, followed by rostrolateral (RL), lateromedial (LM), anterolateral (AL), posteromedial (PM), and anteromedial (AM) areas (Fig. 1C). Here, we consider this visual hierarchy as a first-order approximation of signal processing stages to study signal propagation and information encoding while crucially accounting for the non-stationarity in spiking variability that arises due to influences from fluctuating internal and external factors.

### Identification of oscillation states from local field potentials

Internal brain states can vary without clear external markers, making their quantification challenging. To capture state changes associated with internal processes, we employ a definition of brain states derived using LFPs recorded invasively from six visual areas[22]. LFPs reflect aggregated sub-threshold neural activity and capture the highly dynamic flow of information across brain networks[26]. The spectral decomposition of LFPs reveals different frequency bands that correlate with specific cognitive states[27–29], sensory processing[30–34], and behavior[35–37]. We found that LFPs in the mouse visual areas also revealed a distinct frequency spectrum across time, whose dynamics were strongly coupled to arousal-related behavioral variables (Fig. 2A). Accordingly, we envisioned that a latent state model could reflect the underlying latent brain dynamics by capturing the dynamic patterns of the LFP spectrum, such that each latent state reflects an oscillation state. To extract these oscillation states from LFPs in the visual area, we employ Hidden Markov modeling[23,38,39] on filtered envelopes of LFPs within distinct frequency bands (Fig. 2B, left panel): 3–8 Hz (theta), 10–30 Hz (beta), 30–50 Hz (low gamma), and 50–80 Hz (high gamma). This approach enabled us to fully capture LFP power across the 3–80 Hz frequency range (Supplementary Fig. S1A), while also aligning with the observed frequency boundaries in the spectral decomposition of LFPs. Finally, to capture laminar dependencies, the observations supplied to the HMM also comprised LFPs from superficial, middle and deep layers in all visual areas (one channel each from layer 2/3, layer 4, layer 5/6; Fig. 2B (middle panel), Supplementary Fig. S1E, F).

We found that LFP dynamics in the visual cortex consistently unfolded through three reliable oscillation states across all mice (see Methods; Fig. 2C, $3.08 \pm 0.39$ states, $n = 25$ mice, mean ± std). These states did not depend on stimulus types (Supplementary Fig. S4A, B), specific visual areas (Supplementary Fig. S1B, C), or layers

(Supplementary Fig. S1E, F). The identity of the inferred states was also remarkably consistent across mice, each characterized by a distinct distribution of the power spectrum: a high-frequency state ($S_H$), a low-frequency state ($S_L$), and an intermediate state ($S_I$). While the high-frequency state is characterized by increased power in the low and high gamma bands, slow oscillations dominate the low-frequency state dynamics in the theta frequency ranges (Fig. 2D, E, Supplementary Fig. S2A). LFP power distribution in the intermediate state is more uniform.

These oscillation states demonstrate stable dynamics, as reflected by the large values along the diagonal of the transition matrix, ranging between 0.94 and 0.99 (Supplementary Fig. S3B). Dwell time in a state averaged around $1.5 \pm 0.14$ s (mean ± sem, $n = 3$ states) (Fig. 2F), and the transition intervals between consecutive states (the interval around a transition during which the HMM posterior probability is < 80 %) were significantly shorter than the dwell times, lasting only for about $0.13 \pm 0.006$s (mean ± sem). Additionally, direct transitions between the low- and high-frequency states were rare and required transitioning through the intermediate state, as evident in both two- and three-step transition sequence-probability trends (Fig. 2G). Consequently, mice spent only short durations in the intermediate state ($0.97 \pm 0.001$ s, mean ± sem), while they spent the most prolonged durations in the high-frequency state ($1.92 \pm 0.003$s, mean ± sem, $p_{S_H, S_I} = 1.17$e-167, $p_{S_H, S_L} = 6.6$e-79, $p_{S_I, S_L} = 1$e-11, one-way ANOVA, $n = 25$ mice). Notably, this state property was dependent on stimulus type (Supplementary Fig. S4C). During repeated presentations of the drifting grating stimulus, transitions between the extreme states of low- and high-frequency were much faster and more likely (Supplementary Fig. S4E, F). This significantly reduced the amount of time mice spent in the intermediate state ($0.25 \pm 0.0001$ s, $p = 1.5$e-120, one-way ANOVA, Supplementary Fig. S4C). However, in the absence of any stimulus, mice tended to spend longer durations in the intermediate state ($1.16 \pm 0.001$ s, $p = 3.5$e-29, one-way ANOVA). We attribute these differences to the strong neural responses evoked by sudden transitions of the visual stimulus such as, the onset and offset of drifting gratings stimuli.

### Correlation between oscillation states and body movements

Brain state variations often exhibit strong correlations with the animal's behavioral context[40,41]. Indeed, several studies have reported neural activity changes in the visual cortex associated with various behavioral features[17,42,43]. To this end, we examined the behavioral correlates of the oscillation state patterns, comparing pupil size, running speed, and facial, limb, and tail movements across different states (Fig. 3A–C). Our investigation revealed a strong association between behavioral movements and internal oscillation states across subjects (Fig. 3E). Notably, a shift to the high-frequency state corresponded closely with increased movements and pupil size (Fig. 3D), suggesting increased arousal levels in this state. Conversely, mice tended to be at rest in the low-frequency state while only making small movements in the intermediate state[11,44,45].

Several studies have considered locomotion as an indicator of brain state to examine variations in visual encoding[18,46]. To quantify the relationship between internal oscillation states and different behavioral features, we calculated the mutual information (MI) between the states and each behavioral feature[47]. We found that changes in the oscillation states were more faithfully mimicked by pupil size or facial movements (Fig. 3D), reporting significantly higher MI than all other behavioral responses ($MI_{pupil} = 0.12 \pm 0.006$, $MI_{face} = 0.1 \pm 0.006$, mean ± sem, $n = 25$ mice), including running ($MI_{running} = 0.08 \pm 0.007$, mean ± sem, $n = 25$ mice, Fig. 3F). This held true despite the strong positive correlations between all behavior variables ($r = 0.4 \pm 0.03$, mean ± sem, $n = 25$ mice), and especially between running, facial movement, and pupil size ($r = 0.6 \pm 0.04$, mean ± sem, $n = 25$ mice). Importantly, all behaviors associated with running (movements in the

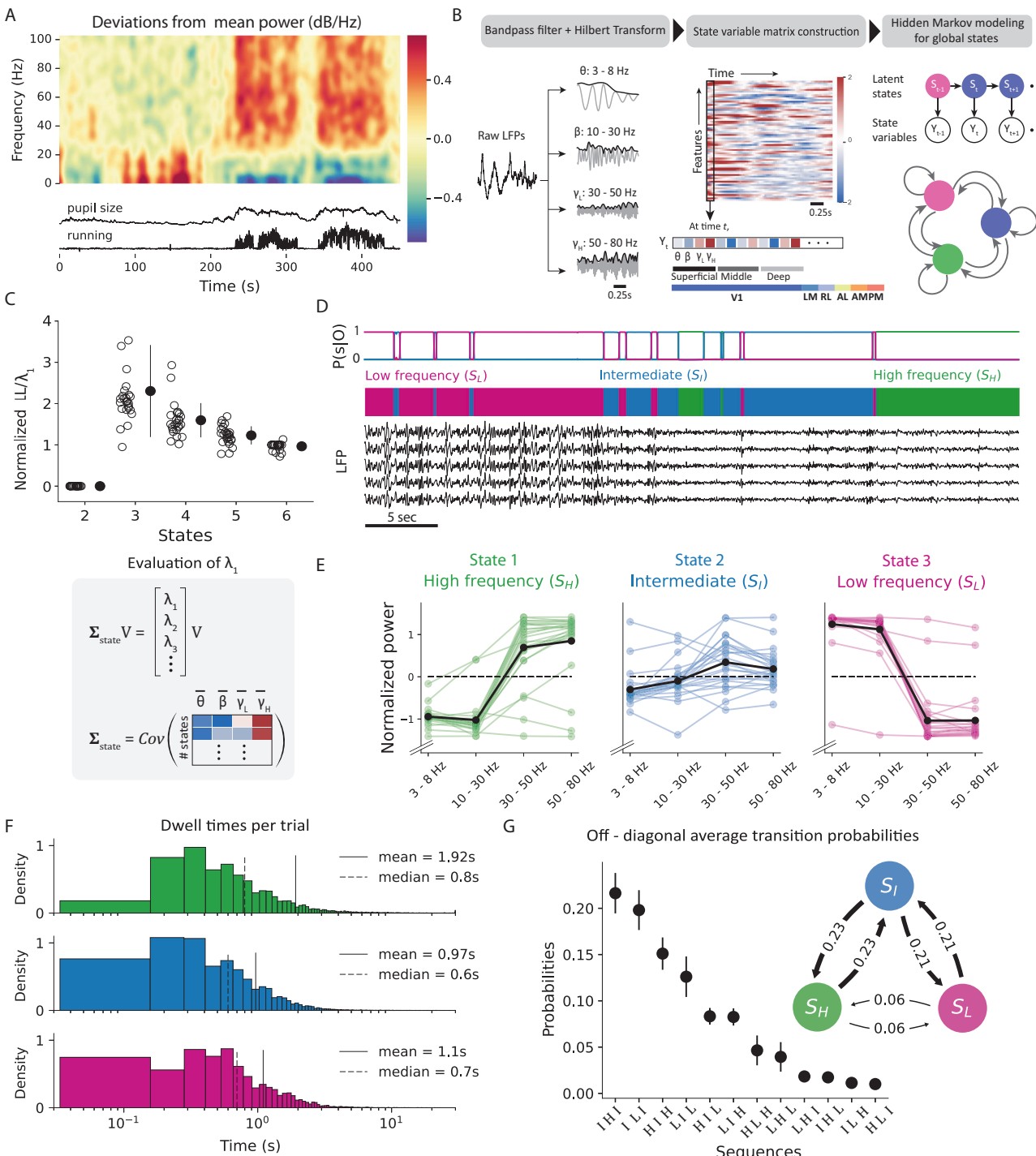

**Fig. 2 | Properties of internal oscillation states identified from local field potentials in awake behaving mice. A** Top: Local field potential (LFP) power modulations in V1 recorded from mice passively viewing a naturalistic movie. Bottom: Time course of running speed and pupil area during the same time period. **B** Schematic to identify oscillation states using LFPs. Discrete states are defined based on frequency-specific transients of LFPs from six visual areas. Hidden Markov model (HMM) uses Hilbert transforms in the theta (3–8 Hz), beta (10–30 Hz), lower gamma (30–50 Hz), and higher gamma (50–80 Hz) frequency ranges. **C** Top: Model comparison among HMMs over a range of latent states using three-fold cross-validation. The cross-validated log-likelihood (LL) estimate, normalized by the top eigenvalue of the state definition matrix, is reported for each mouse (hollow circles) along with across-subject averages (solid circles, $n = 25$ mice, error bars represent s.e.m.). For each mouse, the optimal number of states was identified as the point where the normalized LL was maximized. Final model selection was based

on the majority rule across all mice. Bottom: Evaluation of state similarity ($\lambda_1$) as the top eigenvalue of the state definition matrix. **D** Top: State posterior probabilities identified by the HMM. Bottom: LFPs from randomly selected channels from V1, displayed alongside their respective latent states over the same duration. **E** LFP power distribution in the three-state model. Shaded lines represent the state-specific z-scored power distributions in individual mice, and the solid black line represent the average across all mice (N = 25 mice). In state-1, or the high-frequency state, LFPs are dominated by high-frequency gamma oscillations. State 3, or the low-frequency state, has characteristic slow oscillations in the theta band. **F** Histogram of state dwell times in each trial across all states and all mice. **G** Average probability of observing 3-step or 2-step (inset) transition sequences to different states. Transition probabilities were calculated from observed sequences averaged across all mice ($n = 25$, error bars represent s.e.m.). Source data are provided as a Source Data file.

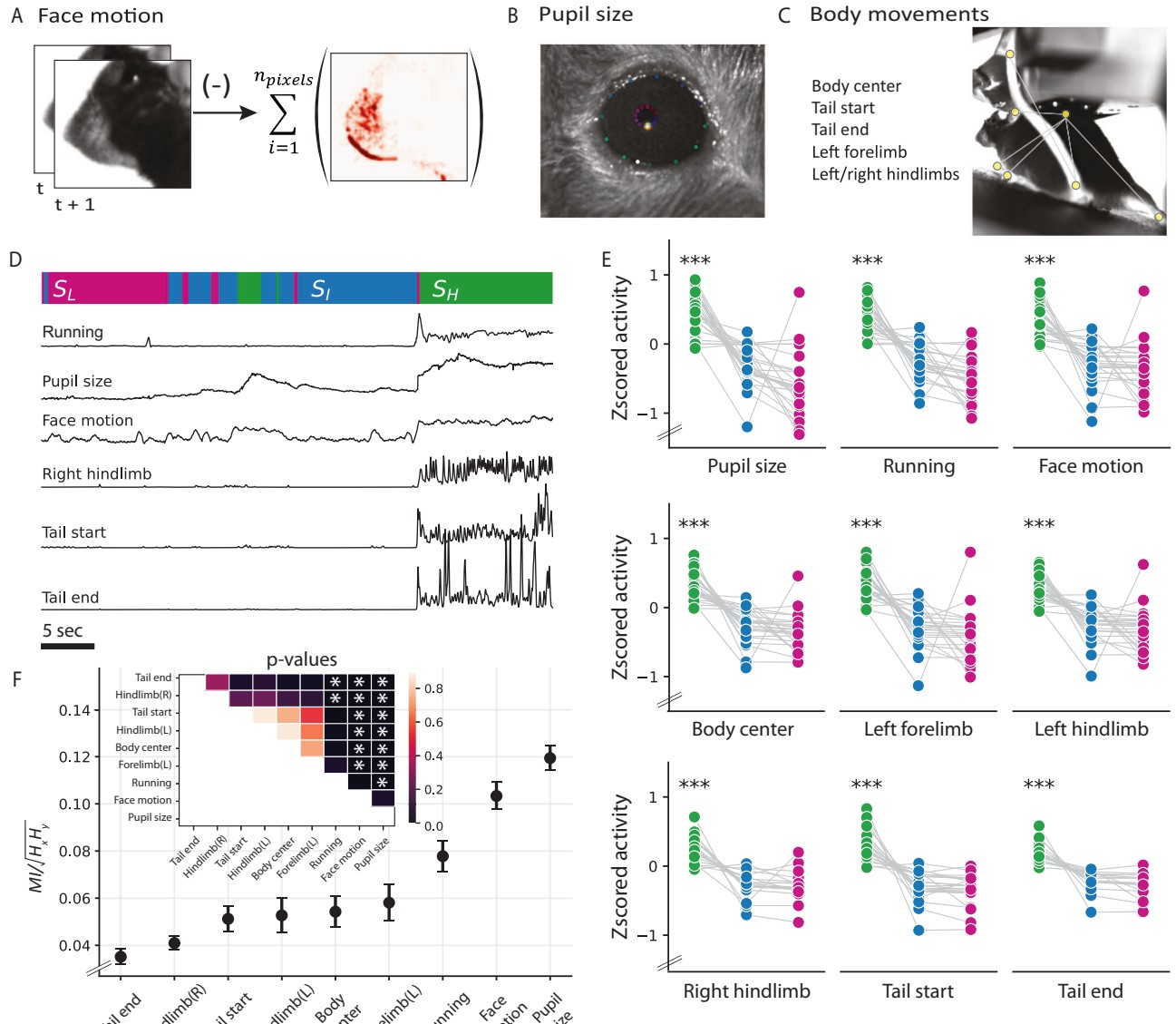

**Fig. 3 | Behavioral correlates of the observed oscillation states. A** Face motion energy evaluated as the absolute value of the difference between consecutive frames. **B** Eye and pupil tracking. Tracking points were identified using a universal tracking model trained in DeepLabCut. **C** Animal pose estimation. Specific, visible body parts were tracked using a universal tracking model trained in SLEAP. **D** Example snippet of behavioral changes alongside the animal's current oscillation state. $S_H$: High-frequency state (green), $S_I$: Intermediate state (blue), and $S_L$: Low-frequency state (pink). **E** Comparison of the average movement of specific body

parts across states ($p_{S_H, S_{L,L}}$, pupil size: $p = 2.8e$-15, running: $p = 2.0e$-17, face motion: $p = 6.3e$-13, body center: $p = 2.6e$-18, left forelimb: $p = 1.2e$-13, left hindlimb: $p = 4.9e$-14, right hindlimb: $p = 3.0e$-11, tail start:, $p = 3.0e$-16, tail end: $p = 2.0e$-11, $n = 25$ mice, one-way ANOVA). **F**, Mutual information (MI) between behavioral variables and the inferred HMM states (mean ± sem, $n = 25$ mice). All statistical tests were performed using one-way ANOVA. Statistical tests in (**E**, **F**) were adjusted for multiple comparisons using the Bonferroni correction (***: $p < 0.0001$, **: $p < 0.001$, *: $p < 0.05$). Source data are provided as a Source Data file.

proximal end of the tail, left limbs, and body center) reported similar MI with the oscillation states. To further validate these results, we used HMMs to quantify behavioral states in individual mice, fitting individual models to pupil size, face motion, and running measures. Upon comparing these behavioral states with oscillation states, stronger correlations emerged with pupil size and face motion than with running speed (Supplementary Fig. S5B; $p = 0.0007$, one-way ANOVA, $n = 25$ mice). We attribute these differences to the dissociation between pupil size and running speed, particularly in cases where pupil dilation occurs, even when the mouse remains stationary (Supplementary Fig. S5A). These results suggest that facial movements serve as a reliable representation of the underlying internal states reflected in voluntary behavior, almost as good as the involuntary changes in pupil size[48].

## Neuronal variability changes across oscillation states and visual hierarchy

After defining the internal oscillation states and establishing their relation to behavior and arousal state, we wondered how spiking variability changes across these states. Across states, we observed distinct variations in population activity and synchronization levels (Fig. 4A–C). Consistent with previous observations of attentional effect[15], increased spiking activity (av. % increase = 7.7 ± 1.6, mean ± sem, $p = 6.3e$-5, pairwise $T$-test, $n = 25$ mice) and decreased correlation (av. % decrease = 36.6 ± 3.4, mean ± sem, $p = 1.3e$-10, pairwise $T$-test, $n = 25$ mice) were typical of the high-frequency state. Moreover, the transition-state-like properties of the intermediate state were broadly consistent across various neuronal properties (Fig. 4B, C) and behavior (Fig. 3E). Bolstered by these findings, we evaluated three types of

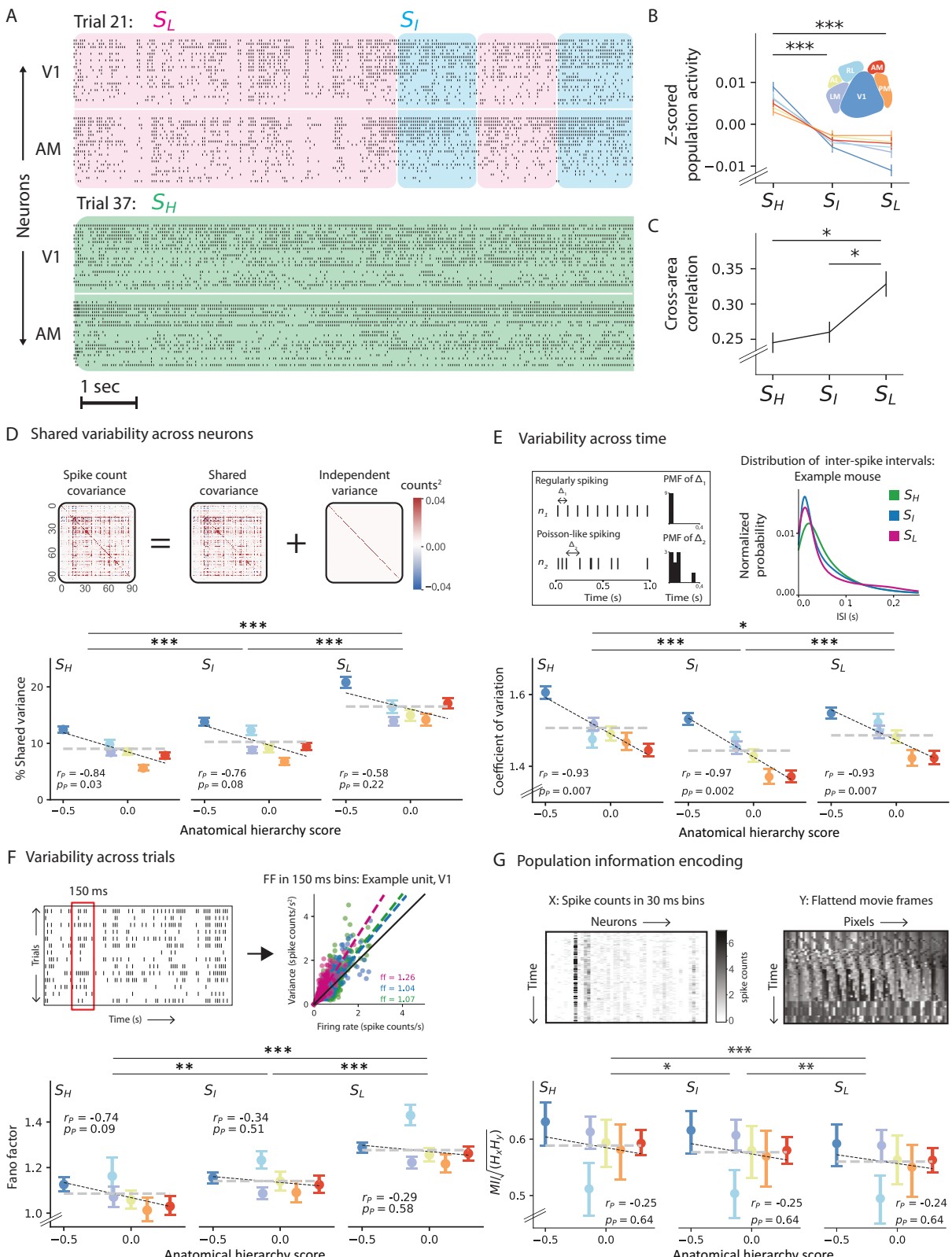

variability in single neurons to capture complementary aspects of neuronal variability: percentage of shared variance within a population, spike timing variability, and variability in spike counts across trials.

Previous studies have shown that variability shared within a neuronal population can constrain information propagation between processing stages[49–52]. This is because shared variance within a population may not average out[53,54], leading to a deterioration of the population's coding capacity. To study how shared variability evolves across various internal states, we used factor analysis (FA)[55] to partition the spike count variability into its shared and independent components (Fig. 4D, top). Within a neuronal population, the shared component quantifies co-fluctuations in firing rates among individual neurons, while the independent component

**Fig. 4 | Neuronal variability and information encoding across states and the visual hierarchy. A** Raster plots (-10 s) showing the response of 25 units, each from V1 and AM, during two trials in which the mouse was in different states. Each row represents the activity of the same single neuron across the two trials. $S_H$: High-frequency state (green), $S_I$: Intermediate state (blue), and $S_L$: Low-frequency state (pink). **B** State and area-specific population activity, z-scored and averaged across all mice ($p_{S_H, S_I} = 1.4e{\text -}05$, $p_{S_H, S_L} = 3.0e{\text -}07$, $p_{S_I, S_L} = 0.90$, one-way ANOVA, $n = 25$ mice). Error bars represent s.e.m. **C** Average pairwise correlation between averaged neuronal population activity in different visual areas as a function of oscillation states ($p_{S_H, S_I} = 1.5$, $p_{S_H, S_L} = 0.002$, $p_{S_I, S_L} = 0.002$, one-way ANOVA, $n = 25$). Error bars represent s.e.m. **D** Population shared variance. Top: Separation of shared and independent variance using factor analysis (FA). FA partitions the spike count covariance matrix into shared and independent components. Bottom: Percentage of shared variance plotted against the anatomical hierarchy scores of the visual areas in each oscillation state, averaged across all units (One-way ANOVA: $p_{S_H, S_I} = 9.6e{\text -}7$, $p_{S_H, S_L} = 1.3e{\text -}146$, $p_{S_I, S_L} = 1.2e{\text -}95$; Two-way ANOVA, states: $F = 431.2$, $p = 1.5e{\text -}189$, areas: $F = 78.8$, $p = 3.3e{\text -}82$, states × area: $F = 3.3$, $p = 2.6e{\text -}4$, $n = 7609$ units). **E** Neuronal variability across time, quantified using the coefficient of variation (CV). Top-left: Simulated distributions of inter-spike-intervals (ISI) for regular and Poisson-like firing. For a very regular spike train, a narrow peak in the ISI histogram corresponds to CV ≈ 0, whereas Poisson-like variability in the spike trains leads to an exponentially distributed ISI histogram with CV = 1. Top-right: Distribution of ISIs in each oscillation state over a 2.5 sec range. Bottom: CV along the visual hierarchy (quantified as anatomical hierarchy scores) and across oscillation

states, averaged across all units (One-way ANOVA: $p_{S_H, S_I} = 4.9e{\text -}23$, $p_{S_H, S_L} = 3.9e{\text -}03$, $p_{S_I, S_L} = 2.8e{\text -}11$; Two-way ANOVA, states: $F = 42.5$, $p = 3.6e{\text -}19$, areas: $F = 88.1$, $p = 4.5e{\text -}92$, states × area: $F = 4.8$, $p = 4.9e{\text -}7$, $n = 7609$ units). **F** Neuronal variability across trials, quantified using Fano factor (FF). Top-left: Evaluation of FF as an average of the FF ratio over non-overlapping windows of 150 ms with at least ten trials in each state. Top-right: Mean spike count versus variance over all times in each state for an example cell in V1. Bottom: FF along the visual hierarchy and across brain states, averaged across all units (One-way ANOVA: $p_{S_H, S_I} = 2.8e{\text -}4$, $p_{S_I, S_L} = 2.4e{\text -}33$, $p_{S_I, S_L} = 3.5e{\text -}39$; Two-way ANOVA, states: $F = 107.7$, $p = 7.5e{\text -}47$, areas: $F = 7.1$, $p = 9.9e{\text -}6$, states × area: $F = 0.6$, $p = 0.8$, $n = 5017$ units). Pearson correlation with hierarchy scores excluding RL, $S_H$: $r_{p{\text -}RL} = -0.94$, $p_{p{\text -}RL} = 0.02$; $S_I$: $r_{p{\text -}RL} = -0.43$, $p_{p{\text -}RL} = 0.5$; $S_L$: $r_{p{\text -}RL} = -0.46$, $p_{p{\text -}RL} = 0.43$. **G** Information encoding along the visual hierarchy across all oscillation states, quantified using mutual information (MI). Top: For each trial, MI was evaluated between the population spike count matrix and a matrix of flattened movie frames at time points corresponding to each state using a matrix-based entropy estimator. Bottom: MI across the visual hierarchy and oscillation states averaged across all mice (Pairwise T-test: $p_{S_H, S_I} = 0.01$, $p_{S_I, S_L} = 7.3e{\text -}10$, $p_{S_I, S_L} = 9.3e{\text -}04$; Two-way ANOVA, states: $F = 3.1$, $p = 0.04$, areas: $F = 2.7$, $p = 0.03$, states × area: $F = 0.02$, $p = 0.99$, $n = 25$). Pearson correlation with hierarchy scores excluding RL, $S_H$: $r_{p{\text -}RL} = -0.9$, $p_{p{\text -}RL} = 0.03$; $S_I$: $r_{p{\text -}RL} = -0.86$, $p_{p{\text -}RL} = 0.06$; $S_L$: $r_{p{\text -}RL} = -0.81$, $p_{p{\text -}RL} = 0.09$. Error bars in **D–G** represent 95% confidence intervals. All statistical tests were adjusted for multiple comparisons using the Bonferroni correction (***: $p < 0.0001$, **: $p < 0.001$, *: $p < 0.05$). Source data are provided as a Source Data file.

captures their Poisson-like variability. Percentage of shared variability was then evaluated as the ratio between each neuron's shared and total variance. Consistent with previous findings that noted more synchronization within a population during low-arousal states[40,41], the percentage of shared variability was highest during the low-frequency state (Fig. 4D, bottom). In this state, fewer factors influenced the observed patterns of variation compared to the other states (number of FA components, $S_H = 21 \pm 1$, $S_I = 19 \pm 1$, $S_L = 16 \pm 1$, $p = 1.8e{\text -}06$, one-way ANOVA). Neurons within V1 reported a larger shared component than neurons within other areas. The percentage of shared variance decreased along the visual hierarchy in the high-frequency state, (Pearson correlation $r = -0.84$ with anatomical hierarchy score, $p = 0.03$), while the trends were not significant in the intermediate and low-frequency states ($S_L$: Pearson's $r = -0.76$, $p = 0.08$, $S_I$: $r = -0.58$, $p = 0.22$). Compared to higher visual areas, neurons in early visual areas are more strongly modulated by the temporal features of visual stimuli[22,56]. Thus, we attribute the observed decreasing trends across the visual hierarchy to the stronger modulation of neurons in lower visual areas by the temporal features of the natural movie, such as rapid variations in luminance or moving edges. This likely induces more temporally coherent activity within populations in the lower visual areas compared to higher visual areas, resulting in greater shared variance.

To study variability in spike timing, we measured the histograms of inter-spike intervals (ISI) and their associated coefficients of variation[57]. Coefficient of variation (CV) of each neuron was evaluated as the ratio between the standard deviation and mean of the ISI distributions. Therefore, the farther a neuron's CV deviates from 0, the more irregular the neuron's firing (Fig. 4E, top left). Evaluating CV in a state-specific manner, we found that neurons during the high-frequency state had broader ISI distributions than during other states (Fig. 4E, top right), and accordingly, fired more irregularly in this state (Fig. 4E, bottom). Along the visual hierarchy, spike timing variability decreased irrespective of the internal state (Fig. 4E, bottom, $S_H$: Pearson's r with anatomical hierarchy score = $-0.94$, $p = 0.006$; $S_I$: Pearson's $r = -0.97$, $p = 0.001$; $S_L$: Pearson's $r = -0.94$, $p = 0.006$). Consistent with our expectation that V1 neurons more faithfully represent the features of the time-varying visual stimuli[22,56,58,59], we found that activity of V1 neurons was the most irregular.

In visual system studies, trial-to-trial variability is commonly assessed using the Fano factor (FF)[60], which quantifies the ratio of

variance to mean spike count across trials. An FF of 1 corresponds to a Poisson process, indicating that individual action potentials are generated randomly according to a constant firing rate. To ensure the relevance of our analysis to the visual stimulus, we evaluated FF of neurons with receptive field locations near the screen's center[57,61] (see Methods, Fig. 4F, top). Overall, single neurons in the visual cortex showed greater-than-Poisson variability with FF averaging around $1.21 \pm 047$ (mean ± std). Specifically, spike counts in the low-frequency state showed the largest trial-to-trial variability, suggesting it is less modulated by visual stimuli. In contrast, trial-wise variability was lowest in the high-frequency states (Fig. 4F, bottom). Interestingly, neurons in RL reported the highest variability across visual areas (Fig. 4F, bottom), even regardless of the animal's internal state and stimulus presented (Supplementary Fig. S6C, D). Accordingly, excluding area RL from the analysis revealed a decreasing trend in the trial-to-trial variability along the hierarchy in the high-frequency state ($S_H$: Pearson's r with anatomical hierarchy score = $-0.94$, $p = 0.02$; $S_I$: Pearson's $r = -0.43$, $p = 0.5$, $S_L$: Pearson's $r = -0.46$, $p = 0.4$).

Based on these results, we hypothesized that lower shared variance and trial-to-trial variability in spiking activity during the high-frequency state would improve stimulus encoding (Fig. 4D, F). Meanwhile, the increased spike timing variability during this state could be due to better encoding of the temporal changes in the natural movie video stimulus (Fig. 4E). We directly validated this hypothesis by evaluating the mutual information (MI) between the population spiking activity and the pixel-level information within each frame of the movie in a trial-by-trial manner in each state (Fig. 4G, top). As expected, spiking activity in the high-frequency state was more informative about the stimulus than the lower-frequency state, with V1 neurons encoding most of that information (Fig. 4G, bottom, Supplementary Fig. S6A). In line with the observed high FF measures (Fig. 4F), neurons in RL reported the lowest MI with the stimulus (see Discussion). Again, omitting the low MI measures in RL, pixel-level information decreased along the hierarchy during the high-frequency state ($S_H$: Pearson's r with anatomical hierarchy score = $-0.90$, $p = 0.038$; $S_I$: Pearson's $r = -0.86$, $p = 0.06$, $S_L$: Pearson's $r = -0.81$, $p = 0.09$). While these findings confirmed the association between spiking variability and stimulus representation across states, they further suggest a loss of pixel-level information along the visual pathway.

In summary, the high-frequency state is characterized by lower population shared variance, trial-to-trial variability, and increased

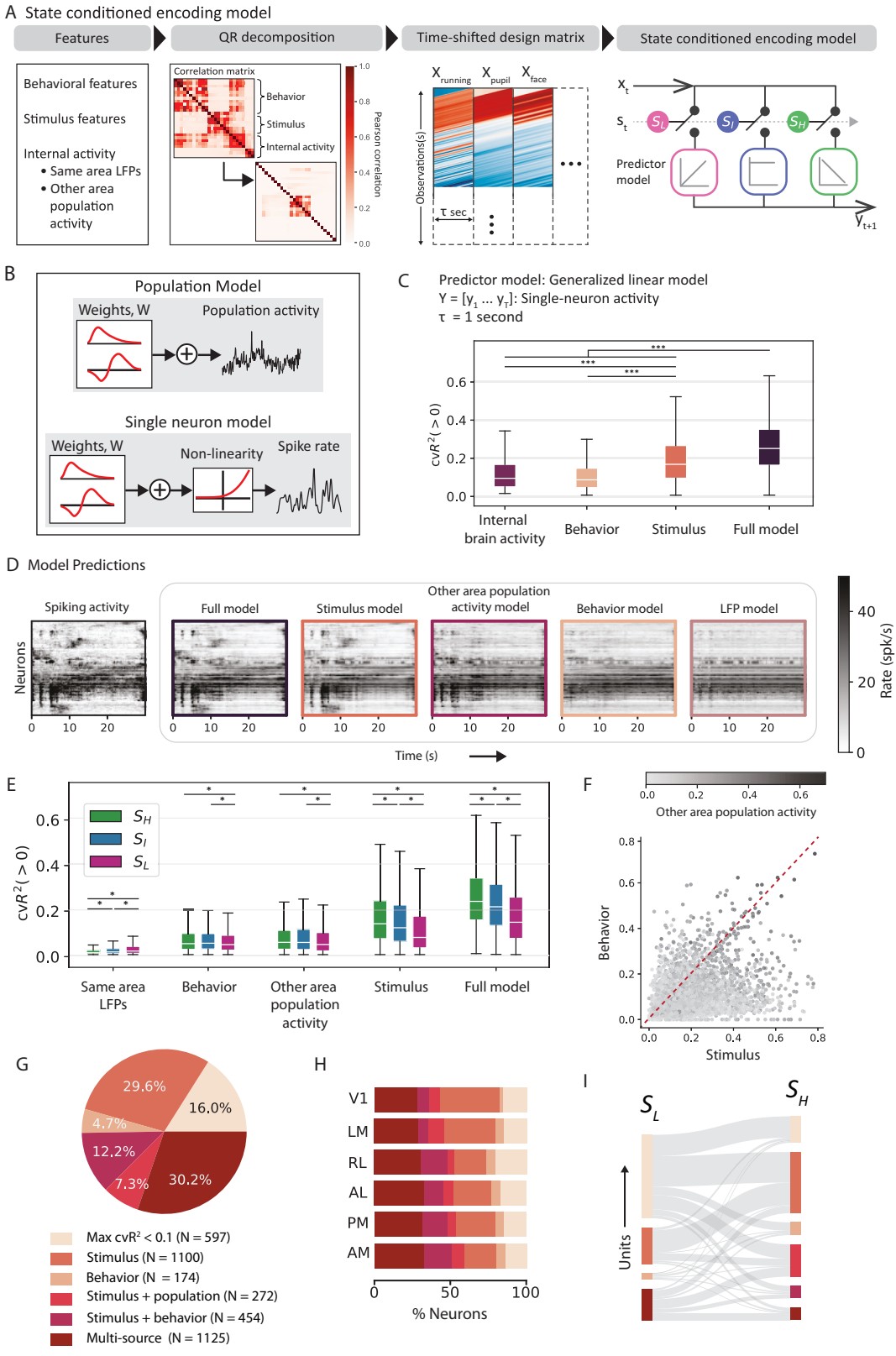

spike timing variability. During this state, variability trends showed strong anti-correlations with the anatomical hierarchy scores such that V1 demonstrated the highest variability across the different visual areas in all three measurements. This could be due to a strong influence of the temporal pattern of sensory drive in early areas, which is validated by the trend of decreasing pixel-level information encoded in V1, especially in the high-frequency state.

**Internal state conditioned neural encoding model**

Given the substantial influence of the internal oscillation states on spiking variability and sensory processing, we next sought to quantify the impact of different variability sources on neural dynamics during the different states. We built an encoding model conditioned on internal states to predict changes in single-trial neural activity in each visual area (Fig. 5A). The resulting framework allows for the

**Fig. 5 | Relative contributions of the different sources to single neuron variability. A** State-conditioned encoding model to account for state-specific contributions of different sources of variability. Design matrices were constructed using decorrelated features to train state-specific regressors. $S_H$: High-frequency state (green), $S_I$: Intermediate state (blue), and $S_L$: Low-frequency state (pink). **B** Regression models to study encoding in population and single neuron models. Population models included a linear weighting of the input features, while in single neuron models, linear weighting was followed by a non-linear exponential projection. **C–G** Results from single-neuron model. **C** Mean explained variance for different categories of input features, averaged across $n = 3923$ neurons and obtained using five-fold cross-validation. **D** (First panel) Neuronal activity, with neurons sorted vertically by a manifold embedding algorithm, Rastermap. (Panels 2–6) Prediction of neuronal activity ($n = 350$ units, best explained units across mice and areas) from respective input feature categories. **E** Contributions from single category models to explaining single-neuron variability during different oscillation states ($n = 3923$ neurons). $S_H$: High-frequency state (green), $S_I$: Intermediate state (blue), and $S_L$: Low-frequency state (pink). **F** Explained variance of all units in each input feature category. **G** Neuronal clusters identified through unsupervised clustering of the final explained variance from single-category models for all units. **H** Distribution of neuronal clusters across areas. **I** Neuronal clusters derived from unsupervised clustering of state-specific explained variance from single-category models for all units, showing how feature encoding dynamics shift across different oscillatory states. Box-plots in (**C**, **E**) show the first and third quartiles, the inner line is the median over all neurons ($n = 3923$), and the whiskers represent $1.5 \times IQR$ (Tukey method). Statistical tests in (**C**, **E**) were adjusted for multiple comparisons using the Bonferroni correction (***$p < 0.0001$, **$p < 0.001$, *$p < 0.05$). Source data are provided as a Source Data file.

quantification of state-specific contributions of stimulus and other source variables to the target single-trial neural activity. Deriving inspiration from an HMM-GLM framework[62], the encoding model has two essential pieces: an HMM governing the distribution over latent LFP states (identified in the preceding section) and a set of state-specific predictors governing the weight distributions over the input features. However, unlike the previously proposed HMM-GLM, the state sequences are pre-determined by the HMM, and we do not re-train the HMM model for optimized prediction. Finally, the model also produces a time-varying kernel ($\tau$ seconds long) for each feature, relating that variable to neural activity in the subsequent time bin (Fig. 5A, panel 3).

Our model considers an extensive array of variables that we classify into three categories: stimulus, behavior, and internal brain activity (Fig. 5A, panel 1). Stimulus features include a set of higher (edges, kurtosis, energy, entropy) and lower-order (intensity, contrast) image features, and behavioral features include the complete set of movement variables determined in the previous section (see Fig. 3). Under internal brain activity, the model includes both the averaged neuronal population activity from simultaneously recorded neighboring visual areas (that is, other than the target visual area) and the raw LFPs from different layers within the target area. The averaged neuronal population activity represents the average activity across all units in a given area. Since model fits to linearly dependent input features are unreliable, we employed QR decomposition to systematically orthogonalize the input features[63] (see Methods).

We derived two separate versions of the encoding model to study neural variability at multiple scales: a population model and a single-neuron model (Fig. 5B). The single-neuron model predicted the single-trial firing rate of the target neuron, while the population model predicted the single-trial averaged neuronal population activity in an area. In the population model, the predictors were linear regressors of the input features, and the model was fit using Ridge regression to prevent overfitting (equation (16)). The single-neuron model accounted for the non-linearity associated with spike generation, wherein the predictors were designed as Poisson regressors of the input features, and the model was optimized by maximizing a regularized log-likelihood function to prevent overfitting (equation (17)). To evaluate how well the model captured the target neural activity, we computed the five-fold cross-validated $R^2$ (cv$R^2$, equation (18)).

Before quantifying state-specific variability from different sources, we assessed the effect of internal states on neuronal variability while controlling for other factors. We ran two separate ANCOVA tests to examine how internal states influenced variability in both single-unit and population-level activity (Supplementary Fig. S7A–D). Internal states significantly affected variability at both levels, with contributions of 3% for single-unit activity and 6% for averaged population activity, indicating that additional factors are required to explain a majority of the observed variability.

## State specific contributions to single-neuron variability

To systematically quantify the relative contributions from the different sources to single-neuron variability in each trial, we constructed a state-conditioned GLM framework. Since a GLM predicts the conditional intensity of the spiking response, we evaluated our model performance against the rate functions of individual neurons obtained after smoothing the spike counts with a Gaussian filter (s.d. 50 ms). To appropriately identify their variability sources, neurons were further selected based on minimal firing rate (>1 spikes/s in all states) criteria and receptive field locations, along with the standard quality control metrics of the dataset[22] (see Methods). After filtering, $n = 3923$ units remained across all mice and were analyzed using the GLM model.

The model explained an average of cv$R_F^2 = 26.7 \pm 13.5\%$ (mean ± std, $n = 3923$ units) of the total variance of single-trial activity across all neurons (Fig. 5C). To quantify the relative contributions of different source variables, we applied the model to individual sub-groups corresponding to each category. Across all factors, stimulus features were the most predictive of single-neuron activity (cv$R_S^2 = 19.8 \pm 13.6\%$, mean ± std), while internal brain activity was the least predictive (cv$R_I^2 = 11.6 \pm 9.5\%$, mean ± std). Within the internal brain activity category, averaged population activity from neighboring visual areas accounted for nearly twice as much explained variance as LFPs from the same area (cv$R_{LFP}^2 = 5.6 \pm 6.4\%$, cv$R_P^2 = 11.1 \pm 9.4\%$, mean ± std, $p = 3.2e^{-179}$, one-way ANOVA). Therefore, these features were analyzed separately. Across different visual areas, single neuron variability was best explained along the anterolateral pathway (LM, AM, and AL, cv$R_F^2 = 26.2 \pm 0.9\%$ (mean ± std), Supplementary Fig. S8, $p = 2.5e-05$, one-way ANOVA). Visualizing the model predictions using Rastermap[18] revealed transient changes in the neural ensemble that were captured solely by the stimulus features (Fig. 5D). Other features were less discerning and captured only the broad changes in the firing patterns. Additionally, we examined cell-type specific contributions from various factors. We found that the explained variance of fast spiking units (FS) significantly surpassed that of regular spiking units (RS) (Supplementary Fig. S8H, $p = 5.32e-11$, one-way ANOVA), with behavior and internal activity contributing more to FS units. In contrast, stimulus features explained variance equally well in both FS and RS cells (Supplementary Fig. S8F–K, behavior: $p = 9e-48$, internal activity: $p = 0$, stimulus: $p = 0.05$, one-way ANOVA).

We further investigated how internal brain states influence neuronal encoding by analyzing state-wise contributions of input features to single-neuron activity. Neuronal activity was the most predictive during the high-frequency state (Fig. 5E, $p = 5e-109$, one-way ANOVA) such that stimulus and behavior-driven variability was highest in this state (stimulus: $p = 2.1e-70$; behavior, $p = 0.008$, one-way ANOVA), and lowest in the low-frequency state. Within internal activity features, population activity from neighboring visual areas contributed more significantly during the high-frequency state ($p = 0.02$, one-way ANOVA), while LFPs from the same area played a more prominent

role in the low-frequency state ($p = 3.9\text{e-}28$, one-way ANOVA). These findings are consistent with prior studies[11,64,65], highlighting the role of slow-oscillatory waves in synchronizing spiking activity during the low-frequency state (Fig. 4C, D), thereby disrupting stimulus encoding in this state.

The influence of different sources was not uniform across neurons; rather, individual neurons appeared to be driven by a diverse array of factors (Fig. 5F), suggesting heterogeneous coding mechanisms within the population. To investigate single-cell diversity in the visual cortex, we used unsupervised clustering based on each neuron's encoding pattern, represented as a 5-element vector that included (cross-validated) explained variance from each feature category (stimulus, behavior, same-area LFPs, and population activity from other visual areas) and the number of categories with >10% explanatory power (Supplementary Fig. S9A). Clustering based on these encoding profiles revealed six distinct groups (Fig. 5G, Supplementary Fig. S9B): one dominated by stimulus, one by behavior, another with high encoding of both stimulus and behavior, a group influenced by both stimulus and averaged population activity from neighboring visual areas, a group with high explained variance across all input feature categories (multi-source), and a final group comprising neurons where no single feature explained >10% of their variance. The two largest clusters comprised units predominantly driven by stimulus features alone (29.5%) and multi-source units (30.2%). These two clusters made up 60% of all units (Fig. 5G). In contrast, units driven solely by behavior formed the smallest cluster, representing just 4.7% of all units, while 12.2% of units were jointly influenced by both stimulus and behavior features.

When examining the distribution of all neuron clusters across visual areas, we found that the fraction of units best predicted by stimulus features peaked in V1 (39.6% of units in V1), decreasing along the hierarchy (Fig. 5H; LM: 33.8%, RL: 21%, AL: 24.8%, PM: 26%, AM: 21.3%; Pearson correlation with hierarchy score, $r_{p-RL} = -0.96$, $p_{p-RL} = 0.01$). Conversely, the influence of behavior increased along the hierarchy. Proportion of units driven by behavior alone (V1: 2.1%, LM: 5.3%, RL: 5.6%, AL: 5.8%, PM: 5.4%, AM: 5.9%; $r_p = 0.84$, $p_p = 0.03$) and of units affected by both stimulus and behavior nearly doubled in higher-order areas (V1: 7.7%, LM: 6.5%, RL: 17.9%, AL: 12.7%, PM: 16.7%, AM: 18.6%). Lastly, the proportion of multi-source units also increased along the hierarchy (V1: 28%, LM: 28.8%, RL: 30%, AL: 32.6%, PM: 31.4%, AM: 32.4%; $r_p = 0.85$, $p_p = 0.03$). These findings point to an increasing functional diversity among neurons as one ascends the visual hierarchy. Supporting this, neurons influenced by multiple factors, especially behavior, had larger receptive field sizes (Supplementary Fig. S9F, $p = 0.0001$, one-way ANOVA), consistent with the known trend of increasing receptive field sizes along the hierarchy[66,67]. Multi-source units also tended to have higher firing rates compared to neurons predominantly explained by a single factor (Supplementary Fig. S9D, $p = 4.3 \times 10^{-71}$, one-way ANOVA). Finally, the ratio of RS to FS units was highest in the stimulus-driven cluster (8:1), exceeding the overall ratio of 4:1 in the neuronal population (Supplementary Fig. S9E). In contrast, clusters driven by behavior or influenced by multiple factors had a lower ratio (2:1), indicating that non-visual factors predominantly modulate FS units, while visual factors primarily modulate RS units.

Given the shift in neural dynamics across different brain states, we explored how the contributing factors to single-unit activity varied between these states. To do so, we performed a similar clustering analysis based on the state-specific explained variance of individual factors in both high- and low-frequency states (Fig. 5I, Supplementary Fig. S9H–K). This analysis revealed notable changes in the dominant factors contributing to single unit variance across states. In the low-frequency state, a large proportion of units (53%) fell into a cluster where no single feature explained >10% of their variance. However, in the high-frequency state, around 40%

of these same units shifted to being predominantly driven by stimulus features alone. Additionally, units influenced by multiple sources in the low-frequency state became more specialized in the high-frequency state. These results indicate a significant state-dependent reorganization of neural representation in the visual cortex.

## State specific contributions to population-level variability

Recent studies[17,18] have reported significant contribution of spontaneous movements in the emergent properties of brain-wide activity. To examine these effects in population dynamics within the visual cortex in a state-specific manner, we constructed a state-dependent linear regression model to predict the averaged neuronal population activity in each of the six visual areas. Using the same input features as the single-neuron model, this population model explained $53.4 \pm 6.6\%$ (mean ± std, $n = 25$ mice, Fig. 6A) of the variance in the averaged neuronal population activity across across the visual areas.

To further explore the contributions of different source variables, we applied the model to individual sub-groups corresponding to each input feature category. Interestingly, internal brain activity had the most predictive power ($\text{cv}R_I^2 = 41.0 \pm 7.6\%$, mean ± std, p = 2.5e-11, one-way ANOVA, $n = 25$ mice), higher even than the combined power of behavioral and stimulus features ($\text{cv}R_{B+S}^2 = 30.1 \pm 9.3\%$, mean ± std, $p = 0.0005$, one-way ANOVA, $n = 25$ mice). This was unlike single neuron activity, which was primarily driven by stimulus features. Stimulus features predicted the variance in the averaged neuronal population activity better than behavioral features ($\text{cv}R_S^2 = 22.8 \pm 8.8\%$, $\text{cv}R_B^2 = 18.9 \pm 7.0\%$, mean ± std, $p = 0.009$, one-way ANOVA, $n = 25$ mice). Although bodily movements significantly contributed to both single-neuron and population-level activity, they were rarely the dominant factor driving visual cortical responses to natural movie stimuli. These successive improvements in the explanatory power resulting from the inclusion of more sources are evident in the prediction traces shown in Fig. 6B. It is worth noting that if single-neuron responses to external stimuli were completely independent, the contribution from stimulus features to population activity would be negligible. Nevertheless, the significant influence of stimulus features on population-level variability is suggestive of stimulus-related neuronal correlations within an area.

The addition of internal brain activity to the combined model of behavioral and stimulus features increased the explained variance by almost 24% ($\Delta r_{F-(B+S)}^2 = 23.5 \pm 10.2\%$, mean ± std, Fig. 6A). Considering that LFP and population activity inherently carry information about stimulus and behavioral features, potentially making part of their contributions redundant, we have deliberately orthogonalized these internal variables against the stimulus and behavior variables[68]. This orthogonalization ensures that internal variables capture variance beyond what can be accounted for by stimulus and behavior variables alone.

To understand the substantial increase in explained variance, we analyzed the contributions of internal brain activity to each state. We found that these variables largely increased the predictability during the low-frequency state ($\Delta r_{S_L, F-(B+S)}^2 = 39.0 \pm 15.8\%$, mean ± std, Fig. 6C, left panel). Activity in this state was poorly explained by the combined model of stimulus and behavioral features ($\text{cv}R_{S_L,(B+S)}^2 = 16.2 \pm 11.5\%$, mean ± std, $p = 8.3\text{e-}6$, one-way ANOVA, $n = 25$ mice). The combined model of stimulus and behavioral features was best at explaining variability in the high-frequency state, and accordingly, activity in this state showed a smaller improvement in its predictability on the inclusion of internal activity features ($\Delta r_{S_H, F-(B+S)}^2 = 14.3 \pm 4.6\%$, mean ± std, $p = 2.3\text{e-}6$, one-way ANOVA, $n = 25$ mice). Consistently, within-area LFPs and averaged population activity from the neighboring visual areas contributed more towards explaining the activity in the low-frequency state ($p = 4.5\text{e-}6$, $p = 8.2\text{e-}13$, respectively; one-way ANOVA, $n = 25$ mice, Fig. 6D). At the same time, both stimulus and

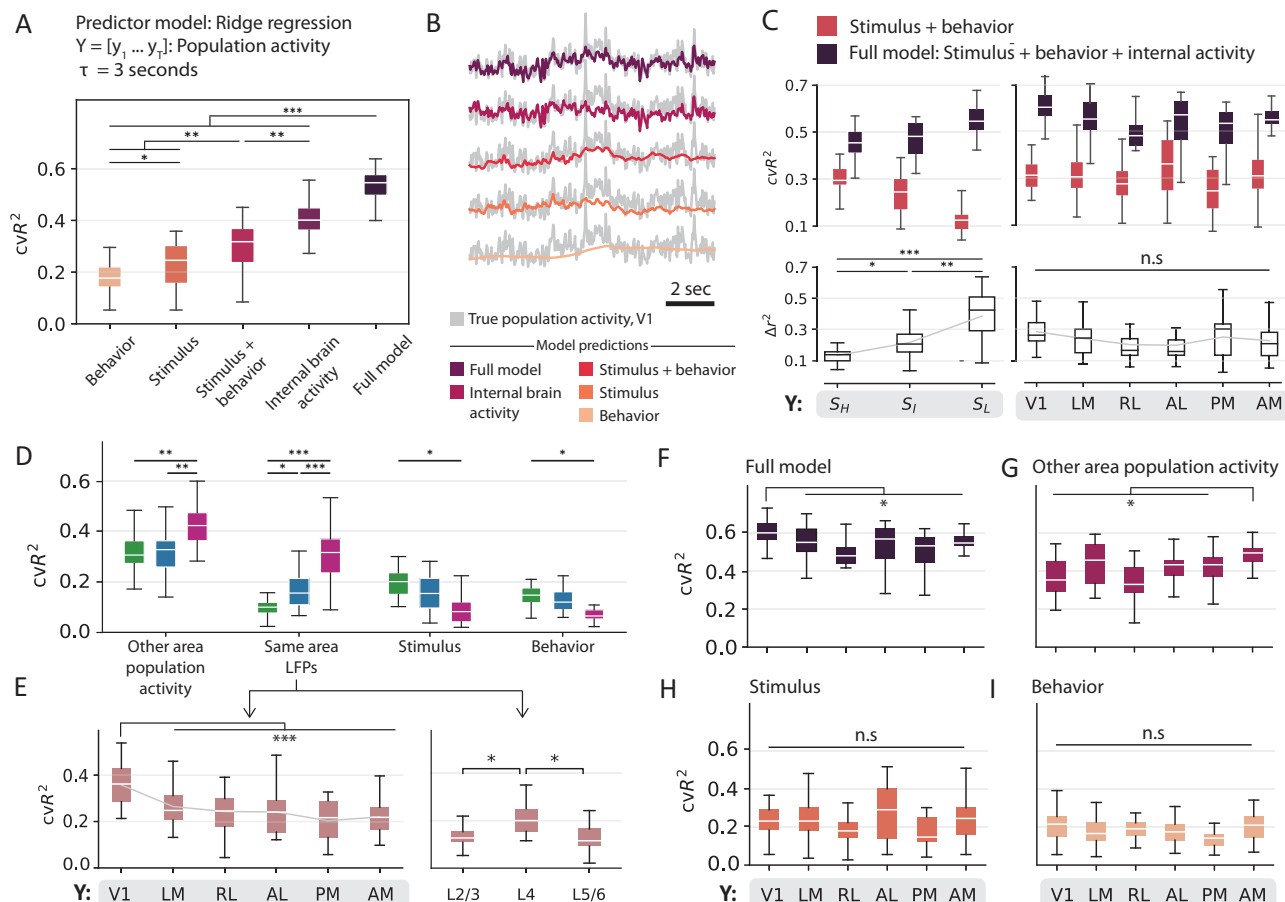

**Fig. 6 | Relative contributions of the different sources to population-level variability. A** Explained variance for different categories of input feature groups, averaged across all mice obtained using five-fold cross-validation. **B** Averaged population responses overlaid with model predictions from respective input feature groups. The prediction traces were generated by concatenating the outputs from three state-conditioned GLMs into a single, continuous prediction. **C** Comparison of predictions in different (left) states and (right) visual areas prior to and post addition of internal brain activity. Top: Cross-validated explained variance for each model. Bottom: Unique contribution of internal brain activity. **D** Contributions from single category models to explaining the variance in

averaged neuronal population activity in different states. **E** Contributions from LFPs in the same area to explain the variance in averaged neuronal population activity from (right) different layers and (left) in different visual areas. **F–I** Same as (**E**) (left), but for different input features, one-way ANOVA (**F**, $p = 5.8e\text{-}5$; (**G**) $p = 6e\text{-}3$; (**H, I**) $p > 0.01$). In all panels, boxplots show the first and third quartiles, the inner line is the median over 25 mice, and the whiskers represent $1.5 \times \text{IQR}$ (Tukey method). All statistical tests were adjusted for multiple comparisons using the Bonferroni correction (***$p < 0.0001$, **$p < 0.001$, *$p < 0.05$). Source data are provided as a Source Data file.

behavioral features demonstrated increased predictive power during the high-frequency state ($p = 0.003$, $p = 0.01$, respectively; one-way ANOVA, $n = 25$ mice).

Using the complete set of input features, we could predict about $61.1 \pm 6.9\%$ (mean ± std, $n = 25$ mice) of the variance in V1's averaged neuronal population activity, the highest among all visual areas (Fig. 6F). Although including internal brain activity did not differentially affect predictability across visual areas ($p = 0.12$, one-way ANOVA, $n = 25$ mice, Fig. 6C, right panel), contributions from its subcomponents revealed interesting differences. Firstly, averaged population activity from neighboring areas explained more variance than within-area LFPs ($p = 1.5e\text{-}9$, one-way ANOVA, $n = 25$ mice, Fig. 6D). Secondly, their across-area prediction showed reversed trends. While LFPs explained significantly more variance in V1 than other visual areas ($p = 1.9e\text{-}8$, one-way ANOVA, Fig. 6E, left panel), averaged population activity explained significantly more variance in AM ($p = 6e\text{-}3$, one-way ANOVA, Fig. 6G). Lastly, the predictive power of LFPs varied across the cortical depth, wherein layer 4 (L4) LFPs contributed more to the variance in the averaged neuronal population activity than LFPs in other layers ($p_{L_{2/3}, L_4} = 5e\text{-}4$, $p_{L_{5/6}, L_4} = 2.5e\text{-}4$, one-way ANOVA, Fig. 6E, right panel).

When disregarding the influence of internal states, stimulus features did not significantly differ in their predictive power across areas (Fig. 6H, $p = 0.13$, one-way ANOVA), even at the level of single features (Supplementary Fig. S10A, B, $p \in [0.33, 1]$). However, state-specific analysis revealed pronounced differences in the high-frequency state (Supplementary Fig. S10D, F). In this state, different stimulus features also showed distinct predictive powers indicating heightened sensitivity to stimulus changes (Supplementary Fig. S10C, E). Specifically, higher-order stimulus features (edges, kurtosis, and energy) reported greater predictive power than stimulus contrast and intensity. Finally, facial movements made the most substantial contribution to the averaged neuronal population activity compared to other behavioral features (Supplementary Fig. S11A, C, $p = 0.02$, one-way ANOVA, $n = 25$ mice).

## Discussion

Our observations provide a comprehensive description of the dynamic aspects of spiking variability in the visual cortex as the brain traverses through distinct oscillation states. We characterized this variability along three dimensions: variability across trials[61], variability in spike times[57], and shared variance within a population[55]. By utilizing cortical

LFPs to define different internal oscillation states, we found that each state captured a distinct profile of spiking variability. Using the state fluctuations as a temporal backbone, we constructed a state-based encoding model to partition and evaluate the relative contributions from three different sources of variability to visual cortical activity: visual stimulus, behavior, and internal brain dynamics. The model accounted for 27% of single-neuron variability and 53% of the variance in averaged population activity. Neurons in the visual cortex are influenced by a diverse array of factors, with the relative contributions of these factors differing across sub-populations. Firstly, the combination of factors affecting variability changes spontaneously and rapidly over time in a state-dependent manner. Secondly, the contributions of each source are further influenced by cell type and anatomical location, becoming increasingly heterogeneous as one ascends the hierarchy. Lastly, while single neurons in the visual cortex are primarily affected by stimulus features, population activity is largely dominated by internal brain activity. Overall, our study underscores the importance of accounting for the constantly changing contributions from internal and external factors on stimulus representation at the level of individual units, enabling a deeper understanding of how neural responses are dynamically shaped in real time.

Identifying and locating the different sources influencing neural variability poses a significant challenge in systems neuroscience[10,69]. Previous research has emphasized the significance of internal brain activity in accounting for neuronal variability[2,70,71]. While these studies did not consider variability induced by externally observable task- and behavior-related variables, recent investigations have predominantly focused on this latter category of input features[12,17,18,72]. In this study, we adopt a comprehensive approach by integrating contributions from both internal brain activity and externally observable variables to understand neuronal variability.

We considered a two-fold contribution from internal brain activity. Firstly, utilizing brain states defined by internal oscillatory rhythms as a temporal framework, we were able to associate the various dynamics of spiking variability with these internal states. Secondly, we incorporated averaged neuronal population activity from each neighboring area and LFPs as input features into the state-based encoding model. These variables played a significant role in explaining neural variability, primarily contributing to activity in the low-frequency state. Consistent with previous findings[70,71], internal variables explained ~40% of the total variability of averaged neuronal population activity within an area, even surpassing the variance explained by the combined model of stimulus and behavioral features by 11% ($cvR_I^2 - cvR_{B+S}^2$). At the level of single neurons, contributions from internal brain activity, although relatively small, remained statistically significant, explaining around 11% of the total variance. However, this was nearly 9% ($cvR_S^2 - cvR_I^2$) less than the variance explained by stimulus features alone.

Recent progress in behavioral video analysis, computational modeling, and large-scale recording techniques has highlighted the impact of movement-related variables on neural activity across the cortex[17,18,72]. Our observations are consistent with these findings. Behavior-related variables explained up to ~20% of the averaged neuronal population activity and ~12% of single-neuron variability in the visual cortex. Moreover, the influence of behavior becomes more pronounced in the high-frequency state (Figs. 5E, 6D) and as one ascends the visual hierarchy, entraining a larger proportion of the neural population (Fig. 5H). However, our findings diverge from those reported in Musall et al.[17], which found that uninstructed movements exerted a greater influence on V1 neural activity than a visual stimulus. We attribute this difference to three reasons: first, our mice are passively viewing the screen without engaging in a behavioral task; second, our naturalistic movie stimulus may engage a broader array of neurons compared to the static, flashed stimuli used in previous research; third, our recording captures single-unit spiking activity,

contrasting with previous wide-field calcium imaging. In addition to behavior, these differences underscore the importance of recording methodologies, experimental conditions and stimuli, prompting a closer examination of the specific factors influencing single-trial neural activity in diverse contexts.

Despite large variability in spiking activity, neuronal populations exhibit a remarkable ability to robustly encode information across different brain regions[24,25,73]. Our results suggest this is state-dependent. A clear pattern emerges throughout our analyses: population dynamics during the high-frequency state are the most effective in representing stimulus information, while stimulus features weakly modulate activity in other states (Figs. 4G, 5E, 6D). While several lines of studies have indirectly confirmed this state-dependence of information encoding either through reports of task performance or via investigations under artificially induced states of anesthesia[5,40,70,74], our findings directly quantify and describe this dependency. Specifically, we find that spiking activity in the high-frequency state has the lowest shared variance, lowest trial-to-trial variability, and the highest spike timing variability (Fig. 4). These characteristics of single-neuron activity may result from enhanced encoding of various temporal and spatial features of the time-varying natural movie stimulus during the high-arousal state (Figs. 5E, 6D). In contrast, the dominance of slow oscillatory activity in low-frequency state, coupled with high shared variance, trial-to-trial variability, and more regular firing, appears to reflect internal dynamics that disrupt the accurate representation of stimulus information. We posit that this observed correlation between heightened sensory encoding capacity and increased arousal during the high-frequency state may arise from the mice's innate survival mechanism, leading them to enhance visual information intake while in a state of heightened alertness or running.

Neurons in the visual cortex can be classified by several criteria, including their morphology, connectivity, developmental history, gene expression, intrinsic physiology, and in vivo encoding strategies. Single-cell RNA sequencing studies have revealed extensive cell-type diversity and their relationships within cortical circuits[75,76]. Different cell types have distinct functional roles, which are further influenced by their position within the cortical hierarchy and the specific inputs they receive across different layers[77–82]. Furthermore, different neuronal types are modulated by various factors such as behavior[17,18], top-down feedback[78], and internal brain states[83]. Through unsupervised clustering of each neuron's encoding patterns, we quantified their encoding diversity, uncovering units with specialized properties within the visual cortex (Fig. 5G). Our findings indicate that neurons in the visual cortex are modulated by a diverse array of factors, with the relative contributions of these factors varying across states, hierarchical positions, layers and cell-types (Fig. 5H, I, Supplementary Fig. S9C, E). Additionally, we observed an increasing representation of pan-modulated units along the visual hierarchy (Fig. 5H), suggesting that while integrative processes may start as early as V1, a larger network of neurons becomes involved in this process higher up the hierarchy. In line with recent studies suggesting that sensory-motor integration begins in early sensory areas[18,46,84], we identified two distinct neuronal clusters likely involved in this process: one driven solely by behavior, and another influenced by both visual stimuli and behavioral factors (Fig. 5G). These findings emphasize the complex and dynamic nature of visual processing, shaped by a multitude of internal and external factors.

Given the hierarchical organization of the visual cortex[22,25], the response variance of a sensory neuron can potentially limit the amount of stimulus information available to downstream circuits[51] (Fig. 4G). While past studies have shown the effects of pair-wise correlations on information encoded by a neuronal population[49,50,53], a more comprehensive population-level perspective is essential to understanding the brain's correlational structure[12,85,86]. Here, we applied shared variance[55] as a generalization of the pair-wise correlations between

single neurons extended to an entire population. Notably, we observed a decrease in the percent of shared variance along the visual hierarchy (Fig. 4D). While this decline might imply the introduction of independent noise at subsequent stages of signal processing, it could alternatively result from the increased diversity of neurons influenced by factors other than the stimulus itself (Fig. 5H). The high variance shared across neurons in V1 can likely be attributed to V1 comprising the largest proportion of neurons exhibiting strong, time-locked responses to the temporal dynamics of stimulus features (Fig. 5H[4,56]). Our findings provide further support for this notion, particularly through the observation that neurons in V1 reported high spike-timing variability, likely corresponding to the variance induced by a constantly changing stimulus (Fig. 4E). Consistently, LFPs have a more pronounced influence on averaged population activity in V1 in comparison to other visual areas (Fig. 6E). This suggests that the collective synaptic inputs into V1, represented by LFPs in the area, may entrain a larger population in V1 than in other areas.

Previous studies have indicated that trial-to-trial variability (Fano factor) increases as information propagates up along the visual pathway from the retinal receptors to the primary visual cortex[61,70,87]. Our observations mirror this trend in the visual cortex when mice were exposed to full-field light flashes, revealing an increase in trial-to-trial variability along the cortical hierarchy (Supplementary Fig. S6D). However, in response to natural movies, trial-to-trial variability decreased along the visual cortical hierarchy (Fig. 4F). We attribute this decrease in variability to the heterogeneous properties of a natural movie frame where, in awake mice, eye movements (even small saccades) across the frame could elicit more variable neuronal responses across trials in early visual areas with smaller receptive fields[88]. Consistent with this, controlling for fluctuations in eye gaze in the evaluation of Fano factor abolished the area-wise differences in trial-to-trial variability across all states (see Methods, Supplementary Fig. S6E–G). Lastly, it is important to note the variability properties of neurons in the rostrolateral visual area (RL), which do not always follow the visual hierarchy trends. This is especially true when considering trends related to stimulus encoding, such as trial-to-trial variability and mutual information (Figs. 4F, G, 5H, Supplementary Fig. S6C, D). We attribute this to two reasons. Firstly, since RL is located at the border of the visual and primary somatosensory (S1) cortices, the functional specialization of neurons in RL is likely more diverse than in other visual areas. This is reflected in our findings where RL had the smallest proportion of neurons influenced by stimulus features and the largest proportion of neurons with low explained variance (Fig. 5H). Secondly, due to the retinotopic center of RL being situated on the boundary between RL and S1[89], it is often challenging to target its precise retinotopic center[90].

The dynamic nature of neuronal variance across time has been consistently demonstrated in theoretical and empirical analyses[3,10,91]. Here, we specifically quantify the magnitude of stimulus-driven neuronal variability associated with internal states. Our findings show that, during passive viewing, mice typically persist in a specific state for an average duration of $1.5 \pm 0.1$ seconds, indicating that state-dependent neuronal variability undergoes changes within seconds (Figs. 2F). The state sequences reveal a smooth transition of neuronal variability between distinct variability profiles, passing through an intermediate state (Figs. 2G, 4). Moreover, each state constitutes a unique composition of sources that influence neuronal variability (Figs. 5E, I, 6D). These rapid shifts in source composition across states arise from the complex interactions between non-stationary source variables, collectively contributing to the dynamics in neuronal variability.

These findings offer additional insights into the dynamic properties of neuronal variability, providing important constraints for theoretical modeling of stimulus-driven variability. Firstly, the dynamically changing source composition indicates that the responsiveness of a neuronal population to sensory input varies over time,

challenging the assumption of a constant stimulus contributing to the responsiveness of a sensory system. Secondly, accounting for the distinct variability profiles associated with different internal states can specifically address the non-stationary stimulus-encoding capability of neuronal populations. Lastly, integrating state fluctuations as a temporal framework can enhance our understanding of the network dynamics contributing to neuronal variability.

Several studies have demonstrated that the structural connectivity of neural networks directly influences neural dynamics[92]. Theoretical studies on biologically plausible models show that neural computations are guided by the interplay between recurrence and changes in dimensionality, enabling flexible computations across different tasks[93,94]. This flexibility in local circuits has been shown to be closely linked to the emergence of metastable states, where neural activity remains in quasi-stable patterns before abruptly transitioning to new states[95]. These metastable states have provided a valuable framework for studying how structural connectivity relates to variability and noise correlations, which in turn influences the state-dependent processing of sensory information[96–100]. While these studies have suggested important neural mechanisms, they are yet to be tested in the scenario of state-dependent changes in local circuit dynamics, which could represent a more global mechanism underlying the dynamic variability observed across various brain areas. To address this gap, our future work will investigate the state-specific functional organization of the cortical circuits and how they adapt to different internal and external stimuli, with a particular focus on the mechanisms that drive state transitions and their impact on sensory processing.

In this study, we make use of the controlled yet dynamic structure of the passive viewing design to trace neuronal variability across discrete oscillation states in awake mice. While our discrete characterization of brain states provides a straightforward interpretation of neural activity, recognizing the possibility of continuous state changes (such as a continuum of pupil size or network activity changes) is vital for exploring the full spectrum of neural responses in awake, behaving animals. Additionally, to fully characterize neuronal variability and its influence on information processing in the cortex, investigating neural activity during active tasks is essential. Recent studies have shown that a subject's engagement during an active task varies drastically from trial to trial, playing out through multiple interleaved strategies[62,101,102], where other work has shown that changes in these strategies can be predicted by the animal's arousal levels, suggesting a direct link between brain states and task performance[11,103]. While the tools in this study can help identify variables that promote task engagement, they do not elucidate the underlying mechanisms causing state transitions. Understanding these dynamics entails a thorough investigation of unit activity in the subcortical regions of the brain.

Our observations, combined with existing studies on spiking variability, suggest that cortical state acts as a key determinant of the variability seen in the cortex. By offering a comprehensive view of this variability, we have been able to directly study both the sensory and non-sensory aspects of neuronal responses in the visual cortex. It is evident that spiking variability in the cortex transcends mere 'neural noise', and explaining neuronal variability by partitioning it into different origins can help us understand its influence on information representation and propagation in the brain, and ultimately resolve its computational contribution to behavior.

## Methods
### Data collection
The data analyzed and discussed in this paper are part of the publicly released Allen Institute Brain Observatory Neuropixels dataset ($n = 25$ mice)[22]. Neural recordings used Neuropixels probes[104] comprising 960 recording sites, of which either 374 for "Neuropixels 3a" or 383 for "Neuropixels 1.0" were configured for recording. The electrode sites

closest to the tip formed a checkerboard pattern on a 70 µm wide x 10 mm long shank. Six Neuropixels probes were inserted at the shallowest 2 mm and at the deepest 3.5 mm into the brain for each recording. These requirements ensured adequate recordings of the cortex while preventing any brain damage. To ensure that the probes were recording from functionally related cells in each visual area, retinotopic centers were determined and targeted accordingly. Targeting the cortical visual areas, AM, PM, V1, LM, AL, and RL, was guided by the angle of approach of the probe, as well as the depth of functionality of the imaging boundaries. All procedures were performed according to protocols approved by the Allen Institute Institutional Animal Care and Use Committee under an assurance with the NIH Office of Laboratory Animal Welfare.

The Open Ephys GUI was used to collect all electrophysiological data. Signals from each recording site were split into a spike band (30 kHz sampling rate, 500 Hz highpass filter) and an LFP band (2.5 kHz sampling rate, 1000 Hz lowpass filter). Spike sorting followed the methods outlined in Siegle, Jia, et al.[22]. Briefly, the spike-band data was subject to DC offset removal, median subtraction, filtering, and whitening before applying the Kilosort2 MATLAB package (https://github.com/MouseLand/Kilosort) for spike time identification and unit assignment[18]. Detailed information about the complete experimental design can be found in Durand et al.[105].

## Statistics and data analyses
For all analyses, Python was used as the primary programming language. Essential analytical tools utilized include Scipy[106] and Scikit-learn[107]. Error bars, unless otherwise specified, were determined as the standard error of the mean. For comparisons across units ($n = 7609$ units after QC filtering, and $n = 3923$ units post-RF filtering), mice ($n = 25$), or states ($n = 3$), we used a one-way ANOVA for Gaussian-distributed metrics and the rank sum test for non-Gaussian distributed metrics. In cases of high subject-to-subject variability, we used a paired T-test. Bonferroni correction was applied for multi-group comparisons. To evaluate the similarity to the previously established anatomical visual hierarchy in mice[25], we computed the correlation between our measured variable and the anatomical hierarchy score (V1: −0.50, RL: −0.14, LM: −0.13, AL: 0.00, PM: 0.12, AM: 0.29), and Pearson's correlation was applied to estimate the significance of correlation.

## Visual stimulus
Custom scripts based on PsychoPy (Peirce, 2007) were used to create visual stimuli, which were then presented on an ASUS PA248Q LCD monitor. The monitor had a resolution of 1920 x 1200 pixels and a refresh rate of 60 Hz, measuring 21.93 inches wide. The stimuli were shown monocularly, with the monitor positioned 15 cm from the right eye of the mouse. The visual space covered by the stimuli was 120° × 95° before any distortion occurred. Each monitor used in the experiment was gamma corrected and maintained a mean luminance of 50 cd/m². To accommodate the mouse's close viewing angle, spherical warping was applied to all stimuli to ensure consistent apparent size, speed, and spatial frequency across the monitor from the mouse's perspective.

**Receptive field mapping.** The receptive field locations were mapped with small Gabor patches randomly flashed at one of 81 locations across the screen. Every degree of drifting grating (3 directions: 0°, 45°, 90°) was characterized by a 2 Hz, 0.04 cycles with a 20° circular mask. The receptive field map (RF) for an individual unit is defined as the average 2D histogram of spike counts at each of the 81 locations, where each pixel corresponds to a 10° × 10° square.

**Stimuli for passive viewing.** The mice were exposed to various types of stimuli during the experiment, including drifting gratings, natural movies, and a flashes stimulus. The gratings stimulus included 4

directional gratings that were repeated 75 times at a frequency of 2 Hz. As for the natural movies, they were divided into 30 s clips, and each clip was repeated 30 times as a block. To introduce variability, there were an additional 20 repeats with temporal shuffling. Lastly, the flashes stimulus included a series of dark or light full field image with luminance = 100 cd/m².

## Quality control metrics
All single-neuron analyses (Figs. 4, 5) were performed on neurons that successfully met three essential quality control thresholds: presence ratio (>0.9), inter-spike interval violations (< 0.5) and amplitude cut-off (<0.1). Specific details of these metrics can be found in Siegle, Jia, et al.[22]. These metrics were implemented to prevent the inclusion of neurons with noisy data in the reported analyses, considering both the physical characteristics of the units' waveforms and potential spike sorting challenges. For single-neurons analyzed in Fig. 5, a tighter threshold on presence ratio (>0.95) was incorporated to avoid inflated values of prediction accuracy. Additionally, analyses in Figs. 4F and 5 were filtered for neurons with receptive fields positioned at least 20 degrees away from the monitor's edge. This criterion was incorporated to facilitate a meaningful comparison of the relative contributions from different sources of variability.

## Local field potentials and time-frequency analysis
Prior to constructing the hidden Markov model (HMM), we identified appropriate frequency ranges in the LFPs. To evaluate their power spectra, we applied short time-Fourier transform (STFT) on single channels using a Hann window of size ~800 ms such that consecutive windows overlapped over ~400 ms. Z-scoring the power spectrum at each frequency revealed LFP modulations in distinct frequency bands (Fig. 2B). Further informed by the literature on LFPs in the mouse cortex[33,34,108–110], the following frequency ranges were selected from the LFP spectrum: 3–8 Hz (theta), 10–30 Hz (beta), 30–50 Hz (low gamma), and 50–80 Hz (high gamma). To filter the LFPs, we constructed four IIR Butterworth filters of order 11, each corresponding to the above frequency ranges. Finally, envelopes of the filtered LFP signals, obtained via the Hilbert transform, were supplied as observations to the HMM.

As part of these observations provided to the HMM model, we included LFPs recorded across different cortical depths. To determine the corresponding layer of each LFP channel, we first estimated the depth of the middle layer of the cortical column. Similar to methods summarized previously[24,111], we applied current source density (CSD) on the LFPs within the 250 ms interval post-presentation of the flashing stimulus. To evaluate the CSD, we calculated each recording site's average evoked (stimulus-locked) LFP response ($s$) and duplicated the uppermost and lowermost LFP traces. Next, we smoothed the signals across sites as shown in equation (1), where $r$ is the coordinate perpendicular to the layers, and $h$ is the spatial sampling distance along the electrode. Finally, the CSD mapping was obtained as the second spatial derivative of the LFP response (equation (2), Supplementary Fig. S1D, right). The CSD map can approximately dissociate the current sinks from current sources, respectively indicated as downward and upward deflections in the density map.

$$\bar{s}(r) = \frac{1}{4}(s(r+h) + 2s(r) + s(r-h)) \quad (1)$$

$$D = \frac{1}{h^2}(\bar{s}(r+h) - 2\bar{s}(r) + \bar{s}(r-h)). \quad (2)$$

To facilitate visualization, we used 2D Gaussian kernels ($\sigma_x = 1$, $\sigma_y = 2$) to smooth the CSD maps. We identified the location of the input layer based on the first appearance of a sink within 100 ms of the stimulus onset. We then designated the center channel of the middle layer (L4) as the input layer and marked eight channels above

and below it as L4. All channels above the middle layer were classified as superficial layers (L2/3), while all channels below the middle layer but above the white matter were categorized as deep layers (L5/6). Lastly, for each mouse, we validated the layer classification against the spectral decomposition of the LFPs across depth (Supplementary Fig. S1D).

## Identification of internal oscillation states−hidden Markov model

We used a hidden Markov model (HMM) to detect latent states or patterns from envelopes of band-passed LFP signals. According to the model, network activity along the visual hierarchy is in one of $M$ hidden "states" at each given time. Each state is a vector, $S_{(a, d)}$, constituting a unique LFP power distribution over all depths ($d$ = [L2/3, L4, L5/6]) across six visual areas ($a$ = [V1 − AM]) in the cortex (emission matrix, Supplementary Fig. S3A). In an HMM-based system, stochastic transitions between states are assumed to behave as a Markov process such that the transition to a subsequent state solely depends on the current state. These transitions are governed by a "transition" probability matrix, $T_{m,n}$, whose elements represent the probability of transitioning from state $m$ to state $n$ at each given time (Supplementary Fig. S3B). We assumed the emission distribution to be Gaussian over the power signals to train a single HMM for each mouse, yielding the emission and transition probabilities between states. To match the frame rate of the natural movie, we averaged the power signals within non-overlapping windows of 30 ms. Each HMM was optimized using the Baum-Welch algorithm with a fixed number of hidden states, M.

In an HMM, the number of states, M, is a hyperparameter. To find the optimum number of states ($\acute{M}$), we used a majority rule across all mice. For each mouse, we optimized the 3-fold cross-validated log-likelihood (LL) estimate of the HMM fit, penalizing the metric if the inferred latent states were similar. The correction for similarity was imperative to determining distinct states with unique definitions. 'Similarity' between the states was quantified as the top eigenvalue of the state definition matrix evaluated as the mean power across the identified frequency ranges (number of states × number of frequency bands, Fig. 2C, right). The top eigenvalue represents the maximum variance in the matrix. In such a case, smaller values indicate lower variance in the definition matrix and, therefore, highly collinear state definitions. To apply this correction, we divided the log-likelihood estimate with the top eigenvalue where both metrics were individually normalized between -1 and 1 over a range of M $\in$ [2, 6]. Normalization was performed to allow equal weighting of the two metrics. The optimal number of states for each mouse was then identified as the number of states at which the ratio between the two metrics was maximized (Fig. 2C). In 24 out of 25 mice, the ratio consistently pointed to $\acute{M}$ = 3 optimal states (Fig. 2C) and was accordingly chosen as the optimal number of states for all HMMs fitted to the LFPs of each individual mouse. We performed several control analysis to confirm the optimal number of states.

- First, we validated our methodology (Supplementary Fig. S2B−D). For this, we applied our model selection criteria to data generated from an HMM with three states. We simulated a 10-dimensional time series with a duration of ~35 min, sampled at 30 Hz ($N$ = 60,000 samples), assuming diagonal Gaussian observations (Supplementary Fig. S2B). The simulation was repeated 30 times, with randomization of the emission covariance matrix for each run. In all simulations, our model selection criteria successfully identified the optimal number of states, where the ratio between the cross-validated log-likelihood (5 - fold) and the top eigenvalue peaked at $\acute{M}$ = 3 states (Supplementary Fig. S2C, D).
- Second, for each mouse, we considered the trend of cross-validated LL over a range of states (Supplementary Fig. S2E, F). As expected, the cross-validated LL increased (negative LL decreased) with the number of states until it reached a plateau

(Supplementary Fig. S2E). We selected the optimal number of states ($\acute{M}$) at the point where the incremental increase in the cross-validated LL had the largest drop (i.e., the point of greatest curvature, Elbow method[112]) before the plateau. In majority of mice, we found that three states were optimal, although in some cases, the Elbow method selected over-fitted models (Supplementary Fig. S2F).

- Next, we used K-means clustering to cluster all the input LFP variables between $K$ = 2 and $K$ = 6 clusters (Supplementary Fig. S2G). To determine the number of clusters (states), we applied the Elbow method to the percentage of variance explained by each clustering model. The percentage of explained variance is the ratio of the variance of the between-cluster sum of squares to the variance in the total sum of squares. Applying the Elbow method to each mouse, we selected the number of clusters, $\acute{K}$. In most mice, the LFPs optimally clustered into three or four separate groups, displaying a remarkably similar power distribution obtained via the HMM.
- Lastly, we applied dimensionality reduction to the input LFP variables using UMAP (Uniform Manifold Approximation and Projection[113]) and evaluated the silhouette scores on the reduced input matrix based on three HMM states (Supplementary Fig. S2H). The distribution of the silhouette scores across all mice further confirmed our model selection.

The LFP variables supplied to the HMM model include LFPs from one randomly selected channel from each layer of the cortical column: L2/3, L4, and L5/6, across all six visual area. This approach aims to achieve smoother states by reducing the number of observation variables provided to the HMM model while ensuring representation across the cortex. We validated this selection using two controls. First, we tested if latent states varied across visual areas. For this, we estimated HMM states using LFPs from each individual area (Supplementary Fig. S1B). Second, we conducted a randomized control test for each session, running 20 independent HMM fits with randomly selected LFP channels from each layer (Supplementary Fig. S1E). The initial guesses for emissions and transition probabilities were kept constant across different runs. Subsequently, for each test, we evaluated the pairwise correlations between state predictions for each pair of the HMM models. The correlation coefficients averaged around $0.54 \pm 0.04$ (mean ± sem, $n$ = 25 mice, Supplementary Fig. S1C) for the area-wise control and around $0.75 \pm 0.04$ (mean ± sem, $n$ = 25 mice, Supplementary Fig. S1F) for the layer-wise control, indicating the robustness of the determined states against area and channel selection.

## Behavioral features

Two synchronized cameras were used to record the mice: one focused on the body at a 30 Hz sampling rate, and the other an infrared camera focused on the pupil at a 60 Hz sampling rate. Running wheels were equipped with encoders to measure distance and speed of the mouses' running during the data acquisition session. Behavioral variables used in regression analyses were quantified using universal mouse models constructed using DeepLabCut[8,22] for pupil size changes and using SLEAP[7] for limb-to-tail movements. SLEAP, a modular UNet-based machine learning system, was trained to recognize up to 7 tracking points on the mouse's body, including the body center, forelimbs, hindlimbs, and the proximal and distal ends of the tail (Fig. 2C). However, the right forelimb was frequently occluded from view and subsequently dropped from analyses. We trained the model on a combined 1311 labeled frames from across all mice, with annotations ranging from 10−300 frames per mouse. Utilizing SLEAP's human-in-the-loop workflow, we alternated between labeling and training the model to achieve incremental improvements in prediction. In frames with resolutions of 478 × 638 pixels, the final model reported an

average pixel error of $7.15 \pm 4.1$ (mean $\pm$ std, $n = 1311$ frames) pixels across all body parts. Input features for the regression models were generated as smoothed Euclidean distances between coordinates of each body part in consecutive frames. Additionally, facial movements were quantified using face motion energy from cropped behavior videos[18]. At each time point, this energy was determined as the sum of the absolute differences between consecutive frames. Lastly, the full set of methodological details for pupil tracking can be found in Siegle, Jia, et al.[22].

Lastly, to identify the behavioral features that could generate states most aligned with the internal brain states, we applied Hidden Markov Models (HMMs) to separately infer behavioral states from the mouse's running speed, pupil size, and face motion. Each HMM was modeled with the same number of states as the internal brain states ($\acute{M} = 3$), ensuring consistency in state comparison. For each feature, we constructed behavioral state sequences by fitting an HMM using Gaussian emissions to model the continuous data. The transition probabilities were learned from the data, using the Expectation-Maximization (EM) algorithm for maximum likelihood estimation. After deriving the behavioral state sequences for each feature, we evaluated the Pearson correlation coefficient between each behavioral state sequence and the internal state sequence derived from cortical LFPs (Supplementary Fig. S5B).

### Variability metrics

**Shared variance.** To investigate the co-variation of diverse neurons within a population, we employed linear dimensionality reduction techniques, as summarized in Williamson et al.[55]. Specifically, we utilized factor analyses (FA) to quantify the percentage of variance shared across neural populations in the visual cortex. FA explicitly divides the spike count covariance into two components: a shared component and an independent component. The shared component captures the variability that is common across neurons within the recorded population, while the independent component quantifies the Poisson-like variability specific to each individual neuron. The FA analysis is performed on a matrix, $\mathbf{x} \in \mathbb{R}^{n \times T}$, comprising spike counts from $n$ simultaneously recorded neurons, along with a corresponding mean spike count vector, $\boldsymbol{\mu} \in \mathbb{R}^{n \times 1}$. As illustrated in Fig. 4D, FA effectively separates the spike count covariance into the shared component represented by $LL^T$ and the independent component represented by $\Psi$.

$$\mathbf{x} \sim \mathcal{N}(\boldsymbol{\mu}, LL^T + \Psi_{kk}) \tag{3}$$

Here, $L \in \mathbb{R}^{n \times m}$ is the loading matrix that relates the '$m$' latent variables to the neural activity, and $\Psi$ is a diagonal matrix comprising independent variances of each neuron. We calculated the percent shared variance for each neuron by utilizing the model estimates of the loading matrix, $L$, and the diagonal matrix, $\Psi$. This enabled us to quantify the degree to which the variability of each neuron was shared with at least one other neuron within the recorded population. For the $k$th neuron, the percent shared variance was evaluated as follows:

$$\% \text{ shared variance} = 100 \times \frac{L_k L_k^T}{L_k L_k^T + \Psi} \tag{4}$$

For our analyses, the FA model parameters, $\mu$, $L$, and $\Psi$, were estimated using singular-value decomposition (sklearn.decomposition.FactorAnalysis). The number of latent variables, $m$, was determined by applying FA to the spike counts and selecting the value for $m$ that maximized a three-fold cross-validated data likelihood ($m = 24 \pm 3$ factors, mean $\pm$ std). Spike counts were evaluated in 30 ms bins and values of shared variances averaged over all neurons in the given analyses (Fig. 4D, Supplementary Fig. S11G).

In Fig. 4D, state-specific shared variance for each neuron was evaluated on spike count matrices, $\mathbf{x} \in \mathbb{R}^{n \times T_s}$, comprised of concatenated epochs from each state. This allowed us to assess how much variability each neuron shared with others during specific oscillation states. Given the sensitivity of factor analysis to sample size differences and the varied occupancy of internal states, it was essential to account for these differences[114]. To address this, we evaluated shared variance of single units while matching sample sizes across states for each subject. Specifically, we matched sample sizes across states by identifying the state with the lowest occupancy for each subject and using bootstrapping ($n = 20$ repeats) on the other two states to equalize their sample sizes. In each bootstrapped iteration, we determined the optimal number of components for factor analysis using three-fold cross-validation of the log-likelihood estimate. The shared variance values reported in Fig. 4D represent averages across all neurons, with each neuron's shared variance calculated as the average over all bootstrapped repeats.

**Coefficient of variation.** In our study, we investigate the spike timing variability of single neurons by analyzing the distributions of their inter-spike-intervals (ISIs). To achieve this, we constructed histograms of the ISIs and quantified their characteristics using the coefficient of variation (CV). The CV is a dimensionless metric that represents the relative width of the ISI histogram. It is calculated as the ratio between the standard deviation of the ISIs ($\sigma_{\Delta t}$) and their mean ($\overline{\Delta}t$). The CV values reported in Fig. 4E, Supplementary Fig. S11G represent the average across all single units.

$$CV = \frac{\sigma_{\Delta t}}{\overline{\Delta t}} \tag{5}$$

To evaluate the coefficient of variation (CV) of individual neurons across different states (Fig. 4E), we created histograms of their inter-spike-intervals (ISIs) based on the spike times observed within each state. Since large differences in dwell times across states could bias the ISI ranges and, consequently, the state-specific CVs, we fixed the range of the ISI histograms. We selected an interval of $t = 2.5$ s, the point where the CV showed the largest incremental increase before plateauing (Supplementary Fig. S6B). In addition, we accounted for differences in firing rates across states by equalizing the firing rate of each unit to match its lowest rate observed across states. For each unit, the CV was then evaluated as an average over 20 repeats and the CV values reported in Fig. 4E represent the average across all single units.

**Fano factor.** We evaluated the trial-to-trial variability of neuronal activity in the visual cortex using Fano factor (FF), calculated as the ratio of variance to the mean spike count across trials, respectively. Similar to previous studies in the visual cortex[57,61], we computed the FF of each neuron within non-overlapping windows of 150 ms and averaged it across time. FF values reported in Fig. 4F, Supplementary Figs. S6C, D, G, S11G present the average FF across units. For all analyses, FF was evaluated only on units whose receptive fields were at least 20 degrees away from the monitor's edge.

Quantifying trial-wise variability in a state-specific manner posed unique challenges (Fig. 4F). Partitioning each session into states over time disrupted the trial structure, necessitating an additional constraint over the number of trials in each window. A time-window was considered for FF evaluation if the mouse remained in the same state across atleast 10 trials for the complete duration of the time-window (150 ms). To account for variations in firing rate and sample size across states, we used bootstrapping ($n = 20$ iterations) to equalize both firing rates and sample sizes. In each bootstrapping iteration, we first matched the sample sizes across states and then dropped spikes to equalize firing rates across states. The FF values reported in Fig. 4F are

averaged across all units, with each unit's FF calculated as an average over the bootstrapped repeats.

A classical metric for assessing variability, the Fano factor (FF), is traditionally evaluated under controlled experimental conditions to manage fluctuations in the subject's gaze[61,70,87]. Additionally, as receptive field size increases along the visual hierarchy, eye movements can significantly influence FF trends. To address these effects (Supplementary Fig. S6E), we evaluated FF by clustering the mouse's gaze into small, consistent bouts of fixed gaze and calculating the metric across trials within each cluster. For each mouse, we applied hierarchical clustering to the gaze data, dividing it into five clusters to identify bouts of stable gaze (Supplementary Fig. S6F). Given that mice lack a fovea and the standard receptive field size in their primary visual areas is ~20°, this clustering approach ensured that the center of the mouse's eye position remained within these constraints (-23.5 ± 1. 5°, mean ± SEM). We then partitioned each session by state and computed the Fano factor for each neuron, averaging it across time. This method of constraining the mouse's gaze eliminated area-wise differences in the Fano factor across all states (Supplementary Fig. S6G).

## Mutual information

Mutual information (MI) measures the reduction in uncertainty about one random variable when the value of another variable is known[115]. For two variables, $X$ and $Y$, it is calculated as the difference between the total entropy of $X$, denoted as $H(X)$, and the entropy that remains in $X$ after learning the value of $Y$, referred to as the conditional entropy $H(X|Y)$.

$$MI(X; Y) = H(X) - H(X|Y) \qquad (6)$$

$$= H(X) + H(Y) - H(X, Y) \qquad (7)$$

Similar to correlation, MI is symmetric in $X$ and $Y$, meaning that $MI(X; Y) = MI(Y; X)$. This is evident when MI is re-written in terms of joint entropy between the variables (equation (7)). However, MI surpasses correlation in its capacity to capture non-linear connections between variables. Given that responses of visual neurons can be highly non-linear functions of the visual input, we favored MI as our primary metric to quantify the amount of pixel-level information embedded in the neuronal activity of the visual cortex. Yet, calculating entropy requires knowledge of the joint probability distribution function (pdf) of the random variables, which is often unavailable. Many studies resort to 'plug-in' estimators that involve intricate evaluations of individual pdfs, a particularly onerous task for sizable datasets like ours. To sidestep the need for pdf estimation, we employed a matrix-based entropy estimator whose properties have been shown to align with the axiomatic properties of Renyi's $\alpha$-order entropy ($\alpha > 0$)[47].

Here, we provide a brief description of the process of entropy evaluation using the estimator, for specific details see Giraldo et al.[47]. First, the sample variable, $X = [x_1, x_2, ..., x_N] \in \mathbb{R}^{N \times M}$, is projected into a reproducing kernel Hilbert space (RKHS) through a positive definite kernel, $\kappa : \mathcal{X} \times \mathcal{X} \mapsto \mathbb{R}$. Next, a corresponding normalized Gram matrix, denoted as $A$, is generated from the pairwise evaluations of the kernel, $\kappa$. In this matrix, each entry $A_{i,j}$ is calculated as $\frac{1}{T} \frac{K_{ij}}{\sqrt{K_{ii}K_{jj}}}$, where $K_{i,j} = \kappa(x_i, x_j)$ and $K \in \mathbb{R}^{N \times N}$. The entropy estimator then defines entropy using the eigenspectrum of the normalized Gram matrix $A$, following the equation (8), where $\lambda_j(A)$ represents the $j^{th}$ eigenvalue of matrix $A$. Finally, the joint entropy, $H(X, Y)$ or $S_\alpha(A, B)$, is evaluated as the entropy of the Hadamard product, $A \circ B$ (equation (9)), where $B$ is the normalized Gram matrix associated with $Y$. The Hadamard product is interpreted as computing a product kernel, $\kappa((x_i, y_i), (x_j, y_j))$.

$$S_\alpha(A) = \frac{1}{1 - \alpha} log_2 \left[ \sum_{j=1}^{N} \lambda_j(A)^\alpha \right], \qquad (8)$$

$$S_\alpha(A, B) = S_\alpha \left( \frac{A \circ B}{trace(A \circ B)} \right) \qquad (9)$$

In our analyses, we consider $X = [x_1, x_2, ..., x_N] \in \mathbb{R}^{N \times M}$ to represent the spike count matrix for all neurons in the population, where each $x_i \in \mathbb{R}^M$ is a vector containing spike counts from $M$ neurons at time $i$. Similarly, $Y = [y_1, y_2, ..., y_N] \in \mathbb{R}^{N \times P}$ is a matrix containing image pixels, with each $y_i$ representing a flattened vector of all pixels in the stimulus image at time $i$. For state-wise analyses of MI (Fig. 4G), we exclusively considered times corresponding to the specific state under examination. MI was computed per trial, but only when the subject had spent at least 3 s in the particular state during the trial, i.e., $i \in [3, 30]$ seconds. Each frame was downsampled by a factor of 5, and spike counts were evaluated in 30 ms bins to match the stimulus frame rate. To constrain the metric between [0, 1], all MI measures were normalized by the geometric mean of the individual entropy of the two variables, $S_\alpha(A)$ and $S_\alpha(B)$[116]. The values presented in the paper are averages taken across all subjects (Figs. 3F, 4G).

Entropy estimation is dependent on two hyperparameters: the order, $\alpha$, and the kernel, $\kappa$. Given the sparsity of neural activity data, we chose the order, $\alpha$, to be 1.01. Next, $\kappa$ is a positive definite kernel that determines the RKHS and thus dictates the mapping of the probability density functions (pdfs) of the input variables to the RKHS. For our analyses, we employed a non-linear Schoenberg kernel (equation (10)). These positive definite kernels are universal, in that, they have been proven to approximate arbitrary functions on spike trains[117]. The window to evaluate spike counts was set to 30 ms to match the frame rate of the visual stimulus, and the kernel width, $\sigma_k$, was determined using Scott's rule[118].

$$\kappa(x_i, x_j) = \exp \left\{ \sum_{m=1}^{M} -\frac{1}{\sigma_\kappa} (x_{i,m} - x_{j,m})^2 \right\} \qquad (10)$$

## Stimulus features

Capitalizing on the ethological significance of a naturalistic stimuli[119-121] and to mitigate sudden changes in neural activity due to abrupt changes in visual stimulus, our analysis centered on neural data obtained from repeated viewings of a 30 s natural movie clip. We anticipated that the statistical properties of the clip would significantly contribute to explaining neuronal variability. In order to reveal any statistical preferences of neurons across the cortical hierarchy, we constructed stimulus features from both low- and high-order (>second-order moments) properties of the pixel distribution. The low-order features included image intensity and contrast, whereas, the high-order features included kurtosis, entropy, energy, and edges.

*Intensity and contrast:* These metrics captured the first and second order statistics of the image, and they were evaluated as the mean ($\mu_m$) and standard deviation ($\sigma_m$) of all the pixel values in each image frame, $I$, respectively.

*Kurtosis:* A higher-order statistic of the pixel distribution, Kurtosis measures the extent to which pixel values tend to cluster in the tails or peaks of the distribution. This metric was computed on the distribution of pixels within each image frame by determining the ratio between the fourth central moment and the square of the variance.

$$Kurt[I] = \frac{E\left[(I - \mu_m)^4\right]}{\sigma_m^4} \qquad (11)$$

*Entropy:* To assess the average information content within each image frame, entropy was calculated based on the sample probabilities ($p_i$) of pixel values spanning the range of 0–255.

$$H[I] = - \sum_{i=1}^{n_{pixels}} p_i log_2(p_i) \qquad (12)$$

*Energy*: Similar to the quantification of face motion energy[18], we evaluated image energy as the absolute sum of the differences between the pixel values of consecutive frames.

$$E[I] = \sum_{i=1}^{n_{pixels}} |I_t - I_{t-1}| \qquad (13)$$

*Edges*: Given the observed line and edge selectivity of visual cortical neurons[122], we devised this metric to quantify the fraction of pixels that contribute to edges within a given image frame. For the identification of edges in each frame, we employed Canny edge detection (cv.Canny). This technique involves several sequential steps. First, a 2D Gaussian filter with dimensions of 5 × 5 pixels was applied to the image to reduce noise. Subsequently, the smoothed image underwent convolution with Sobel kernels in both horizontal and vertical directions, producing first derivatives along the respective axes, as described in equations ((14)–(15)). The resulting edge directions ($\Theta$) were approximated to one of four angles: [0°, 45°, 90°, 135°]. To refine the edges, a process called edge thinning was used. During this step, the entire image was scanned to locate pixels that stood as local maxima within their gradient-oriented vicinity. These selected pixels moved on to the subsequent phase, while the rest were set to zero. Lastly, two threshold values were introduced for edge identification. Edges with intensity gradients below the lower threshold were disregarded, whereas those with gradients above the higher threshold were retained as 'sure edges'. Pixels with gradient intensities falling between these two thresholds were analyzed based on their connection to a 'sure' edge. Ultimately, the output of the Canny edge detector was a binary image outlining the edge-associated pixels. The metric 'edges' was computed as the mean value of this binary image.

$$gradient = \sqrt{G_x^2 + G_y^2} \qquad (14)$$

$$\Theta = tan^{-1} \frac{G_y}{G_x} \qquad (15)$$

## Input data for the neural encoding model

Identical set of features were employed to predict both averaged neuronal population activity and single neuron responses. These features were grouped into three distinct categories to evaluate the respective contributions of each set of variables. The categorization of features is as follows: 1. stimulus features, 2. behavioral features, and 3. features encompassing internal brain dynamics, which included raw LFPs from the same cortical area, as well as averaged neuronal population activity from visual areas other than the target area. For raw LFPs, representative channels were once again selected across the cortical depth, ensuring the inclusion of one channel from each layer. Stimulus and behavioral features were sampled at a frequency of 30 Hz. However, to align with this temporal resolution, both LFPs and averaged population activity were binned into 30 ms bins, where each bin represented an average signal value within the respective time window.

The broad range of input features exhibited pronounced intercorrelations, and constructing an encoding model using a design matrix containing linearly dependent columns inherently jeopardizes model reliability. To avoid this multicollinearity in the design matrix, we systematically orthogonalized the input features using QR decomposition[63]. QR decomposition of a matrix, denoted as $M \in R^{m \times n}$, yields $M = QR$, where $Q \in R^{m \times n}$ denotes an orthonormal matrix and $R \in R^{n \times n}$ represents an upper triangular matrix. Consequently, matrix $Q$ spans the same space as the columns of $M$, ensuring that the columns of $Q$ maintain mutual orthogonality. As QR decomposition systematically decorrelates each column from all preceding ones, the arrangement of columns within the matrix becomes pivotal.

Prior to constructing the time-shifted design matrix, we first orthogonalized internal brain activity relative to all other input features, positioning these columns towards the latter part of the matrix, $M$. This step was aimed at reducing the potential influence of stimulus and behavior features on brain activity[17]. We retained the original definitions of stimulus features due to their limited correlations within and across groups ($r_{within} = 0.3 \pm 0.1$, $r_{across} = 0.06 \pm 0.1$, mean ± std Fig. 5A, panel 2). Given the strong correlations between behavioral features ($r_{within} = 0.4 \pm 0.2$, $r_{across} = 0.07 \pm 0.07$, mean ± std), we applied QR decomposition to decorrelate all behavioral variables among themselves. The final collection of input features for the full model comprised behavioral features that had undergone orthogonalization among themselves, stimulus features in their original form, and internal brain activity features that were orthogonal both within and across the categories of features. Next, each input signal of length $\tau$ was organized such that each row consisted of variables shifted in time by one frame (30 Hz) relative to the original, also known as a Toeplitz matrix. Lastly, to structure the design matrix, the various input signals were time-aligned and concatenated. Including a time-shifted design matrix enabled us to account for the temporal dependency between various sources and neural activity. To determine the appropriate time dependency for each type of neural data (averaged neuronal population and single neuron activity), we tested a range of values (population model: [0.2–6]s, single-neuron model: [0.2–2]s) and chose the dependency that maximized the model's cross-validated explained variation, $cvR^2$ (Supplementary Fig. S7E–H).

Lastly, when quantifying group-specific contributions using unique models, the features of internal brain activity were orthogonalized only within the group. This approach was taken to prevent partial decorrelation across groups, as the designed stimulus features and behavioral features might not encompass the entire array of features encoded in neural activity. Such partial decorrelation could potentially obscure the interpretability of the contributions from each category of input features to spiking variability.

## Internal state conditioned neural encoding model

The encoding model was constructed to predict the averaged neuronal population activity and single-neuron spike rates. Unlike classical linear prediction models that assume constant relative contributions of various sources to spiking variability, our encoding framework deviates from this assumption by accounting for variations in contributions resulting from internal state fluctuations. To achieve this, each predictor model learns regressors only from signals associated with a state. This approach enables state-specific investigations of the relative contributions across the three distinct sources of variability outlined earlier. Each predictor model is tailored specifically to the neural activity in each state. Importantly, it should be highlighted that the HMM states are held constant. In other words, the HMM model is not optimized to improve predictions but maintains its established definitions based on LFPs. To quantify the contributions of the variability sources to the averaged population activity, we used ridge regression, whereas spiking activity was modeled using a generalized linear model (GLM).

**Population model.** To mitigate overfitting, the population model was trained with ridge regression. Ridge regression extends the cost function of ordinary least squares by introducing an additional $l_2$ penalty, ($\lambda$), on the regression coefficients ($\beta$). This penalty effectively shrinks the coefficients of input variables that contribute less to the prediction, promoting smoother and more generalizable regression coefficients (equation (16)). In our HMM based regression model, the design matrix $X_s$ and the regressand, $y_s$, are informed by the HMM, comprising signals corresponding to one of three identified states ($s = [S_H, S_I, S_L]$). The magnitude of the regularization penalty, $\lambda_s$, for weights in each state were individually determined through three-fold cross-validation of $R^2$ on a randomly selected 30% subset of the dataset.

$$\min_{\beta_s}(y_s - X_s\beta_s)^T(y_s - X_s\beta_s) + \lambda_s\beta_s^T\beta_s \qquad (16)$$

**Single neuron model.** A regularized Poisson GLM was used to model the firing rate of each neuron while taking into account variances associated with internal state fluctuations (s = $[S_H, S_I, S_L]$). The encoding model describes spike counts of single neurons as a Poisson distribution whose expected value can be modeled as the exponential of the linear combination of input features, i.e, $E(y_s|X_s) = e^{\theta_s^T X_s}$. The coefficients of the regression model, $\theta_s$, are then estimated by penalized maximum likelihood with an $l_2$ penalty on the coefficients (equation (17))[123]. To avoid overfitting, the magnitude of the regularization penalty, $\lambda_s$, for weights for each neuron in each state were individually determined using nested-five-fold cross-validation of $R^2$ during training[124].

$$\max_{\theta_s}L(\theta_s|X_s, y_s) = log\left(p\left(y_s; e^{\theta_s^T X_s}\right)\right) - \lambda_s\theta_s^T\theta_s \qquad (17)$$

The final evaluation of the reported scores (Figs. 5, 6) includes a five-fold cross-validation of explained variance ($cvR^2$, equation (18)), where $\hat{y}$ is the predicted spike rate and $\bar{y}$ is the mean of the true spike rate. The $cvR^2$ values in Fig. 5 were computed on spike counts of single neurons smoothed with a 50 ms Gaussian for each trial.

$$R^2 = 1 - \frac{\sum_i(y_i - \hat{y}_i)^2}{\sum_i(y_i - \bar{y}_i)^2} \qquad (18)$$

To quantify the state-wise contributions of the input features, we partition the dataset into training and testing sets such that each fold contains an equitable representation of signals from every state. This step was crucial to prevent any potential biases in estimating contributions due to an imbalance in the number of data points in each state. State-specific contributions were evaluated on the respective performance of the state-wise regressors, while overall performance was evaluated by concatenating the predictions across the three state models in each fold.

### Clustering of neuronal encoding patterns

To identify neuronal clusters with similar source contributions, we performed clustering on their encoding patterns. The feature coding for each neuron was represented by a five-element vector, which comprised the cross-validated explained variance from each source category: stimulus, behavior, local field potentials (LFPs) from the target area, averaged population activity from other visual areas, and the number of source categories with more than 10% explanatory power. All single units that passed the selection criteria for inclusion in the single-neuron model were included in this clustering analysis.

We implemented a clustering workflow that used mean-shift clustering in conjunction with a consensus clustering method[125] to reduce sensitivity to random initial conditions and hyperparameter selection (Supplementary Fig. S9A, left). To minimize noise, we applied UMAP (Uniform Manifold Approximation and Projection)[113], a manifold-based dimensionality reduction technique, to project the feature matrix into a lower-dimensional space.

To achieve stable co-clustering results, we first built a co-clustering association matrix by performing 200 repeats of mean-shift clustering on the UMAP projections. Each entry in the co-clustering association matrix represents the probability of two units belonging to the same cluster (Supplementary Fig. S9A, top-right). Since UMAP projections are highly sensitive to hyperparameter selection, we randomized these hyperparameters in each iteration (min _dist $\in [0.02, 0.5]$, n_neighbors $\in [5, 10]$, n_components $\in [2, 5]$). Overall, each iteration included a random initialization of the mean-shift clustering model as well as a random dimensionality reduction.

Next, we clustered the association matrix with hierarchical clustering to determine the cluster labels (Supplementary Fig. S9A, center-right). The optimal number of clusters was determined by evaluating the silhouette scores on a range of clusters obtained via hierarchical clustering (Supplementary Fig. S9A, bottom-right), and selecting the point of maximum curvature as the optimal number of clusters[112]. Clustering based on the final explained variance of each unit resulted in 8 clusters (Supplementary Fig. S9B). We then manually combined clusters with dominant explained variances in the same source categories. Finally, using the same clustering workflow, we also clustered the units based on their state-specific explained variance values (Supplementary Fig. S9J, K).

### Reporting summary

Further information on research design is available in the Nature Portfolio Reporting Summary linked to this article.

## Data availability

Data is freely accessible via the Allen SDK[22] (Details at https://portal.brainmap.org/circuits-behavior/visual-coding-neuropixels). Source data for all figures are provided with this paper. Source data are provided with this paper.

## Code availability

The complete code for reproducing all figures is available on the author's GitHub repository[126]: https://github.com/shailajaAkella/Deciphering-neuronal-variability.

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

## Acknowledgements

Primary funding for this project was provided by the Tsinghua-Peking Center for Life Sciences and the Allen Institute to X.J. Additional funding was provided by the National Natural Science Foundation of China (92370116), and Tsinghua University Initiative Scientific Research Program to X.J. We thank the Allen Institute founder, Paul G. Allen, for his vision, encouragement and support. We thank Ruben Coen-Cagli, Ramakrishnan Iyer and Eero Simoncelli for suggestions during the initial phase of the project; Scott Linderman, Lukasz Kusmierz, Eric Shea-Brown, Nick Steinmentz and Ying Zhou for discussions; Linzy Casal for assistance with planning and budgeting of the project; Benjie Miao for technical support.

## Author contributions

Conceptualization: X.J., S.A. Supervision and funding acquisition: X.J. Investigation and formal analyses: S.A., X.J. Implementation: S.A. Validation and methodology: S.A., X.J., P.L., J.H.S., S.R.O., M.A.B, D.D. Software and visualization: S.A., X.J., S.R.O. Data collection: S.D., H.B., X.J., J.H.S. Original draft written by S.A., X.J., with input and editing from S.R.O., J.H.S, P.L., M.A.B, D.D., C.K., H.B., S.D. All co-authors reviewed the manuscript.

## Competing interests

The authors declare no competing interests.
