## [Transparent Peer Review file · Nature Communications]

Deciphering neuronal variability across states reveals dynamic sensory encoding

Corresponding Author: Dr Shailaja Akella

Version 0:

Reviewer comments:

Reviewer #1

(Remarks to the Author)

The authors address the long-standing problem of identifying the origin of the variability in neuronal responses to stimuli. Their results provide a comprehensive description of the non-stationary aspects of spiking variability in the visual cortex, by explaining how such variability depends on three internal brain states identified in the cortical LFP power spectrum. They show that neural variability in these different brain states is differentially modulated by stimulus, behavior, and activity in other brain areas. They found that sensory inputs and behavior exerted the most influence in the brain state dominated by high-frequency oscillations. Projections from other brain areas explained most of the variance in the state dominated by low-frequency oscillations.

The results are extremely interesting and novel, and provide crucial new insight into the long-standing question of what the sources of neural variability are. The approach and methods are sophisticated and the combination of HMM-based identification of brain state and state-dependent encoding models is new. These methods will likely be adopted widely by the systems neuroscience community. The paper is very well written, including comprehensive introductions and discussions and a detailed Methods section.

The results are supported by the methods and their analyses. However, I identified some issues with the statistical analyses that require additional controls to confirm the results before publication. I strongly recommend the paper for publication, provided these concerns are properly addressed. Please find my comments below.

Main concerns

HMM analysis

Model selection. Is model selection performed on the concatenated data from all mice, or rather is it performed on single mice? What are the points in Fig. 2C?

Usually model selection for HMMs is performed using cross-validated MLE/MAP estimate, or in alternative BIC/AIC. The authors are using an interesting yet very unusual criterion combining likelihood and state similarity. I recommend further validating this new procedure with the following controls. First, please compare this with more standard procedures such as BIC/AIC or the elbow of a hold-out log-likelihood. Second, apply this model selection procedure to data generated from an HMM with the three states and the same sample size as the data, and check whether this procedure leads to recovering the correct number of states.

Neural variability

The comparison of neural variability between conditions is challenged by the fact that most measures, such as the ones adopted by the authors, strongly depend on changes in firing rates and on differences in sample size. The authors should perform a series of controls to make sure the results are not due to these two confounding effects.

For comparison of factor analysis between condition, please note that such measures strongly depend on the number of samples used for estimation (Recanatesi et al, Neuron, 2022; Mazzucato et al, Frontiers, 2016). This is a critical control given the fact that different HMM states have different occupancy. Specifically, conditions with larger sample size have larger

shared covariance (Mazzucato et al, Frontiers, 2016). Therefore, the authors should first equalize sample sizes, for which we recommend eliminating bins from conditions with more bins until all conditions have the same number of bins to run factor analysis.

For the CV of ISI, I recommend an alternative way to control for different durations of the three HMM states: given three distribution of ISI with N_s samples (ISI) in each state s , pick the smallest N_s among the three and then randomly subsample the other distributions to obtain the same sample size, then perform statistical tests. This procedure avoids choosing an interval as in Fig. S6B and controls for the bias introduced by differences in sample size between conditions.

Comparison of Fano factor between different conditions is very sensitive to firing rate and sample size (see above). Because of the large difference in both firing rate and sample size between different HMM states. First, in order to equalize firing rates the authors should proceed as follows (Papadopoulos et al, 2024, biorxiv). Given average firing rate r_s in HMM state s , pick the smallest r_s among the three states and randomly eliminate spikes from the others to equalize the firing rate. Then, to equalize sample sizes, we recommend eliminating bins from conditions with more bins until all conditions have the same number of bins to estimate the spike count variance.

Because I believe the sample size confound goes in the same direction as their trends in Fig. 4, the proposed controls are necessary to confirm the results.

Regarding the mutual information between neural activity and stimulus, its comparison between HMM states is affected by a critical confound that the authors must address. It is known that neural populations in V1 (Niell & Stryker, 2010) and in higher visual areas (Wyrick et al, J Neuro, 2021) encode more information about visual stimuli during running compared to resting periods. This effect was believed to be due to a decrease in noise correlations during running periods (Dadgarlat et al, J Neuro, 20), and the results in Fig 4G are consistent with the literature. However, recent work (Wyrick et al, J Neuro, 2021) showed that the increase in stimulus-encoding information during running is entirely due to the firing rate increase induced by running. Once controlling for the difference in firing rate (by equalizing firing rates between conditions), the peak visual decoding performance (or in other words, the information per spike) is the same during running vs resting. Please perform a control analysis by equalizing the firing rates between the three HMM states and then reestimating the MI, I expect the difference between conditions to go away.

For the bottom plots in Fig. 4D-E-F-G, please perform a 2-way anova with factors hierarchy score and HMM state and report p-values for the two factors and interaction term.

HMM - predictor model

The "HMM based predictor model" is not itself an HMM, but rather, it is a set of 3 different GLMs, each one conditioned on the HMM internal states $S_{L,I,H}$ obtained in the previous section. Therefore, I recommend the authors rephrase the lines 218-228 and change notation to clarify this point, replacing "HMM based predictor model," "HMM-GLM" and similar notations throughout the paper with "Neural encoding model conditioned on internal states".

What is missing here and in the previous section on neural variability is a direct estimation of how much variance of population activity and of single neuron activity can be explained directly and solely from the knowledge of the three brain states $S_{\{I,H,L\}}$. Namely, the authors estimate the cvR^2 of the various regressors, conditioned on the brain states, but don't explicitly measure the variance explained by knowledge of the three states. I believe this can be done directly using the ANCOVA or analysis of covariance that combines continuous and categorical regressors (the three brain states). Is there a reason why the authors did not perform this analysis? If not, I recommend they carry this out in the revised version for both the population averaged activity and for the single cell activity.

In Figg. 5 and 6, several analyses include comparison of the contribution to population activity in one area of the brain activity from other areas. Because area have different number of neurons per area (the authors mention this issue in the Discussion), this comparison can only be performed after equalizing the neuron number from other areas by subsampling neurons to the smallest common sample size. Please fix this control.

line 308: "individual contributions from

308 different sources showed a reversed trend compared to the population model." This is a very strange result. How is that possible, since the population average activity is the average of the single neuron activity. The authors need to explain this contradiction in their results.

When repeating the encoding model on other stimulus classes such as drifting grating do they get similar results? The natural images might have much sparser evoked responses, but the drifting grating should drive neurons enough to be able to create an encoding model.

Minor concerns

- line 33: Ashwood et al. 2022 does not analyze neural activity but just behavior.

Fig. 2E: what are the shaded lines representing? Different areas and layer or different recording sessions?

The authors frequently mention "inputs to the HMM" but these actually refer to the observations. Input-driven HMMs are a different kind of model, so please adjust the notation to avoid confusion.

- S4D, there's a 1 in the tpm, please replace with actual value with 3 digit precision.

HMM - Behavior

The HMM fits to behavior and comparison with LFP HMMs in Fig. S5B are not explained, can you please elaborate on it? I don't see behavioral HMM in this figure, but only LFP HMMs. What correlation has been estimated in panel B exactly? No mention of the behavioral HMM is done anywhere, including the Methods section.

line 171: The following interpretation needs clarification: "Thus, we attribute the observed decreasing trends to rapid variations in luminance or moving edges in the natural movie, that likely induce stronger temporally coherent activity within a population in lower visual areas than in higher visual areas."

- Internal states measured with pupil, whisking and running have been recently shown to predict task performance states in Hulseley et al, Cell reports, 2024; please add discussion.

Fig. 4E: please add legend for color codes.

Fig. 5D, is this the performance of the average between the three models obtained from the three different HMM states $S_{\{I,L,H\}}$? Please clarify.

Fig. 6F: "variability" typo

Discussion

Regarding the fact that "Recent studies have shown that a subject's engagement during an active task varies drastically from trial to trial, playing out through multiple interleaved strategies", it's worth mentioning that recent work in Hulseley et al, 2024 showed that such changes in strategy are predicted by behavioral variables (pupil, whisking, running) suggesting a direct relationship between the findings in the current manuscript and task performance.

The authors should add a paragraph on what the neural mechanisms might mediate the relationship between state-dependence, variability and stimulus coding. One possible explanation is based on the fact that changes in brain state, captured by the three HMM states, modulate the intrinsic metastable activity of the cortical circuit. This mechanism explained the interaction between changes in brain states, neural variability and stimulus coding in gustatory cortex (Mazzucato et al, Nat Neuro, 2019), hierarchically connected visual areas (Wyrick et al, J Neuro, 2021) and auditory cortex (Papadopoulos et al, 2024); and primate V4 (Shi et al, Nat Comm, 2022; Zeraati et al, Nat Comm, 2023).

Reviewer #2

(Remarks to the Author)

In this paper, Akella et al. perform a comprehensive analysis of a widely used dataset in visual neuroscience. They focus on the question of how different sources dynamically contribute to the non-stationarity of neuronal variability in the mouse visual system. There are a lot of analyses and results reported in the manuscript. There is not a single key take-away, other than that spiking activity in the mouse visual system is driven by visual stimuli, brain states, and behaviors that are not directly linked to brain states. The paper will be interesting for other scientists working with this data set, but less so for a broad audience. In many places, there is an uncomfortable mismatch between the historical logic underlying the analysis of response variability and the realities of this type of experiment. In general, these analyses provide an estimate of the 'noisiness' of neural activity if you manage to measure this activity under repeated presentations of exactly the same experimental condition. That standard is absolutely not met here. There is likely trial-to-trial variability in the position of the eyes, resulting in stimulus-variability across repeated trials. There is clearly a lot of cross-trial variation in the animals' behavior (running at various speeds) and brain state (dozing off or being wakeful). These factors are known to systematically impact neural activity in visual cortex. As a consequence, it is unclear what statistics like the Fano factor or coefficient of variability try to measure here. As an example, V1 Fano factors were a bit higher than those in higher visual areas. Might that just reflect that V1 is more heavily modulated by running speed and this variable was not controlled for? What is the point of knowing a Fano factor for a set of trials that effectively come from different experimental conditions? Some of the findings seem a bit circular to me. For example, one measure of population activity (LFP) turns out to be a good predictor for a different measure of population activity (population rate). I believe that. I am not sure that it reveals a deeper insight. I found the most interesting finding to be at the end of the manuscript (Fig 6). This analysis pedals back some popular claims that behavior is the main driver of response variability in mouse V1. Apparently, this is not true if you look at single neuron spiking activity as opposed to (global) calcium measurements. I think this finding could be turned into a future paper that is interesting to a broader readership.

Minor comments

- Fig 1 presents this behavior, state, and stimulus as a 3-D space. This gives the wrong intuition. Internal fluctuations are so closely related to behavior that this leads the reader in wrong direction.
- Fig 1D: What are the different LFP traces? Is not described in legend.
- Fig 3E: What was the statistical test?
- Panel 4c, legend mentions P-value of 1.5
- Fig 6F seems bizarre in light of earlier FF result. How can 75% of V1 neurons have a value below 1? The grand average was substantially higher.
- Fig 7 is weird. It is 90% a summary of the methods used in this paper and only 10% a cartoon illustration of the results.

Version 1:

Reviewer comments:

Reviewer #2

(Remarks to the Author)

The authors have done an admirable job revising their manuscript. They have taken the reviewer suggestions to heart, have reorganized the paper to highlight a key finding, and conducted many control analyses that build further confidence in their work.

Response to Reviewer Comments

We thank all the reviewers for their thoughtful and constructive suggestions. We have addressed each point accordingly which significantly improved our manuscript. Please find the details below.

Reviewer #1: The authors address the long-standing problem of identifying the origin of the variability in neuronal responses to stimuli. Their results provide a comprehensive description of the non-stationary aspects of spiking variability in the visual cortex, by explaining how such variability depends on three internal brain states identified in the cortical LFP power spectrum. They show that neural variability in these different brain states is differentially modulated by stimulus, behavior, and activity in other brain areas. They found that sensory inputs and behavior exerted the most influence in the brain state dominated by high-frequency oscillations. Projections from other brain areas explained most of the variance in the state dominated by low-frequency oscillations.

The results are extremely interesting and novel, and provide crucial new insight into the long-standing question of what the sources of neural variability are. The approach and methods are sophisticated and the combination of HMM-based identification of brain state and state-dependent encoding models is new. These methods will likely be adopted widely by the systems neuroscience community. The paper is very well written, including comprehensive introductions and discussions and a detailed Methods section.

The results are supported by the methods and their analyses. However, I identified some issues with the statistical analyses that require additional controls to confirm the results before publication. I strongly recommend the paper for publication, provided these concerns are properly addressed. Please find my comments below.

Model Selection:

1. Is model selection performed on the concatenated data from all mice, or rather is it performed on single mice? What are the points in Fig. 2C?

We thank the reviewer for raising these clarifying questions. Model selection for the Hidden Markov Model (HMM) was based on a majority rule across mice. For each mouse, we calculated the cross-validated log-likelihood (LL) estimate of the HMM fit, normalized by the top eigenvalue of the state definition matrix. The optimal number of states for each mouse was identified as the number of states at which this normalized LL was maximized. In 24 out of 25 mice, the ratio peaked at three states (Figure 2C), which was then selected as the optimal number of states for each individual HMM fitted to the local field potentials (LFPs) from each mouse. Individual markers in Figure 2C represent the ratio for each mouse and number of states. These clarifications are now included in the text (in Methods, lines 663-665) and in the legend of Figure 2C.

Lines from the legend of Figure 2C: *“The cross-validated log-likelihood (LL) estimate, normalized by the*

top eigenvalue of the state definition matrix, is reported for each mouse (hollow circles) along with across-subject averages (solid circles). For each mouse, the optimal number of states was identified as the point where the normalized LL was maximized. Final model selection was based on the majority rule across all mice."

Lines 663-665 from the methods section: "To find the optimum number of states (M^*), we used a majority rule across all mice. For each mouse, we optimized the 3-fold cross-validated log-likelihood estimate of the HMM fit, penalizing the metric if the inferred latent states were similar."

2. Usually model selection for HMMs is performed using cross-validated MLE/MAP estimate, or in alternative BIC/AIC. The authors are using an interesting yet very unusual criterion combining likelihood and state similarity. I recommend further validating this new procedure with the following controls. First, please compare this with more standard procedures such as BIC/AIC or the elbow of a hold-out log-likelihood. Second, apply this model selection procedure to data generated from an HMM with the three states and the same sample size as the data, and check whether this procedure leads to recovering the correct number of states.

Figure R1.1: **Selection of optimal number of states:** **A** Model selection for a representative session, showing the minimization of negative log-likelihood. **B** Histogram of the optimal number of states across all mice, determined by identifying the point of maximum curvature in the plot of negative log-likelihood vs. number of states. **C** Simulated three-state HMM time-series, with each color representing a different state. Two dimensions of the 10-dimensional time-series are shown ($N = 60,000$ samples). **D** Identification of the optimal state from the simulated time-series. **E** Comparison between the inferred states using the HMM and the true states.

We acknowledge that our method is non-traditional and appreciate the reviewer's recognition of its potential to improve model selection. In response, we have now conducted the suggested control analyses.

First, we compared our model selection criteria with the traditional log-likelihood (LL) approach (Figure R1.1A, B) and found that three states were optimal for most mice. In most sessions, the cross-validated LL increased (negative LL decreased) with the number of states before plateau. By selecting the optimal number of states (M^*) at the point of maximum curvature for each session, we confirmed that three states were optimal in the majority of cases. However, the elbow method occasionally selected over-fitted models. We believe our approach, which penalizes models for selecting states with highly similar definitions, effectively mitigates such overfitting.

Second, to validate our model selection methodology, we applied it to data generated from an HMM with three states (Figure R1.1C). Using our methodology, we were able to successfully recover the optimal number of states in the simulated time-series (Figure R1.1D, E). We simulated a 10-dimensional time series with a duration of 35 minutes, sampled at 30 Hz, assuming diagonal Gaussian observations. The simulation was repeated 30 times, with randomization of the emission covariance matrix for each run. In all simulations, our model selection criteria successfully identified the optimal number of states, where the ratio between the cross-validated log-likelihood (5 - fold) and the top eigenvalue peaked at $M^* = 3$ states. Along with previous control analyses using K-means and post-hoc silhouette scores, these control analyses have now been included in the updated manuscript (Methods, lines 677 - 700, Figure S2).

Lines 677 - 700 from the methods section:

- *First, we validated our methodology (Figure S2B-D). For this, we applied our model selection criteria to data generated from an HMM with three states. We simulated a 10-dimensional time series with a duration of ~ 35 minutes, sampled at 30 Hz ($N = 60000$ samples), assuming diagonal Gaussian observations (Figure S2B). The simulation was repeated 30 times, with randomization of the emission covariance matrix for each run. In all simulations, our model selection criteria successfully identified the optimal number of states, where the ratio between the cross-validated log-likelihood (5 - fold) and the top eigenvalue peaked at $M^* = 3$ states (Figure S2C, D).*
- *Second, for each mouse, we considered the trend of cross-validated LL over a range of states (Figure S2E,F). As expected, the cross-validated LL increased (negative LL decreased) with the number of states until it reached a plateau (Figure S2E). We selected the optimal number of states (M^*) at the point where the incremental increase in the cross-validated LL had the largest drop (i.e., the point of greatest curvature, Elbow method, Satopaa et al. 2011) before the plateau. In majority of mice, we found that three states were optimal, although in some cases, the Elbow method selected over-fitted models (Figure S2F).*
- *Next, we used K-means clustering to cluster all the input LFP variables between $K = 2$ and $K = 6$ clusters (Figure S2G). To determine the number of clusters (states), we applied the Elbow method to the percentage of variance explained by each clustering model. The percentage of explained variance is the ratio of the variance of the between-cluster sum of squares to the variance in the total sum of squares. Applying the Elbow method to each mouse, we selected the number of clusters, K^* . In most mice, the LFPs optimally clustered into three or four separate groups, displaying a remarkably similar power distribution obtained via the HMM.*
- *Lastly, we applied dimensionality reduction to the input LFP variables using UMAP (Uniform Manifold Approximation and Projection, McInnes et al. 2018) and evaluated the silhouette scores on the reduced input matrix based on three HMM states (Figure S2H). The distribution of the silhouette scores across all mice further confirmed our model selection.*

Neural variability:

The comparison of neural variability between conditions is challenged by the fact that most measures, such as the ones adopted by the authors, strongly depend on changes in firing rates and on differences in sample size. The authors should perform a series of controls to make sure the results are not due to these two confounding effects.

We thank the reviewer for highlighting these challenges. Below, we outline the specific controls applied for each spiking variability metric. These results have now been incorporated into the main manuscript, and all corresponding plots in Figure 4D-F have been updated accordingly.

3. For comparison of factor analysis between condition, please note that such measures strongly depend on the number of samples used for estimation (Recanatesi et al, Neuron, 2022; Mazzucato et al, Frontiers, 2016). This is a critical control given the fact that different HMM states have different occupancy. Specifically, conditions with larger sample size have larger shared covariance (Mazzucato et al, Frontiers, 2016). Therefore, the authors should first equalize sample sizes, for which we recommend eliminating bins from conditions with more bins until all conditions have the same number of bins to run factor analysis.

Figure R1.2: **Factor analysis - Controlled for sample size differences across states:** Population shared variance evaluated after equalizing sample sizes across all states. Percent shared variance is plotted against the anatomical hierarchy scores of the visual areas in each oscillation state, averaged across all units (One-way ANOVA: $p_{S_H, S_I} = 9.6e-7$, $p_{S_H, S_L} = 1.3e-146$, $p_{S_I, S_L} = 1.2e-95$, $n = 7,609$ units).

We agree with the reviewer and we have now accounted for the potential biases in factor analysis that could arise from sample size differences across states. After correcting for these biases, we found that the trends across states and visual areas remained consistent despite the sample size differences. For each subject, we identified the state with the lowest occupancy and employed bootstrapping ($n = 20$ repeats) on the other two states to match their sample sizes to that of the state with the lowest occupancy. The final shared variance for each unit was then calculated as the average across the 20 repeats. A detailed description of this control analysis is now included in the updated manuscript (Methods, see lines 763 - 771) and the main figure in Figure 4D has been updated.

Lines 763 - 771 from the methods section: *“Given the sensitivity of factor analysis to sample size differences and the varied occupancy of internal states, it was essential to account for these differences (Mazzucato et al. 2016). To address this, we evaluated shared variance of single units while matching sample sizes across states for each subject. Specifically, we matched sample sizes across states by identifying the state with the lowest occupancy for each subject and using bootstrapping ($n = 20$ repeats) on the other two states to equalize their sample sizes. In each bootstrapped iteration, we determined the optimal number of components for factor analysis using 3-fold cross-validation of the log-likelihood estimate. The shared variance values reported in Figure 4D represent averages across all neurons, with each neuron’s shared variance calculated as the average over all bootstrapped repeats.”*

4. For the CV of ISI, I recommend an alternative way to control for different durations of the three HMM states: given three distribution of ISI with N_s samples (ISI) in each state s , pick the smallest N_s among the three and then randomly subsample the other distributions to obtain the same sample size, then perform statistical tests. This procedure avoids choosing an interval as in Fig. S6B and controls for the bias introduced by differences in sample size between conditions.

Controlling for state-wise differences in the distribution of inter-spike intervals (ISI) when evaluating the coefficient of variation (CV), we found that trends across states and areas remained consistent (Figure R1.3B). To achieve this, we first identified the state with the fewest ISI samples for each unit. We then randomly subsampled ISI values from the distributions of the other two states to match the total number of ISI samples. This process was repeated over 20 iterations, and the final CV for each unit was calculated as the average across these repeats.

While the previously reported trends remained consistent after this correction, there were minor differences (Figure R1.3A). Specifically, the CV in the high-frequency state increased significantly once sample size differences were accounted for. Mice tend to spend longer durations in the high-frequency state, resulting in more long-tailed ISI distributions (Figure 2F). Consequently, not fixing the range of ISI histograms (as was done previously to $\tau = 2.5s$) would increase the spread of the histogram in the high-frequency state, leading to a higher CV than initially reported. Although significant, this increase in CV is within the marginal change shown in Figure S6B after reaching a plateau. In the final revised version of the manuscript, both controls have been applied to our analysis to account for firing rate and dwell time differences across states. This analysis is now included in lines 778 - 785 and the main figure in Figure 4E has been updated.

Lines 778 - 785 from the methods section: *“To evaluate the coefficient of variation (CV) of individual neurons across different states (Figure 4E), we created histograms of their inter-spike-intervals (ISIs) based on the spike times observed within each state. Since large differences in dwell times across states could bias the ISI ranges and, consequently, the state-specific CVs, we fixed the range of the ISI histograms. We selected an interval of $t = 2.5s$, the point where the CV showed the largest incremental increase before plateauing (Figure S6B). In addition, we accounted for differences in firing rates across states by equalizing the firing rate of each unit to match its lowest rate observed across states. For each unit, the CV was then evaluated as an average over 20 repeats and the CV values reported in Figure 4E) represent the average across all single units.”*

Figure R1.3: **Coefficient of variation (CV) - Controlled for sample size differences across states: A** CV evaluated after equalizing the number of ISI samples across all states. CV is plotted along the visual hierarchy (quantified as anatomical hierarchy scores) and across oscillation states, averaged across all units (One-way ANOVA: $p_{S_H, S_I} = 4.5e-184$, $p_{S_H, S_L} = 2.9e-139$, $p_{S_I, S_L} = 4.1e-6$, $n = 7609$ units). **B** CV evaluated after equalizing the number of ISI samples and fixing the range of the ISI distribution in each state to $\tau = 2.5s$ ($p_{S_H, S_I} = 4.9e-23$, $p_{S_H, S_L} = 3.9e-3$, $p_{S_I, S_L} = 2.8e-11$, $n = 7609$ units).

5. Comparison of Fano factor between different conditions is very sensitive to firing rate and sample size (see above). Because of the large difference in both firing rate and sample size between different HMM states. First, in order to equalize firing rates the authors should proceed as follows (Papadopoulos et al, 2024, biorxiv). Given average firing rate r_s in HMM state s , pick the smallest r_s among the three states and randomly eliminate spikes from the others to equalize the firing rate. Then, to equalize sample sizes, we recommend eliminating bins from conditions with more bins until all conditions have the same number of bins to estimate the spike count variance.

Because I believe the sample size confound goes in the same direction as their trends in Fig. 4, the proposed controls are necessary to confirm the results.

Correcting for firing rate and sample size biases in the evaluation of the Fano factor (FF) revealed increased differences in FF values across states. For the control analysis, we first identified the state with the lowest occupancy and the state with the lowest firing rate. For each unit, we then employed bootstrapping (20 repeats) on the other two states to match their sample sizes to that of the state with the lowest occupancy. In each iteration of bootstrapping, we additionally dropped spikes to match the firing rate to that of the state with the lowest firing rate. We followed this order of processing, opposite to the one suggested by the reviewer, to reduce the variance in firing rate that might occur if sample sizes were matched after correcting for firing rate differences across states. The final FF for each unit was then calculated as the average across the 20 repeats. On applying this correction, FF values significantly decreased in the high-frequency and intermediate states. In the high-frequency state, the correlation between FF and anatomical hierarchy scores across visual areas remained significant ($S_H : r_{p-RL} = -0.94, p_{p-RL} = 0.02$). As previously reported, no such correlation was observed in the intermediate and low-frequency states. This analysis is now included in lines 796 - 800 and the main figure in Figure **4F** has been updated.

Lines 796 - 800 from the methods section: “ *To account for variations in firing rate and sample size across states, we used bootstrapping ($n = 20$ iterations) to equalize both firing rates and sample sizes. In each bootstrapping iteration, we first matched the sample sizes across states and then dropped spikes to equalize firing rates across states. The FF values reported in Figure **4F** are averaged across all units, with each unit’s FF calculated as an average over the bootstrapped repeats.*”

As a classical variability metric, the Fano factor (FF) is typically evaluated under strict experimental setups that control for fluctuations in the subject’s gaze (Kara et al. 2000; Bair 1999; Schölvinck et al. 2015). Additionally, it is well-established that the receptive field size of single units increases along the visual hierarchy, making eye movements a significant factor that could influence FF trends along the hierarchy. To account for these effects, we evaluated FF by clustering the mouse’s gaze into small, consistent bouts of fixed gaze and calculating the metric across trials within each cluster. Constraining the mouse’s gaze in this manner abolished the area-wise differences in the Fano factor across all states.

For each mouse, we applied hierarchical clustering on the mouse’s gaze on the screen to identify bouts of fixed gaze, dividing the gaze into five clusters. Given that mice do not have a fovea and the standard receptive field size in mouse primary visual areas is approximately $\sim 20^\circ$, this clustering ensured that the center of the mouse’s eye position remained within these constraints ($\sim 23.5 \pm 1.5^\circ$, mean \pm sem). Additionally, only neurons whose receptive field locations were positioned at least 20 degrees away from the monitor’s edge were included in the analysis. Each session was partitioned into states over time. Lastly, within non-overlapping windows of 150 ms where the mouse remained in the same state across at least 10 trials, we computed the Fano factor for each neuron and averaged it across time. Due to the constraints imposed, which reduce the number of sample points per state and per window, we refrained from replacing the results in the main manuscript. These results have now been included in the supplemental figure **S6G**, and the findings are explained in Lines 801 - 811.

Figure R1.4: **Fano factor (FF) control analyses - sample size, rate and eye gaze:** textbfA FF computed after equalizing sample size and firing rate across all states. FF is plotted along the visual hierarchy and across brain states, averaged across all units (One-way ANOVA: $p_{S_H, S_I} = 2.8e-4$, $p_{S_H, S_L} = 2.4e-33$, $p_{S_I, S_L} = 3.6e-32$, $n = 5017$ units). Pearson correlation with hierarchy scores excluding RL, S_H : $r_{p-RL} = -0.94$, $p_{p-RL} = 0.02$; S_I : $r_{p-RL} = -0.43$, $p_{p-RL} = 0.47$; S_L : $r_{p-RL} = -0.46$, $p_{p-RL} = 0.44$. **B** Example session showing mouse's gaze on screen colored by trial. **C** Hierarchical clustering applied to mouse's gaze to identify bouts of fixed gaze. **D** FF computed across trials within each cluster of fixed gaze. FF is plotted along the visual hierarchy and across brain states, averaged across all units (One-way ANOVA: $p_{S_H, S_I} = 1.1e-3$, $p_{S_H, S_L} = 5.6e-30$, $p_{S_I, S_L} = 1.9e-14$, $n = 5017$ units). Pearson correlation with hierarchy scores excluding RL, S_H : $r_{p-RL} = -0.55$, $p_{p-RL} = 0.33$; S_I : $r_{p-RL} = -0.28$, $p_{p-RL} = 0.65$; S_L : $r_{p-RL} = 0.01$, $p_{p-RL} = 0.99$.

Lines 801 - 811 from the methods section: *“A classical metric for assessing variability, the Fano factor (FF), is traditionally evaluated under controlled experimental conditions to manage fluctuations in the subject’s gaze (Kara et al. 2000; Bair 1999; Schölvinck et al. 2015). Additionally, as receptive field size increases along the visual hierarchy, eye movements can significantly influence FF trends. To address these effects (Figure S6E), we evaluated FF by clustering the mouse’s gaze into small, consistent bouts of fixed gaze and calculating the metric across trials within each cluster. For each mouse, we applied hierarchical clustering to the gaze data, dividing it into five clusters to identify bouts of stable gaze (Figure S6F). Given that mice lack a fovea and the standard receptive field size in their primary visual areas is approximately $\sim 20^\circ$, this clustering approach ensured that the center of the mouse’s eye position remained within these constraints ($\sim 23.5 \pm 1.5^\circ$, mean \pm SEM). We then partitioned each session by state and computed the Fano factor for each neuron, averaging it across time. This method of constraining the mouse’s gaze eliminated area-wise differences in the Fano factor across all states (Figure S6G).”*

6. Regarding the mutual information between neural activity and stimulus, its comparison between HMM states is affected by a critical confound that the authors must address. It is known that neural populations in V1 (Niell & Stryker, 2010) and in higher visual areas (Wyrick et al, J Neuro, 2021) encode more information about visual stimuli during running compared to resting periods. This effect was believed to be due to a decrease in noise correlations during running periods (Dadarlat et al, J Neuro, 20), and the results in Fig 4G are consistent with the literature. However, recent work (Wyrick et al, J Neuro, 2021) showed that the increase in stimulus-encoding information during running is entirely due to the firing rate increase induced by running. Once controlling for the difference in firing rate (by equalizing firing rates between conditions), the peak visual decoding performance (or in other words, the information per spike) is the same during running vs resting. Please perform a control analysis by equalizing the firing rates between the three HMM states and then re-estimating the MI, I expect the difference between conditions to go away.

We are grateful to the reviewer for highlighting this important issue. To address it, we performed a control analysis by equalizing firing rates across the three states using bootstrapping, and then re-estimated mutual information (MI). Despite this correction, the differences in MI between states remained significant (Figure R1.5A).

Believing this to be a consequence of using a non-linear measure of information like MI, we built two linear decoders using support vector machines (SVM) to predict the label of the most dominant object in the movie based on population spiking activity (Figure R1.5B). In both decoders, we equalized the sample size across states (i.e., number of input observations) and the number of units (i.e., number of input features) across areas. In the second decoder, we additionally equalized firing rates across states. As expected, the first decoder showed higher accuracy in the high-frequency state and significantly lower accuracy in the low-frequency state (Figure R1.5C). Interestingly, this trend persisted even after equalizing firing rates in the second decoder (Figure R1.5D). These results suggest that differences in stimulus influence across states are not solely due to variations in firing rates, but rather reflect distinct underlying neural dynamics across different brain states.

Figure R1.5: **State conditioned stimulus decoding** **A**. Mutual information (MI) computed after equalizing firing rate across all states. MI is plotted across the visual hierarchy and oscillation states averaged across all mice (Pairwise T-test: $p_{S_H, S_I} = 6.54e-04$, $p_{S_H, S_L} = 1.52e-11$, $p_{S_I, S_L} = 1.74e-04$, $n = 25$ mice). **B**. Image labels for linear decoder. **C**. Decoder accuracy computed after equalizing number of units across areas and sample sizes across states. Decoder accuracy is plotted across the visual hierarchy and oscillation states averaged across all mice (Pairwise T-test: $p_{S_H, S_I} = 1.54e-03$, $p_{S_H, S_L} = 2.68e-7$, $p_{S_I, S_L} = 3.25e-05$, $n = 25$). **D**. Decoder accuracy computed after equalizing firing rate across states along with number of units across areas and samples across states. Decoder accuracy is plotted across the visual hierarchy and oscillation states averaged across all mice (Pairwise T-test: $p_{S_H, S_I} = 9.55e-3$, $p_{S_H, S_L} = 1.58e-8$, $p_{S_I, S_L} = 8.69e-06$, $n = 25$).

7. For the bottom plots in Fig. 4D-E-F-G, please perform a 2-way anova with factors hierarchy score and HMM state and report p-values for the two factors and interaction term.

In the revised manuscript, we have now included the results from a two-way ANOVA analysis as performed against internal states and hierarchy score on each variability metric, and mutual information. We find significant effect of both factors on all the measures of variability reported in this paper. The results from the two-way ANOVA have now been included in the manuscript, please see the legend of Figure 4.

Lines from the legend of Figure 4: “...**D**, Population shared variance ... Percent shared variance plotted against the anatomical hierarchy scores of the visual areas in each oscillation state, averaged across all units (One-way ANOVA: $p_{S_H, S_I} = 9.6e-7$, $p_{S_H, S_L} = 1.3e-146$, $p_{S_I, S_L} = 1.2e-95$; Two-way ANOVA, states: $F = 431.2$, $p = 1.5e-189$, areas: $F = 78.8$, $p = 3.3e-82$, states \times area: $F = 3.3$, $p = 2.6e-4$, $n = 7609$ units). **E**, Neuronal variability across time, quantified using the coefficient of variation (CV) ... Bottom: CV along the visual hierarchy (quantified as anatomical hierarchy scores) and across oscillation states, averaged across all units (One-way ANOVA: $p_{S_H, S_I} = 4.9e-23$, $p_{S_H, S_L} = 3.9e-03$, $p_{S_I, S_L} = 2.8e-11$; Two-way ANOVA,

states: $F = 42.5, p = 3.6e - 19$, areas: $F = 88.1, p = 4.5e - 92$, states \times area: $F = 4.8, p = 4.9e - 7$, $n = 7609$ units). **F**, Neuronal variability across trials, quantified using Fano factor (FF) ... Bottom: FF along the visual hierarchy and across brain states, averaged across all units (One-way ANOVA: $p_{S_H, S_I} = 2.8e-4$, $p_{S_H, S_L} = 2.4-33$, $p_{S_I, S_L} = 3.5e-39$; Two-way ANOVA, states: $F = 107.7, p = 7.5e - 47$, areas: $F = 7.1, p = 9.9e - 6$, states \times area: $F = 0.6, p = 0.8$, $n = 5017$ units) ... **G**, Information encoding along the visual hierarchy across all oscillation states, quantified using mutual information (MI) ... Bottom: MI across the visual hierarchy and oscillation states averaged across all mice (Pairwise T-test: $p_{S_H, S_I} = 0.01$, $p_{S_H, S_L} = 7.3e-10$, $p_{S_I, S_L} = 9.3e-04$; Two-way ANOVA, states: $F = 3.1, p = 0.04$, areas: $F = 2.7, p = 0.03$, states \times area: $F = 0.02, p = 0.99$, $n = 25$) ..."

HMM - predictor model:

8. The "HMM based predictor model" is not itself an HMM, but rather, it is a set of 3 different GLMs, each one conditioned on the HMM internal states $S_{L,I,H}$ obtained in the previous section. Therefore, I recommend the authors rephrase the lines 218-228 and change notation to clarify this point, replacing "HMM based predictor model," "HMM-GLM" and similar notations throughout the paper with "Neural encoding model conditioned on internal states".

We appreciate the reviewer's suggestion regarding the naming of the model. We agree that the original name 'HMM-GLM' could be misinterpreted. In response, we have modified the naming of the model to be "Internal state-conditioned neural encoding model" throughout the manuscript.

9. What is missing here and in the previous section on neural variability is a direct estimation of how much variance of population activity and of single neuron activity can be explained directly and solely from the knowledge of the three brain states $S_{I,H,L}$. Namely, the authors estimate the cvR^2 of the various regressors, conditioned on the brain states, but don't explicitly measure the variance explained by knowledge of the three states. I believe this can be done directly using the ANCOVA or analysis of covariance that combines continuous and categorical regressors (the three brain states). Is there a reason why the authors did not perform this analysis? If not, I recommend they carry this out in the revised version for both the population averaged activity and for the single cell activity.

We thank the reviewer for highlighting this important control analysis. As suggested, we conducted two separate ANCOVA tests to investigate the effect of internal states on variability in both single-unit and population-level activity while controlling for variability induced by other factors (stimulus, internal brain activity, and behavior). Our results showed a significant effect of internal states in approximately 75% of the units within each visual area ($p < 0.05$), explaining, on average, $2.5 \pm 0.6\%$ ($N = 6398$ units with rate > 1 Hz) of the variability. At the population level, a significant effect of brain state was observed across all six brain areas in the majority of mice ($p < 0.05$), with internal states accounting for $5.9 \pm 1.1\%$ ($N = 25$ mice) of the variance in population activity. These results suggest that although state contributes consistently to neuronal variability, other factors are needed to explain the majority. We have now included the results from this analysis in the main manuscript (lines 248 - 253), with supporting figures presented in Supplementary Figure **S7A-D**.

Lines 248 - 253 from the results section: "Before quantifying state-specific variability from different sources, we assessed the effect of internal states on neuronal variability while controlling for other factors. We ran two separate ANCOVA tests to examine how internal states influenced variability in both single-unit and population-level activity (Figure **S7A-D**). Internal states significantly affected variability at both levels, with contributions of 3% for single-unit activity and 6% for averaged population activity, indicating that

additional factors are required to explain a majority of the observed variability."

Figure R1.6: **ANCOVA – Impact of internal state on spiking variability beyond the influence of other factors:** **A-B** Effect of internal state on averaged population activity. **A** Proportion of subjects showing significant state encoding ($p < 0.05$, ANCOVA). **B** Distribution of cross-validated explained variance (cvR^2) for averaged population activity, with internal states as the predictive feature. **C-D** Effect of internal state on single-neuron activity. **C** Proportion of units in each area significantly influenced by internal state ($p < 0.05$, ANCOVA). **D** Distribution of cross-validated explained variance (cvR^2) for single-neuron activity, using internal states as the predictive feature.

10. In Fig. 5 and 6, several analyses include comparison of the contribution to population activity in one area of the brain activity from other areas. Because area have different number of neurons per area (the authors mention this issue in the Discussion), this comparison can only be performed after equalizing the neuron number from other areas by subsampling neurons to the smallest common sample size. Please fix this control.

We agree that varying number of units across different areas could affect the final reported accuracy of both the population activity and single-unit models. However, we would like to clarify that the feature 'other area population activity' used here only includes the averaged activity across all units in a given area. Thus, the input to each regression model for a target area comprises the averaged population activity from other non-target areas. This difference has now been made more explicit in the manuscript in lines 235-236.

Lines 235-236 from the results section: “The averaged neuronal population activity represents the average activity across all units in a given area.”

Nevertheless, recognizing that these inconsistencies in the number of units could still impact the final variance explained, we conducted a control analysis to equalize the number of units included in the averages across all areas (Figure R1.7). Since population activity from non-target areas had minimal influence on single-cell variability and in the interest of time, we performed this control analysis only for the population model. At the population level, we found no significant difference in the explained variance of the models before and after equalizing the number of units across areas ($p = 0.63, N = 25$ mice, ANOVA). The methodology applied is detailed below.

Figure R1.7: **Explained variance: Control analysis after equalizing unit counts across areas** Comparison of cross-validated explained variance (cvR^2) before and after equalizing the number of units across areas in the evaluation of the input feature ‘other area population activity’.

For each subject, we first identified the non-target area with the minimum number of units. We then used bootstrapping ($n = 20$ repeats) on the other non-target areas to equalize the number of units that are included when evaluating the average population activity in each area. In each iteration of bootstrapping, these averages were included as inputs in the linear regression model to predict the population activity in the target area. The final explained variance was calculated as the average across the repeats. These final variances were not significantly different from the explained variances reported in the paper ($p = 0.63, N = 25$ mice, ANOVA).

11. line 308: "individual contributions from 308 different sources showed a reversed trend compared to the population model." This is a very strange result. How is that possible, since the population average activity is the average of the single neuron activity. The authors need to explain this contradiction in their results.

We thank the reviewer for this clarifying question. Upon further analysis, we found that the trends are not entirely reversed. Instead, stimulus features have a significant contribution at the level of individual units, while averaged population activity is more influenced by internal activity. This is likely because of the diversity of encoding at the single-cell level in the visual cortex (Figure 5G), wherein averaging single-unit activities can obscure individual differences and primarily retain only the shared variance across units. This

shared variance is largely captured by local field potentials and the averaged population activity from other visual areas (Figure 6A). The original line has now been modified (Lines 338 - 341).

Lines 338 - 341 from the results section: *“Interestingly, internal brain activity had the most predictive power ($cvR_I^2 = 41.0 \pm 7.6\%$, mean \pm std, $p = 2.5e-11$, $n = 25$ mice), higher even than the combined power of behavioral and stimulus features ($cvR_{B+S}^2 = 30.1 \pm 9.3\%$, mean \pm std, $p = 0.0005$, $n = 25$ mice). This was unlike single neuron activity, which was primarily driven by stimulus features.”*

12. When repeating the encoding model on other stimulus classes such as drifting grating do they get similar results? The natural images might have much sparser evoked responses, but the drifting grating should drive neurons enough to be able to create an encoding model.

Figure R1.8: Encoding results during viewings of drifting gratings: **A - C** Results from the population model. **A** Explained variance for different categories of input feature groups, averaged across all mice obtained using five-fold cross-validation. The box shows the first and third quartiles, the inner line is the median over 25 mice, and the whiskers represent the minimum and maximum values. **B** Average variance explained in averaged neuronal population activity in different visual areas. **C** Contributions from single category models to explaining the variance in averaged neuronal population activity in different states **D - F** Results from single neuron model **D** Mean explained variance for different categories of input features, averaged across $n = 4585$ units ($cvR^2 > 0.005$ and firing rate > 1 Hz) and obtained using five-fold cross-validation. The box shows the first and third quartiles, the inner line is the median over all neurons, and the whiskers represent the minimum and maximum values. **E** Average variance explained in single units in different visual areas. **F** Contributions from single category models to explaining single-neuron variability during different oscillation states.

To identify the stimulus-dependent aspects of our results, we conducted a similar analysis of neural responses in the visual cortex to drifting gratings. Our findings were largely consistent across viewings of

natural movie and drifting gratings (Figure R1.8). First, we identified oscillation states using LFPs recorded during the presentation of drifting gratings. Consistent with the LFPs recorded during natural movie viewings, we found three distinct oscillation states, with their identities and properties remaining consistent across both stimuli (Figure S4A).

Next, we constructed state-conditioned encoding models for each state identified during the drifting gratings viewing. Similar to our analysis on natural movies viewings, we developed two versions of the encoding model to examine neural variability at multiple scales: a population model (Figures R1.8A - C) and a single-neuron model (Figures R1.8D - F). However, to save time, the single-neuron model was implemented as a linear regression model rather than the Poisson regression used previously. In each state, we quantified the individual contributions of several factors to neural activity, including visual stimulus, behavior, and internal brain activity. For the stimulus features, we included three components: stimulus period, orientation, and contrast of the drifting gratings.

Our findings at both the population and single-neuron levels were consistent with those from the natural movie viewings. The only notable difference was a significantly higher contribution of internal activity to single-neuron variability in the high-frequency state (Figures R1.8C). We believe this heightened contribution of internal activity, which includes averaged population activity from non-target visual areas and LFPs from the target visual area, is due to an increase in gamma oscillations during stimulus presentation (Jia and Kohn 2011), which strongly correlates with neuronal variability in the high-frequency state. Since natural movies lack an on-off transition, this phenomenon is not observed.

Minor concerns:

13. line 33: Ashwood et al. 2022 does not analyze neural activity but just behavior.

We have now replaced it with the more relevant reference: Linderman et al., 2019, in the main manuscript (Line 33).

14. Fig. 2E: what are the shaded lines representing? Different areas and layer or different recording sessions?

In Figure 2E, the shaded lines represent the state-specific power distributions (z-scored) in individual mice. The solid NavyBlue line represent the average across all mice (N = 25). This description has now been added to the legend of Figure 2E.

Line from the legend of Figure 2E: "*E*, LFP power distribution in the three-state model. Shaded lines represent the state-specific z-scored power distributions in individual mice, and the solid NavyBlue line represent the average across all mice (N = 25 mice). In state-1, or the high-frequency state, LFPs are dominated by high-frequency gamma oscillations. State 3, or the low-frequency state, has characteristic slow oscillations in the theta band. "

15. The authors frequently mention "inputs to the HMM" but these actually refer to the observations. Input-driven HMMs are a different kind of model, so please adjust the notation to avoid confusion.

We have now modified the text to reflect this correction.

16. S4D, there's a 1 in the tpm, please replace with actual value with 3 digit precision.

The transition probabilities are now presented with higher precision in Figure S4D.

17. HMM - Behavior. The HMM fits to behavior and comparison with LFP HMMs in Fig. S5B are not explained, can you please elaborate on it? I don't see behavioral HMM in this figure, but only LFP HMMs. What correlation has been estimated in panel B exactly? No mention of the behavioral HMM is done anywhere, including the Methods section.

We separately quantified behavioral states from each measure—pupil size, face motion, and running speed—for each mouse. To obtain these states, we fit each measure to a three-state HMM. To identify the behavioral measure that best reflects the brain's internal state (quantified using LFPs), we evaluated the Pearson correlation between each sequence of behavioral states and internal states. We found stronger correlations between internal states and behavioral states derived from pupil size and face motion compared to running speed (Figure S5B; $p = 0.0007$, $n = 25$ mice). We have now incorporated a detailed explanation of the quantification of behavioral states in the methods section (Lines 731 - 738).

Lines 731 - 738 from the methods section: *“Lastly, to identify the behavioral features that could generate states most aligned with the internal brain states, we applied Hidden Markov Models (HMMs) to separately infer behavioral states from the mouse's running speed, pupil size, and face motion. Each HMM was modeled with the same number of states as the internal brain states ($M^* = 3$), ensuring consistency in state comparison. For each feature, we constructed behavioral state sequences by fitting an HMM using Gaussian emissions to model the continuous data. The transition probabilities were learned from the data, using the Expectation-Maximization (EM) algorithm for maximum likelihood estimation. After deriving the behavioral state sequences for each feature, we evaluated the Pearson correlation coefficient between each behavioral state sequence and the internal state sequence derived from cortical LFPs (Figure S5B).”*

18. line 171: The following interpretation needs clarification: "Thus, we attribute the observed decreasing trends to rapid variations in luminance or moving edges in the natural movie, that likely induce stronger temporally coherent activity within a population in lower visual areas than in higher visual areas."

Our findings show that the variance shared across single units in a population decreases along the visual anatomical hierarchy. Neurons in early visual areas are more strongly modulated by the temporal features of visual stimuli. As a result, these neurons tend to covary with the temporal dynamics of the stimulus, leading to greater shared variance within the population. As one moves higher along the anatomical hierarchy, neurons become more functionally diverse (Figure 5H), responding differentially to a broader range of environmental features. This clarification is included in the main manuscript at lines 171-174.

Lines 171-174 from the methods section: *“Thus, we attribute the observed decreasing trends across the visual hierarchy to the stronger modulation of neurons in lower visual areas by the temporal features of the natural movie, such as rapid variations in luminance or moving edges. This likely induces more temporally coherent activity within populations in the lower visual areas compared to higher visual areas, resulting in greater shared variance.”*

19. Internal states measured with pupil, whisking and running have been recently shown to predict task performance states in Hulsey et al, Cell reports, 2024; please add discussion.

These findings have now been cited in lines 552 - 555.

Lines 552 - 555 from the discussion section: *“Recent studies have shown that a subject’s engagement during an active task varies drastically from trial to trial, playing out through multiple interleaved strategies (Ashwood et al. 2022; Zhuang et al. 2021; Piet et al. 2023), where other work has shown that changes in these strategies can be predicted by the animal’s arousal levels, suggesting a direct link between brain states and task performance (Hulsey et al. 2024; McGinley et al. 2015).”*

20. Fig. 4E: please add legend for color codes.

A legend has now been added.

21. Fig. 5D, is this the performance of the average between the three models obtained from the three different HMM states $S_{I,L,H}$? Please clarify.

Each final prediction shown in Figure 5D is created by combining the individual predictions of activity in each state made by the three separate state-conditioned generalized linear models (GLM). That is, the final prediction shown is the result of concatenating the outputs from these state-specific models to form a single, continuous prediction. This has now been clarified in the legend of Figure 6B.

Line from the legend of Figure 6B: *“B, Averaged population responses overlaid with model predictions from respective input feature groups. The prediction traces were generated by concatenating the outputs from three state-conditioned GLMs into a single, continuous prediction.”*

22. Fig. 6F: "variability" typo

This figure has now been removed from the paper.

Discussion:

23. Regarding the fact that "Recent studies have shown that a subject’s engagement during an active task varies drastically from trial to trial, playing out through multiple interleaved strategies", it’s worth mentioning that recent work in Hulsey et al, 2024 showed that such changes in strategy are predicted by behavioral variables (pupil, whisking, running) suggesting a direct relationship between the findings in the current manuscript and task performance.

These findings have now been cited in lines 552 - 555.

Lines 552 - 555 from the discussion section: *“Recent studies have shown that a subject’s engagement during an active task varies drastically from trial to trial, playing out through multiple interleaved strategies (Ashwood et al. 2022; Zhuang et al. 2021; Piet et al. 2023), where other work has shown that changes in these strategies can be predicted by the animal’s arousal levels, suggesting a direct link between brain states and task performance (Hulsey et al. 2024; McGinley et al. 2015).”*

24. The authors should add a paragraph on what the neural mechanisms might mediate the relationship between state-dependence, variability and stimulus coding. One possible explanation is based on the fact that changes in brain state, captured by the three HMM states, modulate the intrinsic metastable activity of the cortical circuit. This mechanism explained the interaction between changes in brain states, neural variability and stimulus coding in gustatory cortex (Mazzucato et al, Nat Neuro, 2019), hierarchically connected visual areas (Wyrick et al, J Neuro, 2021) and auditory cortex (Papadopoulos et al, 2024); and primate V4 (Shi et

al, Nat Comm, 2022; Zeraati et al, Nat Comm, 2023).

We thank the reviewer for this interesting direction of discussion. We have now included a paragraph discussing the recently published mechanisms, tying them up with our current findings. Please see lines 531 - 545.

Lines 531 - 545 from the discussion section: *“Several studies have demonstrated that the structural connectivity of neural networks directly influences neural dynamics (Engel and Steinmetz 2019). Theoretical studies on biologically plausible models show that neural computations are guided by the interplay between recurrence and dynamic changes in dimensionality, enabling flexible computations across different tasks (Dahmen et al. 2022; Dubreuil et al. 2022). This dynamic flexibility in local circuits has been shown to be closely linked to the emergence of metastable states, reflecting transient periods of stable neural activity before the network transitions to new patterns. These metastable states have provided a valuable framework for studying how structural connectivity relates to noise correlations, which in turn influences the state-dependent processing of sensory information (Wyrick and Mazzucato 2021; Mazzucato et al. 2019; Papadopoulos et al. 2024; Shi et al. 2022; Zeraati et al. 2023). While these studies have revealed important neural mechanisms, they are yet to explore the state-dependent changes in local circuit dynamics, which could represent a more global mechanism underlying the variability observed across various brain areas. To address this gap, our future work will investigate the state-specific functional organization of the cortical circuits and how they adapt to different internal and external stimuli, with a particular focus on the mechanisms that drive state transitions and their impact on sensory processing.”*

Reviewer #2: In this paper, Akella et al. perform a comprehensive analysis of a widely used dataset in visual neuroscience. They focus on the question of how different sources dynamically contribute to the non-stationarity of neuronal variability in the mouse visual system. There are a lot of analyses and results reported in the manuscript.

1. There is not a single key take-away, other than that spiking activity in the mouse visual system is driven by visual stimuli, brain states, and behaviors that are not directly linked to brain states. The paper will be interesting for other scientists working with this data set, but less so for a broad audience.

We thank the reviewer for this general comment. We recognize that our previous writing style did not effectively emphasize the key points of the paper. To address this, we have reorganized the results section, presenting section 6 on single-neuron variability before section 5 on population-level variability, as well as expanding several sections to highlight the key-findings (see the detailed changes below).

In this paper, we have focused on understanding the relative contribution of internal and external factors to neuronal variability, how that evolves over time and its impact on information processing. To rigorously quantify the contribution from different factors, we have incorporated several sources of variability and used brain states as a temporal framework. Below, we have highlighted three key findings from our analysis:

1. We demonstrate that neural variability is dynamically shaped by multiple factors, and importantly, that the combinations of contributing factors fluctuates spontaneously and rapidly in a state-dependent manner. (Figures **4D-G**, **5E**, **I**, **6D**).
2. We show that the relative contribution from different sources is not uniform across neurons. Instead, individual neurons have diverse patterns of influence, with various combinations of factors affecting different sub-populations. For instance, single units become increasingly heterogeneous along the hierarchy, reflecting more influences from non-visual factors on the variability patterns of individual neurons (Figure **4D-G** **5G-H**).
3. Surprisingly, we found that the dominant factors influencing population activity differ from those driving single neurons. While single neurons are primarily influenced by stimulus features, population activity is largely dominated by internal brain activity. These results show, contrary to recent findings (Musall et al. 2019; Stringer et al. 2019), that movements were not the dominant influencing factor (Figure **5C**, **D**, **6A,B**).

We have clarified these main points in the abstract, in the last paragraph of the introduction (lines 44-54), and in the discussion sections (lines 392 - 404, and 456 - 474) to make these take-away messages more explicit. These lines have been reproduced below for your convenience.

Lines from the abstract: *“Regression models within each state revealed a dynamic composition of factors contributing to the observed spiking variability, with the primary influencing factor switching within seconds. During the state dominated by high-frequency oscillations, sensory inputs exerted the most influence on individual neurons, while internal brain activity was the most predictive of population dynamics. The state-conditioned regression model further uncovered extensive diversity in source contributions across units in the visual cortex that varied in accordance with anatomical hierarchy and internal state. This heterogeneity in encoding across states underscores the importance of partitioning variability over time, particularly when considering the dynamic influence of non-stationary factors on sensory processing.”*

Lines 44-54 from the introduction section: *“Quantifying various aspects of variability across individual trials and neuronal populations, we uncovered significant changes in neuronal variability across states. These findings indicated dynamic shifts in the efficiency of sensory processing over time. To disentangle the sources of non-stationary sensory processing, we designed a novel neural encoding framework conditioned on internal states to partition variability across three crucial factors: internal brain dynamics, spontaneous behavior, and external visual stimuli. Through this model, we quantified the time-varying contributions of these sources to single-trial neuronal and population dynamics. Our findings revealed that, even during persistent sensory drive, neurons dramatically changed the degree to which they were impacted by sensory and non-sensory factors within seconds. Additionally, we observed considerable diversity in neural encoding across visual cortical units, with the relative influence of these sources varying based on their anatomical location and cell type. Taken together, our results provide compelling evidence for the dynamic nature of sensory processing, while emphasizing the role of latent internal states as a dynamic backbone of neural coding.”*

Lines 392 - 404 from the discussion section: *“Using the state fluctuations as a temporal backbone, we constructed a state-based encoding model to partition and evaluate the relative contributions from three different sources of variability to visual cortical activity: visual stimulus, behavior, and internal brain dynamics. The model accounted for 27% of single-neuron variability and 53% of the variance in averaged population activity. Neurons in the visual cortex are influenced by a diverse array of factors, with the relative contributions of these factors differing across sub-populations. Firstly, the combination of factors affecting variability changes spontaneously and rapidly over time in a state-dependent manner. Secondly, the contributions of each source are further influenced by cell type and anatomical location, becoming increasingly heterogeneous as one ascends the hierarchy. Lastly, while single neurons in the visual cortex are primarily affected by stimulus features, population activity is largely dominated by internal brain activity. Overall, our study underscores the importance of accounting for the constantly changing contributions from internal and external factors on stimulus representation at the level of individual units, enabling a deeper understanding of how neural responses are dynamically shaped in real time.”*

Lines 456 - 474 from the discussion section: *“Neurons in the visual cortex can be classified by several criteria, including their morphology, connectivity, developmental history, gene expression, intrinsic physiology, and in vivo encoding strategies. Single-cell RNA sequencing studies have revealed extensive cell-type diversity and their relationships within cortical circuits (Gouwens et al. 2020; Saunders et al. 2018). Different cell types have distinct functional roles, which are further influenced by their position within the cortical hierarchy and the specific inputs they receive across different layers (Pfeffer et al. 2013; Garrett et al. 2023; Ma et al. 2024; Kamigaki 2019; Olsen et al. 2012; Senzai et al. 2019). Furthermore, different neuronal types are modulated by various factors such as behavior (Musall et al. 2019; Stringer et al. 2019), top-down feedback (Garrett et al. 2023), and internal brain states (Ferguson et al. 2023). Through unsupervised clustering of each neuron’s encoding patterns, we quantified their encoding diversity, uncovering units with specialized properties within the visual cortex (Figure 5G). Our findings indicate that neurons in the visual cortex are modulated by a diverse array of factors, with the relative contributions of these factors varying across states, hierarchical positions, layers and cell-types (Figures 5H,I, S9C, E). Additionally, we observed an increasing representation of pan-modulated units along the visual hierarchy (Figure 5H), suggesting that while integrative processes may start as early as V1, a larger network of neurons becomes involved in this process higher up the hierarchy. In line with recent studies suggesting that sensory-motor integration begins in early sensory areas (Stringer et al. 2019; Erisken et al. 2014; Saleem et al. 2013), we identified two distinct neuronal clusters likely involved in this process : one driven solely by behavior, and another influenced by both visual stimuli and behavioral factors (Figure 5G). These findings emphasize the*

complex and dynamic nature of visual processing, shaped by a multitude of internal and external factors."

Additionally, we realized that the previous arrangement of figures, where the most important result was presented last, might have obscured our key findings. To rectify this, we have reorganized the sequence of the figures to better highlight this critical finding. The section on single-neuron variability has been moved up in the manuscript (see lines 254 - 329), followed by the introduction of the population model (see lines 331 - 386) as a control to address earlier studies that reported substantial contributions of behavior to emergent neuronal variability (Musall et al. 2019; Stringer et al. 2019). We have revised the sequence and logic of the figures to ensure that the main take-away is presented more clearly and earlier (lines 331-334).

Lines 331-334 from the results section: *"Recent studies (Musall et al. 2019; Stringer et al. 2019) have reported significant contribution of spontaneous movements in the emergent properties of brain-wide activity. To examine these effects in population dynamics within the visual cortex in a state-specific manner, we constructed a state-dependent linear regression model to predict the averaged neuronal population activity in each of the six visual areas."*

2. In many places, there is an uncomfortable mismatch between the historical logic underlying the analysis of response variability and the realities of this type of experiment. In general, these analyses provide an estimate of the ‘noisiness’ of neural activity if you manage to measure this activity under repeated presentations of exactly the same experimental condition. That standard is absolutely not met here. There is likely trial-to-trial variability in the position of the eyes, resulting in stimulus-variability across repeated trials.

We acknowledge that, in general, experimental conditions in mouse visual studies are not as tightly controlled across trials as in classical non-human primate research, where subjects are either anesthetized or their gaze is restricted within a small fixation window. To maintain the rigor of classical variability metrics, we implemented several controls to test how variability changes across gaze, states, and brain areas. First, we controlled for differences in firing rates across states while evaluating the variability metrics, which further confirmed our findings. Second, we accounted for variability in eye position by calculating the metrics by clustering the mouse’s gaze into small, consistent bouts of fixed gaze and then calculating the metric across trials within each cluster. Constraining the mouse’s gaze in this manner abolished the area-wise differences in the Fano factor across all states, but did not affect the hierarchical trends for coefficient of variation. The specific details of each method are listed below:

1. Control for firing rate:

a. Fano factor (Figure R2.1A): Correcting for rate and sample size biases in the evaluation of the Fano factor (FF), revealed increased differences in their values across states. For this, we first identified the state with the lowest occupancy and the state with the lowest firing rate. For each unit, we then employed bootstrapping (20 repeats) on the other two states to match their sample sizes to that of the state with the lowest occupancy. In each iteration of bootstrapping, we additionally dropped spikes to match the firing rate to that of the state with the lowest firing rate. The final FF for each unit was then calculated as the average across the 20 repeats. A detailed description of this control analysis is now included in the updated manuscript (lines 796 - 800, Figure 4F).

b. Coefficient of variation (Figure R2.1B): To implement this control, we first identified the state with the lowest firing rate for each unit. To match the firing rate of the unit across states, we randomly sub-sampled values from the distribution of inter-spike-intervals (ISI) of the other

two states. Additionally, to account for the dwell time differences across states, the range of each state-specific ISI histogram was fixed to 2.5 secs. The sub-sampling process was repeated over 20 iterations and the final coefficient of variation (CV) for each unit was calculated as the average across these repeats. A detailed description of this control analysis is now included in the updated manuscript (lines 778 - 785, Figure 4E).

Figure R2.1: **Control for firing rate:** **A** FF computed after equalizing sample size and firing rate across all states. FF is plotted along the visual hierarchy and across brain states, averaged across all units (One-way ANOVA: $p_{S_H, S_I} = 2.8e-4$, $p_{S_H, S_L} = 2.4e-33$, $p_{S_I, S_L} = 3.6e-32$, $n = 5017$ units). Pearson correlation with hierarchy scores excluding RL, S_H : $r_{p-RL} = -0.94$, $p_{p-RL} = 0.02$; S_I : $r_{p-RL} = -0.43$, $p_{p-RL} = 0.47$; S_L : $r_{p-RL} = -0.46$, $p_{p-RL} = 0.44$. **B** CV evaluated after equalizing the firing rate and fixing the range of the ISI distribution in each state to $\tau = 2.5s$ (One-way ANOVA: $p_{S_H, S_I} = 4.9e-23$, $p_{S_H, S_L} = 3.9e-03$, $p_{S_I, S_L} = 2.8e-11$, $n = 7609$ units).

2. Control for variability in eye position: To control for fluctuations in the subject's gaze (Figure R2.2A), we applied hierarchical clustering on the mouse's gaze on the screen to identify bouts of fixed gaze, dividing the gaze into five clusters (Figure R2.2B). Given that mice do not have a fovea and the standard receptive

field size in mouse primary visual areas is approximately $\sim 20^\circ$, this clustering ensured that the center of the mouse's eye position remained within these constraints ($\sim 23.5 \pm 1.5^\circ$, mean \pm sem).

a. Fano Factor (Figure R2.2C): Only neurons whose receptive field locations were positioned at least 20 degrees away from the monitor's edge were included in the analysis. Each session was further partitioned into states over time. Lastly, within non-overlapping windows of 150 ms where the mouse remained in the same state across at least 10 trials in each cluster, we computed the Fano factor for each neuron and averaged it across time. Due to the constraints imposed, which reduce the number of sample points per state and per window, we refrained from replacing the results in the main manuscript. However, these results have now been included in the supplemental figure (Figure S6E-G), and the findings are explained in the results section (Lines 801 - 811).

b. Coefficient of variation (Figure R2.2D):: By dividing each session into distinct states and clusters of consistent gaze bouts, we evaluated the CV for each neuron in each state, averaging across the different clusters. While we observed an overall reduction in CV values, the trends across states and visual areas remained consistent.

For your convenience, we have provided the relevant lines from the methods section (Lines 796 - 800, 801 - 811, and 778 - 785) corresponding to each control analysis below.

Fano factor - control for firing rate: Lines 796 - 800 from the methods section: "*To account for variations in firing rate and sample size across states, we used bootstrapping ($n = 20$ iterations) to equalize both firing rates and sample sizes. In each bootstrapping iteration, we first matched the sample sizes across states and then dropped spikes to equalize firing rates across states. The FF values reported in Figure 4F are averaged across all units, with each unit's FF calculated as an average over the bootstrapped repeats.*"

Fano factor - control for eye gaze: Lines 801 - 811 from the methods section: "*A classical metric for assessing variability, the Fano factor (FF), is traditionally evaluated under controlled experimental conditions to manage fluctuations in the subject's gaze (Kara et al. 2000; Bair 1999; Schölvinck et al. 2015). Additionally, as receptive field size increases along the visual hierarchy, eye movements can significantly influence FF trends. To address these effects (Figure S6E), we evaluated FF by clustering the mouse's gaze into small, consistent bouts of fixed gaze and calculating the metric across trials within each cluster. For each mouse, we applied hierarchical clustering to the gaze data, dividing it into five clusters to identify bouts of stable gaze (Figure S6F). Given that mice lack a fovea and the standard receptive field size in their primary visual areas is approximately $\sim 20^\circ$, this clustering approach ensured that the center of the mouse's eye position remained within these constraints ($\sim 23.5 \pm 1.5^\circ$, mean \pm SEM). We then partitioned each session by state and computed the Fano factor for each neuron, averaging it across time. This method of constraining the mouse's gaze eliminated area-wise differences in the Fano factor across all states (Figure S6G).*"

Figure R2.2: **Control for eye gaze:** **A** Example session showing mouse's gaze on screen colored by trial. **B** Hierarchical clustering applied to mouse's gaze to identify bouts of fixed gaze. **C** FF computed across trials within each cluster of fixed gaze. FF is plotted along the visual hierarchy and across brain states, averaged across all units (One-way ANOVA: $p_{S_H, S_I} = 1.1e-03$, $p_{S_H, S_L} = 5.6e-30$, $p_{S_I, S_L} = 1.9e-14$, $n = 5017$ units). Pearson correlation with hierarchy scores excluding RL, S_H : $r_{p-RL} = -0.55$, $p_{p-RL} = 0.33$; S_I : $r_{p-RL} = -0.28$, $p_{p-RL} = 0.65$; S_L : $r_{p-RL} = 0.01$, $p_{p-RL} = 0.99$. **D** CV computed across trials within each cluster of fixed gaze, plotted along the visual hierarchy and across brain states, and averaged across all units (One-way ANOVA: $p_{S_H, S_I} = 4.1e-40$, $p_{S_H, S_L} = 2.2e-44$, $p_{S_I, S_L} = 1$, $n = 7042$ units). Pearson correlation with hierarchy scores excluding RL, S_H : $r_{p-RL} = -0.96$, $p_{p-RL} = 0.002$; S_I : $r_{p-RL} = -0.97$, $p_{p-RL} = 0.001$; S_L : $r_{p-RL} = -0.86$, $p_{p-RL} = 0.02$.

Coefficient of variation - control for firing rate: Lines 778 - 785 from the methods section: “To evaluate the coefficient of variation (CV) of individual neurons across different states (Figure 4E), we created histograms of their inter-spike-intervals (ISIs) based on the spike times observed within each state. Since large differences in dwell times across states could bias the ISI ranges and, consequently, the state-specific CVs, we fixed the range of the ISI histograms. We selected an interval of $t = 2.5s$, the point where the CV showed the largest incremental increase before plateauing (Figure S6B). In addition, we accounted for differences in firing rates across states by equalizing the firing rate of each unit to match its lowest rate observed across states. For each unit, the CV was then evaluated as an average over 20 repeats and the CV values reported in Figure 4E) represent the average across all single units. ”

3. There is clearly a lot of cross-trial variation in the animals’ behavior (running at various speeds) and brain state (dozing off or being wakeful). These factors are known to systematically impact neural activity in visual cortex. As a consequence, it is unclear what statistics like the Fano factor or coefficient of variability try to measure here. As an example, V1 Fano factors were a bit higher than those in higher visual areas. Might that just reflect that V1 is more heavily modulated by running speed and this variable was not controlled for? What is the point of knowing a Fano factor for a set of trials that effectively come from different experimental conditions?

As the reviewer has pointed out, it is well-known that behavior and brain states significantly influence neuronal activity. In this study, we account for the effects of trial-to-trial variations in experimental and internal factors on neuronal variability by conditioning our analyses using classical variability metrics, such as the Fano factor and coefficient of variation, on continuously changing internal brain states. These states, estimated based on internal oscillations, show significant correlations with behaviors like running, facial motion etc, thereby accounting for the effects of behavioral changes (Figures 3E, S5B). Further, to specifically understand how different sources affect variability within each state, we applied a regression model to directly quantify the relative contribution of each factor to neuronal activity within each state, a key result of our study. (Figures 5, 6).

Figure R2.3: **Effect of behavior on neuronal variability** **A** Distribution of neuronal clusters across areas, derived through unsupervised clustering based on the encoding patterns of single neurons. **B** Contributions of behavioral features (running, pupil size, facial motion, and body part movements) to explain variance in average neuronal population activity across different visual areas.

With regard to the example of Fano-factor mentioned by the reviewer, we’d like to make two clarifications.

Firstly, we quantified and compared Fano factor across two dimensions: internal states and visual areas (Figure 4E,F). Secondly, because the effect of behavior modulation is equivalent across the simultaneously-recorded visual areas (Figure 5H, 6I, reproduced here as R2.3A, B, respectively), we believe that the higher Fano factor values observed in V1, relative to other visual areas, are likely due to eye movements rather than behavioral effects. As detailed in our response to question (2), controlling for eye gaze eliminated area-wise differences in the Fano factor (Line number of corresponding changes in the manuscript are listed here: Lines 796 - 800, 801 - 811, 778 - 785, 763 - 771, Figure R2.2). Importantly, after controlling for these factors, differences across states remained significant for both the Fano factor and coefficient of variation (Figure R2.1, R2.2).

4. Some of the findings seem a bit circular to me. For example, one measure of population activity (LFP) turns out to be a good predictor for a different measure of population activity (population rate). I believe that. I am not sure that it reveals a deeper insight.

It is well-established that population activity correlates with specific frequency bands of local field potentials (LFPs) and with population activity in other brain areas. In our study, we included internal brain activity as a factor in the encoding model to rigorously partition sources of neuronal variability. This approach builds on previous research exploring the relationships among LFPs, population activity, and single-neuron activity (Ahmadi et al., 2021; Herreras, 2016; Gallego-Carracedo et al., 2022).

The population neural encoding model was included as a control to address findings from earlier studies that reported substantial contributions of behavior to emergent neuronal variability (see lines 331 - 334, Musall et al. 2019; Stringer et al. 2019). To appropriately assign contributions to each factor, we incorporated all the factors used in the single-neuron encoding model. While our findings on the relationship between LFP and population activity with single-neuron activity are consistent with prior research, we additionally found that when partitioning variability across states, internal brain activity has a particularly dominant influence in the low-frequency state, despite being highly correlated with other population measures throughout (lines 338 - 341, 351 - 367, Figure 6C, D).

Lines 331 - 334 from the results section: *“Recent studies (Musall et al. 2019; Stringer et al. 2019) have reported significant contribution of spontaneous movements in the emergent properties of brain-wide activity. To examine these effects in population dynamics within the visual cortex in a state-specific manner, we constructed a state-dependent linear regression model to predict the averaged neuronal population activity in each of the six visual areas.”*

Lines 338 - 341 from the results section (logic to inclusion as control): *“Interestingly, internal brain activity had the most predictive power ($cvR_I^2 = 41.0 \pm 7.6\%$, mean \pm std, $p = 2.5e-11$, $n = 25$ mice), higher even than the combined power of behavioral and stimulus features ($cvR_{B+S}^2 = 30.1 \pm 9.3\%$, mean \pm std, $p = 0.0005$, $n = 25$ mice). This was unlike single neuron activity, which was primarily driven by stimulus features.”*

Lines 351 - 367 from the results section (results on internal activity contributions): *“The addition of internal brain activity to the combined model of behavioral and stimulus features increased the explained variance by almost 24% ($\Delta r_{F-(B+S)}^2 = 23.5 \pm 10.2\%$, mean \pm std, 6A). Considering that LFP and population activity inherently carry information about stimulus and behavioral features, potentially making part of their contributions redundant, we have deliberately orthogonalized these internal variables against the stimulus and behavior variables (Mante et al. 2013). This orthogonalization ensures that internal variables capture variance beyond what can be accounted for by stimulus and behavior variables alone.*

To understand the substantial increase in explained variance, we analyzed the contributions of internal

brain activity to each state. We found that these variables largely increased the predictability during the low-frequency state ($\Delta r_{S_L, F-(B+S)}^2 = 39.0 \pm 15.8\%$, mean \pm std, Figure 6C, left panel). Activity in this state was poorly explained by the combined model of stimulus and behavioral features ($cvR_{S_L, (B+S)}^2 = 16.2 \pm 11.5\%$, mean \pm std, $p = 8.3e-6$, $n = 25$ mice). The combined model of stimulus and behavioral features was best at explaining variability in the high-frequency state, and accordingly, activity in this state showed a smaller improvement in its predictability on the inclusion of internal activity features ($\Delta r_{S_H, F-(B+S)}^2 = 14.3 \pm 4.6\%$, mean \pm std, $p = 2.3e-6$, $n = 25$ mice). Consistently, within-area LFPs and averaged population activity from the neighboring visual areas contributed more towards explaining the activity in the low-frequency state ($p = 4.5e-6$, $p = 8.2e-13$, respectively; $n = 25$ mice, Figure 6D)."

5. I found the most interesting finding to be at the end of the manuscript (Fig 6). This analysis pedals back some popular claims that behavior is the main driver of response variability in mouse V1. Apparently, this is not true if you look at single neuron spiking activity as opposed to (global) calcium measurements. I think this finding could be turned into a future paper that is interesting to a broader readership.

We appreciate the reviewer for recognizing the importance of this figure, which represents a central focus and key finding of our study. Upon review, we realized that the previous organization of the manuscript unintentionally downplayed the significance of these results. To rectify this, we have reorganized the sequence of the figures to better highlight this critical finding. The section on single-neuron variability has been moved up in the manuscript (see lines 255 - 329), followed by the introduction of the population model (see lines 331 - 386) as a control to address earlier studies that reported substantial contributions of behavior to emergent neuronal variability (Musall et al. 2019; Stringer et al. 2019).

We also added new analysis to demonstrate the heterogeneity of factors influencing individual neuron activity (Methods, Lines 967 - 991, Results, 290 - 329, Discussion, 456 - 474). Using unsupervised clustering on each neuron's encoding pattern, we identified six distinct neuronal clusters in the visual cortex (Figures 5G, S9B, R2.4A, B): one dominated by stimulus, one by behavior, another with high encoding of both stimulus and behavior, a group influenced by both stimulus and averaged population activity from neighboring visual areas, a group with high explained variance across all input feature categories (multi-source), and a final group comprising neurons where no single feature explained more than 10% of their variance. Our findings indicate that neurons in the visual cortex are modulated by a diverse array of factors, with the relative contributions of these factors varying across states, hierarchical positions, layers and cell-types (Figures 5H, I, S9C, E, R2.4C-H). We also observed an increasing representation of pan-modulated units along the visual hierarchy (Figures 5H, R2.4D), suggesting that while integrative processes may start as early as V1, a larger network of neurons becomes involved in this process higher up the hierarchy.

For your reference, we have copied the specific additions made to the main manuscript that correspond to this new clustering analysis. These include lines 967 - 991 from the Methods section, lines 290 - 329 from the Results section, and lines 456 - 474 from the Discussion section.

Figure R2.4: **Heterogeneity of factors influencing visual cortical activity** **A** Clustering methods applied to categorize single units into distinct functional clusters based on the dominantly contributing features. **B** Cluster centers corresponding to each identified cluster. Both stimulus-driven clusters (stimulus1 and stimulus2) and multi-source (multi-source1 and multi-source2) clusters were combined for all subsequent analyses. **C** Neuronal clusters identified through unsupervised clustering of the final explained variance from single-category models for all units. **E**, Distribution of neuronal clusters distribution across areas. **E**, Neuronal clusters derived from unsupervised clustering of state-specific explained variance from single-category models for all units, showing how feature encoding dynamics shift across different oscillatory states. **F** Layer-wise distribution of units within different functional clusters. **G** Ratio of regular spiking to fast-spiking cells in each cluster. **H** Mean receptive field area of units within each cluster. **I** (left to right) Variability across trials (Fano factor), variability across time (coefficient of variation), and shared variability of neurons within each cluster.

Lines 967 - 991 from the methods section: “To identify neuronal clusters with similar source contributions, we performed clustering on their encoding patterns. The feature coding for each neuron was represented by a five-element vector, which comprised the cross-validated explained variance from each source category: stimulus, behavior, local field potentials (LFPs) from the target area, averaged population activity

from other visual areas, and the number of source categories with more than 10% explanatory power. All single units that passed the selection criteria for inclusion in the single-neuron model were included in this clustering analysis.

We implemented a clustering workflow that used mean-shift clustering in conjunction with a consensus clustering method (Monti et al. 2003) to reduce sensitivity to random initial conditions and hyperparameter selection (Figure S9A, left). To minimize noise, we applied UMAP (Uniform Manifold Approximation and Projection, McInnes et al. 2018), a manifold-based dimensionality reduction technique, to project the feature matrix into a lower-dimensional space.

To achieve stable co-clustering results, we first built a co-clustering association matrix by performing 200 repeats of mean-shift clustering on the UMAP projections. Each entry in the co-clustering association matrix represents the probability of two units belonging to the same cluster (Figure 9A, top-right). Since UMAP projections are highly sensitive to hyperparameter selection, we randomized these hyperparameters in each iteration ($\text{min_dist} \in [0.02, 0.5]$, $n_neighbors \in [5, 10]$, $n_components \in [2, 5]$). Overall, each iteration included a random initialization of the mean-shift clustering model as well as a random dimensionality reduction.

Next, we clustered the association matrix with hierarchical clustering to determine the cluster labels (Figure 9A, center-right). The optimal number of clusters was determined by evaluating the silhouette scores on a range of clusters obtained via hierarchical clustering (Figure 9A, bottom-right), and selecting the point of maximum curvature as the optimal number of clusters (Satopaa et al. 2011). Clustering based on the final explained variance of each unit resulted in 8 clusters (Figure 9B). We then manually combined clusters with dominant explained variances in the same source categories. Finally, using the same clustering workflow, we also clustered the units based on their state-specific explained variance values (Figure 9J, K)."

Lines 290 - 329 from the results section: "The influence of different sources was not uniform across neurons; rather, individual neurons appeared to be driven by a diverse array of factors (Figure 5F), suggesting heterogeneous coding mechanisms within the population. To investigate single-cell diversity in the visual cortex, we used unsupervised clustering based on each neuron's encoding pattern, represented as a 5-element vector that included (cross-validated) explained variance from each feature category (stimulus, behavior, same-area LFPs, and population activity from other visual areas) and the number of categories with more than 10% explanatory power (Figure 9A). Clustering based on these encoding profiles revealed six distinct groups (Figures 5G, 9B): one dominated by stimulus, one by behavior, another with high encoding of both stimulus and behavior, a group influenced by both stimulus and averaged population activity from neighboring visual areas, a group with high explained variance across all input feature categories (multi-source), and a final group comprising neurons where no single feature explained more than 10% of their variance. The two largest clusters comprised units predominantly driven by stimulus features alone (29.5%) and multi-source units (30.2%). These two clusters made up 60% of all units (Figure 5G). In contrast, units driven solely by behavior formed the smallest cluster, representing just 4.7% of all units, while 12.2% of units were jointly influenced by both stimulus and behavior features.

When examining the distribution of all neuron clusters across visual areas, we found that the fraction of units best predicted by stimulus features peaked in V1 (39.6% of units in V1), decreasing along the hierarchy (Figure 5H; LM: 33.8%, RL: 21%, AL: 24.8%, PM: 26%, AM: 21.3%; Pearson correlation with hierarchy score, $r_{p-RL} = -0.96$, $p_{p-RL} = 0.01$). Conversely, the influence of behavior increased along the hierarchy. Proportion of units driven by behavior alone (V1: 2.1%, LM: 5.3%, RL: 5.6%, AL: 5.8%, PM: 5.4%, AM: 5.9%; $r_p = 0.84$, $p = 0.03$) and of units affected by both stimulus and behavior nearly doubled in higher-order areas (V1: 7.7%, LM: 6.5%, RL: 17.9%, AL: 12.7%, PM: 16.7%, AM: 18.6%). Lastly, the proportion of multi-source units also increased along the hierarchy (V1: 28%, LM: 28.8%, RL: 30%, AL: 32.6%, PM: 31.4%, AM: 32.4%; $r_p = 0.85$, $p_p = 0.03$). These findings point to an increasing functional diversity among neurons as one ascends the visual hierarchy. Supporting this, neurons influenced by mul-

multiple factors, especially behavior, had larger receptive field sizes (Figure 9F, $p = 0.0001$), consistent with the known trend of increasing receptive field sizes along the hierarchy (Wang and Burkhalter 2007; Lennie 1998). Multi-source units also tended to have higher firing rates compared to neurons predominantly explained by a single factor (Figure 9D, $p = 4.3 \times 10^{-71}$). Finally, the ratio of RS to FS units was highest in the stimulus-driven cluster (8:1), exceeding the overall ratio of 4:1 in the neuronal population (Figure 9E). In contrast, clusters driven by behavior or influenced by multiple factors had a lower ratio (2:1), indicating that non-visual factors predominantly modulate FS units, while visual factors primarily modulate RS units.

Given the shift in neural dynamics across different brain states, we explored how the contributing factors to single-unit activity varied between these states. To do so, we performed a similar clustering analysis based on the state-specific explained variance of individual factors in both high- and low-frequency states (Figures 5I, 9H-K). This analysis revealed notable changes in the dominant factors contributing to single unit variance across states. In the low-frequency state, a large proportion of units (53%) fell into a cluster where no single feature explained more than 10% of their variance. However, in the high-frequency state, around 40% of these same units shifted to being predominantly driven by stimulus features alone. Additionally, units influenced by multiple sources in the low-frequency state became more specialized in the high-frequency state. These results indicate a significant state-dependent reorganization of neural representation in the visual cortex. "

Lines 456 - 474 from the discussion section: "Neurons in the visual cortex can be classified by several criteria, including their morphology, connectivity, developmental history, gene expression, intrinsic physiology, and in vivo encoding strategies. Single-cell RNA sequencing studies have revealed extensive cell-type diversity and their relationships within cortical circuits (Gouwens et al. 2020; Saunders et al. 2018). Different cell types have distinct functional roles, which are further influenced by their position within the cortical hierarchy and the specific inputs they receive across different layers (Pfeffer et al. 2013; Garrett et al. 2023; Ma et al. 2024; Kamigaki 2019; Olsen et al. 2012; Senzai et al. 2019). Furthermore, different neuronal types are modulated by various factors such as behavior (Musall et al. 2019; Stringer et al. 2019), top-down feedback (Garrett et al. 2023), and internal brain states (Ferguson et al. 2023). Through unsupervised clustering of each neuron's encoding patterns, we quantified their encoding diversity, uncovering units with specialized properties within the visual cortex (Figure 5G). Our findings indicate that neurons in the visual cortex are modulated by a diverse array of factors, with the relative contributions of these factors varying across states, hierarchical positions, layers and cell-types (Figures 5H,I, S9C, E). Additionally, we observed an increasing representation of pan-modulated units along the visual hierarchy (Figure 5H), suggesting that while integrative processes may start as early as V1, a larger network of neurons becomes involved in this process higher up the hierarchy. In line with recent studies suggesting that sensory-motor integration begins in early sensory areas (Stringer et al. 2019; Erisken et al. 2014; Saleem et al. 2013), we identified two distinct neuronal clusters likely involved in this process : one driven solely by behavior, and another influenced by both visual stimuli and behavioral factors (Figure 5G). These findings emphasize the complex and dynamic nature of visual processing, shaped by a multitude of internal and external factors."

Lastly, we have also clarified the key take-home messages in the abstract, introduction, and discussion sections to ensure the significance of our findings are clearly communicated (please see response and listed modifications to question 1).

Minor comments:

6. Fig 1 presents this behavior, state, and stimulus as a 3-D space. This gives the wrong intuition. Internal fluctuations are so closely related to behavior that this leads the reader in wrong direction.

Figure 1A is only an illustration of the paper's main focus, posing the question of how different sources contribute to trial-wise variability in the visual cortex and how these contributions change over time. Any correlations in this illustration are unintentional and are meant only to demonstrate a hypothetical curve. To avoid confusion, the axes corresponding to each source have been deliberately kept orthogonal. This approach is consistent with our analysis, where we ensure that all features are orthogonalized against one another (Lines 907 - 916).

7. Fig 1D: What are the different LFP traces? Is not described in legend.

The LFP traces are from randomly selected channels in V1. This has now been clarified in the text.

8. Fig 3E: What was the statistical test?

The statistical test used in Figure 3E is one-way ANOVA. This has now been clarified in the text.

9. Panel 4c, legend mentions P-value of 1.5

We thank the reviewer for pointing out this mistake. We have now updated Figure 4 to reflect results from the new control analysis, and all p-values have been accordingly reported.

10. Fig 6F seems bizarre in light of earlier FF result. How can 75% of V1 neurons have a value below 1? The grand average was substantially higher.

We'd like to clarify that the averages of the Fano factor (FF) values reported in Figure S9G (previously 6F) and Figure 4F are not directly comparable. In Figure S9, the FF was calculated across all units in all areas that showed high stimulus encoding, regardless of the mouse's internal state. In contrast, the grand averages (represented by the silver dashed line) reported in Figure 4F were computed across units and areas within specific states, without considering their encoding properties. Figure 6F has now been updated, and moved to Figure S9G.

11. Fig 7 is weird. It is 90% a summary of the methods used in this paper and only 10% a cartoon illustration of the results.

This figure has now been removed from the paper.

References

- Ashwood, Z. C., Roy, N. A., Stone, I. R., Urai, A. E., Churchland, A. K., Pouget, A., and Pillow, J. W. (2022). Mice alternate between discrete strategies during perceptual decision-making. *Nature Neuroscience* 2022 25:2, 25(2):201–212.
- Bair, W. (1999). Spike timing in the mammalian visual system. *Current Opinion in Neurobiology*, 9(4):447–453.
- Dahmen, D., Recanatesi, S., Jia, X., Ocker, G. K., Campagnola, L., Jarsky, T., Seeman, S., Helias, M., and Shea-Brown, E. (2022). Strong and localized recurrence controls dimensionality of neural activity across brain areas. *bioRxiv*.
- Dubreuil, A., Valente, A., Beiran, M., Mastrogiuseppe, F., and Ostojic, S. (2022). The role of population structure in computations through neural dynamics. *Nature Neuroscience*, 25(6):783–794.

- Engel, T. A. and Steinmetz, N. A. (2019). New perspectives on dimensionality and variability from large-scale cortical dynamics. *Current Opinion in Neurobiology*, 58:181–190.
- Erisken, S., Vaiceliunaite, A., Jurjut, O., Fiorini, M., Katzner, S., and Busse, L. (2014). Effects of Locomotion Extend throughout the Mouse Early Visual System. *Current Biology*, 24(24):2899–2907.
- Ferguson, K. A., Salameh, J., Alba, C., Selwyn, H., Barnes, C., Lohani, S., and Cardin, J. A. (2023). VIP interneurons regulate cortical size tuning and visual perception. *Cell Reports*, 42(9).
- Garrett, M., Groblewski, P., Piet, A., Ollerenshaw, D., Najafi, F., Yavorska, I., Amster, A., Bennett, C., Buice, M., Caldejon, S., Casal, L., DOrazi, F., Daniel, S., de Vries, S. E. J., Kapner, D., Kiggins, J., Lecoq, J., Ledochowitsch, P., Manavi, S., Mei, N., Morrison, C. B., Naylor, S., Orlova, N., Perkins, J., Ponvert, N., Roll, C., Seid, S., Williams, D., Williford, A., Ahmed, R., Amine, D., Billeh, Y., Bowman, C., Cain, N., Cho, A., Dawe, T., Departee, M., Desoto, M., Feng, D., Gale, S., Gelfand, E., Gradis, N., Grasso, C., Hancock, N., Hu, B., Hytten, R., Jia, X., Johnson, T., Kato, I., Kivikas, S., Kuan, L., LHeureux, Q., Lambert, S., Leon, A., Liang, E., Long, F., Mace, K., de Abril, I., Mochizuki, C., Nayan, C., North, K., Ng, L., Ocker, G. K., Oliver, M., Rhoads, P., Ronellenfitch, K., Schelonka, K., Sevigny, J., Sullivan, D., Sutton, B., Swapp, J., Nguyen, T. K., Waughman, X., Wilkes, J., Wang, M., Farrell, C., Wakeman, W., Zeng, H., Phillips, J., Mihalas, S., Arkhipov, A., Koch, C., and Olsen, S. R. (2023). Stimulus novelty uncovers coding diversity in visual cortical circuits. *bioRxiv*.
- Gouwens, N. W., Sorensen, S. A., Baftizadeh, F., Budzillo, A., Lee, B. R., Jarsky, T., Alfiler, L., Baker, K., Barkan, E., Berry, K., Bertagnolli, D., Bickley, K., Bomben, J., Braun, T., Brouner, K., Casper, T., Crichton, K., Daigle, T. L., Dalley, R., de Frates, R. A., Dee, N., Desta, T., Lee, S. D., Dotson, N., Egdorf, T., Ellingwood, L., Enstrom, R., Esposito, L., Farrell, C., Feng, D., Fong, O., Gala, R., Gamlin, C., Gary, A., Glandon, A., Goldy, J., Gorham, M., Graybuck, L., Gu, H., Hadley, K., Hawrylycz, M. J., Henry, A. M., Hill, D., Hupp, M., Kebede, S., Kim, T. K., Kim, L., Kroll, M., Lee, C., Link, K. E., Mallory, M., Mann, R., Maxwell, M., McGraw, M., McMillen, D., Mukora, A., Ng, L., Ngo, K., Nicovich, P. R., Oldre, A., Park, D., Peng, H., Penn, O., Pham, T., Pom, A., Popović, Z., Potekhina, L., Rajanbabu, R., Ransford, S., Reid, D., Rimorin, C., Robertson, M., Ronellenfitch, K., Ruiz, A., Sandman, D., Smith, K., Sulc, J., Sunkin, S. M., Szafer, A., Tieu, M., Torkelson, A., Trinh, J., Tung, H., Wakeman, W., Ward, K., Williams, G., Zhou, Z., Ting, J. T., Arkhipov, A., Sömbül, U., Lein, E. S., Koch, C., Yao, Z., Tasic, B., Berg, J., Murphy, G. J., and Zeng, H. (2020). Integrated Morphoelectric and Transcriptomic Classification of Cortical GABAergic Cells. *Cell*, 183(4):935–953.
- Hulsey, D., Zumwalt, K., Mazzucato, L., McCormick, D. A., and Jaramillo, S. (2024). Decision-making dynamics are predicted by arousal and uninstructed movements. *Cell Reports*, 43(2).
- Jia, X. and Kohn, A. (2011). Gamma Rhythms in the Brain. *PLOS Biology*, 9(4):e1001045–.
- Kamigaki, T. (2019). Dissecting executive control circuits with neuron types. *Neuroscience Research*, 141:13–22.
- Kara, P., Reinagel, P., and Reid, R. C. (2000). Low Response Variability in Simultaneously Recorded Retinal, Thalamic, and Cortical Neurons. *Neuron*, 27(3):635–646.
- Lennie, P. (1998). Single Units and Visual Cortical Organization. *Perception*, 27(8):889–935.
- Ma, G., Liu, Y., Wang, L., Xiao, Z., Song, K., Wang, Y., Peng, W., Liu, X., Wang, Z., Jin, S., Tao, Z., Li, C. T., Xu, T., Xu, F., Xu, M., and Zhang, S. (2024). Hierarchy in sensory processing reflected by innervation balance on cortical interneurons. *Science Advances*, 7(20):eabf5676.

- Mante, V., Sussillo, D., Shenoy, K. V., and Newsome, W. T. (2013). Context-dependent computation by recurrent dynamics in prefrontal cortex. *Nature*, 503(7474):78–84.
- Mazzucato, L., Fontanini, A., and La Camera, G. (2016). Stimuli Reduce the Dimensionality of Cortical Activity. *Frontiers in Systems Neuroscience*, 10.
- Mazzucato, L., La Camera, G., and Fontanini, A. (2019). Expectation-induced modulation of metastable activity underlies faster coding of sensory stimuli. *Nature Neuroscience*, 22(5):787–796.
- McGinley, M. J., David, S. V., and McCormick, D. A. (2015). Cortical Membrane Potential Signature of Optimal States for Sensory Signal Detection. *Neuron*, 87(1):179–192.
- McInnes, L., Healy, J., and Melville, J. (2018). UMAP: Uniform Manifold Approximation and Projection for Dimension Reduction.
- Monti, S., Tamayo, P., Mesirov, J., and Golub, T. (2003). Consensus Clustering: A Resampling-Based Method for Class Discovery and Visualization of Gene Expression Microarray Data. *Machine Learning*, 52(1):91–118.
- Musall, S., Kaufman, M. T., Juavinett, A. L., Gluf, S., and Churchland, A. K. (2019). Single-trial neural dynamics are dominated by richly varied movements. *Nature Neuroscience*, 22(10):1677–1686.
- Olsen, S. R., Bortone, D. S., Adesnik, H., and Scanziani, M. (2012). Gain control by layer six in cortical circuits of vision. *Nature*, 483(7387):47–52.
- Papadopoulos, L., Jo, S., Zumwalt, K., Wehr, M., McCormick, D. A., and Mazzucato, L. (2024). Modulation of metastable ensemble dynamics explains optimal coding at moderate arousal in auditory cortex. *bioRxiv*.
- Pfeffer, C. K., Xue, M., He, M., Huang, Z. J., and Scanziani, M. (2013). Inhibition of inhibition in visual cortex: the logic of connections between molecularly distinct interneurons. *Nature Neuroscience*, 16(8):1068–1076.
- Piet, A., Ponvert, N., Ollerenshaw, D., Garrett, M., Groblewski, P. A., Olsen, S., Koch, C., and Arkhipov, A. (2023). Behavioral strategy shapes activation of the Vip-Sst disinhibitory circuit in visual cortex. *bioRxiv*.
- Saleem, A. B., Ayaz, A. I., Jeffery, K. J., Harris, K. D., and Carandini, M. (2013). Integration of visual motion and locomotion in mouse visual cortex. *Nature Neuroscience*, 16(12):1864–1869.
- Satopaa, V., Albrecht, J., Irwin, D., and Raghavan, B. (2011). Finding a "Kneedle" in a Haystack: Detecting Knee Points in System Behavior. In *2011 31st International Conference on Distributed Computing Systems Workshops*, pages 166–171.
- Saunders, A., Macosko, E. Z., Wysoker, A., Goldman, M., Krienen, F. M., de Rivera, H., Bien, E., Baum, M., Bortolin, L., Wang, S., Goeva, A., Nemesh, J., Kamitaki, N., Brumbaugh, S., Kulp, D., and McCarroll, S. A. (2018). Molecular Diversity and Specializations among the Cells of the Adult Mouse Brain. *Cell*, 174(4):1015–1030.
- Schölvinck, M. L., Saleem, A. B., Benucci, A., Harris, K. D., and Carandini, M. (2015). Cortical state determines global variability and correlations in visual cortex. *Journal of Neuroscience*, 35(1):170–178.
- Senzai, Y., Fernandez-Ruiz, A., and Buzsáki, G. (2019). Layer-Specific Physiological Features and Inter-laminar Interactions in the Primary Visual Cortex of the Mouse. *Neuron*, 101(3):500–513.

- Shi, Y.-L., Steinmetz, N. A., Moore, T., Boahen, K., and Engel, T. A. (2022). Cortical state dynamics and selective attention define the spatial pattern of correlated variability in neocortex. *Nature Communications*, 13(1):44.
- Stringer, C., Pachitariu, M., Steinmetz, N., Reddy, C. B., Carandini, M., and Harris, K. D. (2019). Spontaneous behaviors drive multidimensional, brainwide activity. *Science*, 364(6437).
- Wang, Q. and Burkhalter, A. (2007). Area map of mouse visual cortex. *Journal of Comparative Neurology*, 502(3):339–357.
- Wyrick, D. and Mazzucato, L. (2021). State-Dependent Regulation of Cortical Processing Speed via Gain Modulation.
- Zeraati, R., Shi, Y.-L., Steinmetz, N. A., Gieselmann, M. A., Thiele, A., Moore, T., Levina, A., and Engel, T. A. (2023). Intrinsic timescales in the visual cortex change with selective attention and reflect spatial connectivity. *Nature Communications*, 14(1):1858.
- Zhuang, C., Yan, S., Nayebi, A., Schrimpf, M., Frank, M. C., DiCarlo, J. J., and Yamins, D. L. (2021). Unsupervised neural network models of the ventral visual stream. *Proceedings of the National Academy of Sciences of the United States of America*, 118(3).